# A Tale of Two Geometries: Adaptive Optimizers and Non-Euclidean Descent

**Shuo Xie**[1]* **Tianhao Wang**[2]* **Beining Wu**[3] **Zhiyuan Li**[1]

[1]Toyota Technological Institute at Chicago  [2]University of California, San Diego
[3]University of Chicago
{shuox,zhiyuanli}@ttic.edu, tianhaowang@ucsd.edu, beiningw@uchicago.edu

## Abstract

Adaptive optimizers can reduce to normalized steepest descent (NSD) when only adapting to the current gradient, suggesting a close connection between the two algorithmic families. A key distinction between their analyses, however, lies in the geometries, e.g., smoothness notions, they rely on. In the convex setting, adaptive optimizers are governed by a stronger adaptive smoothness condition, while NSD relies on the standard notion of smoothness. We extend the theory of adaptive smoothness to the nonconvex setting and show that it precisely characterizes the convergence of adaptive optimizers. Moreover, we establish that adaptive smoothness enables acceleration of adaptive optimizers with Nesterov momentum in the convex setting, a guarantee unattainable under standard smoothness for certain non-Euclidean geometry. We further develop an analogous comparison for stochastic optimization by introducing adaptive gradient variance, which parallels adaptive smoothness and leads to dimension-free convergence guarantees that cannot be achieved under standard gradient variance for certain non-Euclidean geometry.

## 1 Introduction

Adaptive optimizers such as Adam have been indispensable for training large-scale machine learning models (Bi et al., 2024; Dubey et al., 2024; Yang et al., 2025; Wen et al., 2025). Their dominance in training efficiency, however, has recently been challenged by the surprising effectiveness of simpler Normalized Steepest Descent (NSD)-type methods such as Muon and Lion (Jordan et al., 2024; Chen et al., 2023; Team et al., 2025; Liu et al., 2025; Shah et al., 2025). Behind this competition of two family of optimizers, a broader consensus has begun to emerge: their superior performance is critically related to their ability to exploit non-Euclidean geometry of the loss landscape (Balles et al., 2020; Xie & Li, 2024; Zhang et al., 2024; Pethick et al., 2025).

Recent studies have rigorously characterized how adaptive optimizers exploit non-Euclidean geometry. For example, Maladkar et al. (2024) and Xie et al. (2025a) show that AdaGrad and Adam benefit from exploiting the $\ell_\infty$-geometry of loss functions, and a one-sided variant of Shampoo has been shown to leverage the geometry induced by the matrix spectral norm (Xie et al., 2025b; An et al., 2025). Notably, Bernstein & Newhouse (2024) proposed a striking connection between adaptive optimizers and NSD: with exponential moving average (EMA) turned off, certain adaptive optimizers reduce exactly to their NSD counterparts. For example, without EMA, Adam coincides with NSD under the $\ell_\infty$ norm, and Shampoo coincides with NSD under the matrix spectral norm, which is proposed to be an independent algorithm Muon. Yet, beyond these connections, there is no formal result that systematically characterizes the relationship between the two families of algorithms. This naturally motivates the following question:

**Q1.** *Do adaptive methods (like Adam, Shampoo) and their corresponding non-Euclidean descent (like Lion, Muon) exploit the non-Euclidean geometry of loss landscape in the same way?*

To address this question, we adopt a theoretical perspective and focus on comparing different types of smoothness assumptions that underpin the analysis of these methods. In fact, even under the

---

*Equal contribution.

same geometry, two distinct notions of smoothness arise. The first is the standard smoothness under a general norm (cf. Definition 2.3), which governs the convergence rate of NSD. The second is called the *adaptive smoothness* (cf. Definition 2.4), introduced by Xie et al. (2025b) and shown to govern the convergence rate of adaptive optimizers in the convex case. Indeed, a main contribution of our work is to show that adaptive smoothness also characterizes the convergence rate of adaptive optimizers in the nonconvex setting. Therefore, while both adaptive optimizers and NSD can exploit non-Euclidean geometry, they rely on fundamentally different smoothness assumptions.

This difference is not merely terminological but quantitative: adaptive smoothness is always no smaller than the standard smoothness under the same geometry. In other words, from the standpoint of technical conditions, the adaptive smoothness represents a stronger assumption, which in turn motivates our second question:

**Q2.** *Does the stronger smoothness assumption in adaptive methods offer optimization benefit?*

We answer this question affirmatively. In particular, we show that by leveraging Nesterov acceleration, adaptive optimizers can attain an accelerated $O(T^{-2})$ rate under adaptive smoothness in the convex setting. In sharp contrast, it has been shown by Guzmán & Nemirovski (2015) that the convergence rate of any optimizer is no better than $\Omega(T^{-1})$ under the standard $\ell_\infty$ smoothness assumption. This establishes a clear separation: adaptive smoothness enables adaptive optimizers to achieve acceleration under non-Euclidean geometry, while the standard smoothness fails. Therefore, the stronger adaptive smoothness assumption indeed translates into concrete optimization benefits, showing its difference from the standard smoothness.

In fact this difference has a direct and interesting analogy in terms of the noise assumption in the stochastic setting. When gradient noise is present, its variability can be measured in two distinct ways: the standard variance considers gradient variation under a fixed norm, whereas the *adaptive variance* (cf. Definition 4.1) measures noise in a more stringent but also more adaptive way that requires uniform control over the geometry prescribed by each preconditioner under consideration. By construction, adaptive variance is always no smaller than standard variance, directly paralleling the relationship between adaptive and standard smoothness. Analogous to adaptive smoothness that enables acceleration under a stronger requirement, adaptive variance can likewise yield benefits despite being larger. We demonstrate this through a careful analysis of NSD under two types of noise assumptions: adaptive variance enables a dimension-free rate, which is not attainable in the worst case under the standard variance condition.

Taken together, our results demonstrate that adaptive smoothness and adaptive variance are different from their standard counterparts as adaptive smoothness enables an acceleration rate and adaptive noise enables a dimension-free rate. These findings reveal an intricate interplay between adaptivity and non-Euclidean geometry, deepening our theoretical understanding of adaptivity in optimization.

Below we summarize our main contributions.

- In Section 3, we show the convergence rate for adaptive optimizers on nonconvex functions (Theorems D.2, D.7 and D.8), which depends on the adaptive smoothness and matches optimal $\tilde{O}(T^{-1/4})$ rate. It theoretically justifies that adaptive methods and NSD exploit the geometry through different smoothness notions in the nonconvex setting.

- In Section 4.2, we identify the benefit of the adaptive smoothness by showing it enables an acceleration rate $\tilde{O}(T^{-2})$ of adaptive optimizers equipped with Nesterov momentum (Theorem 4.3) in contrast to the convergence rate $\Omega(T^{-1})$ the standard $\ell_\infty$ smoothness.

- In Section 4, we extend the benefit of adaptive geometry to noise assumptions by introducing adaptive noise (Definition 4.1). We show that this stronger notion of noise can provide a new type of convergence rate for NSD with momentum on nonconvex functions which gets rid of dependence on parameter size $d$ (Theorem 4.5). We complement its superiorty by providing a lower bound under the standard noise (Theorem 4.7).

- Our analysis of adaptive optimizers is carried out through a unified framework that covers a broad class of methods, including AdaGrad, AdaGrad-Norm, and one-sided Shampoo. The proof technique developed in this framework may be of independent interest.

## 1.1 NOTATIONS

Let $\mathcal{M}^d$ be the set of all $d$-by-$d$ matrices, $\mathcal{S}^d \subset \mathcal{M}^d$ be the subset of all symmetric matrices. We use $\mathcal{S}^d_+$ to denote the set of positive semi-definite matrices. We denote by $\boldsymbol{I}_d \in \mathcal{M}^d$ the identity matrix. For matrices $\boldsymbol{A}, \boldsymbol{B}$, we denote their inner product by $\langle \boldsymbol{A}, \boldsymbol{B} \rangle = \mathrm{Tr}(\boldsymbol{A}^\top \boldsymbol{B})$.

For $\boldsymbol{H} \in \mathcal{S}^d_+$, $\|\boldsymbol{x}\|_{\boldsymbol{H}} := \sqrt{\boldsymbol{x}^\top \boldsymbol{H} \boldsymbol{x}}$ is the (semi-)norm of $\boldsymbol{x} \in \mathbb{R}^d$ with respect to $\boldsymbol{H}$. For a convex set $\mathcal{H} \subseteq \mathcal{S}^d_+$, we define the induced $\mathcal{H}$-norm as

$$\|\boldsymbol{x}\|_{\mathcal{H}} := \sup_{\boldsymbol{H} \in \mathcal{H}, \mathrm{Tr}(\boldsymbol{H}) \leq 1} \|\boldsymbol{x}\|_{\boldsymbol{H}} . \tag{1}$$

Throughout the paper, we reserve $f$ for the loss function and $\boldsymbol{x}_0$ for the initialization of an optimization algorithm. For convenience, we denote the initial suboptimality as $\Delta_0 = f(\boldsymbol{x}_0) - \min_{\boldsymbol{x}} f(\boldsymbol{x})$.

## 2 FROM ADAM/SIGNGD TO ADAPTIVE SMOOTHNESS

We use the example of Adam and SignGD to motivate the notion of adaptive smoothness in Section 2.1, and then present the formal definition in Section 2.2, along with some related background.

### 2.1 ADAM AND SIGNGD CAN EXPLOIT $\ell_\infty$ GEOMETRY, BUT IN DIFFERENT WAYS

We start by discussing a specific pair of algorithms, Adam and SignGD, to illustrate the problem of interest. It is known that SignGD can be viewed as Normalized Steepest Descent (NSD) under the $\ell_\infty$ norm and its convergence rate for deterministic nonconvex functions admits the following form (Xie et al., 2025a)

$$\min_{t \in [T]} \|\nabla f(\boldsymbol{x}_t)\|_1 \leq O\left( \sqrt{\frac{\Delta_0 L_{\|\cdot\|_\infty}(f)}{T}} \right)$$

where $L_{\|\cdot\|_\infty}(f)$ is the standard smoothness of $f$ under the $\ell_\infty$ norm (see Definition 2.3). Note that SignGD can also be viewed as a special case of Adam with $\beta_1 = \beta_2 = 0$. However, the convergence rate of Adam for general $\beta_1, \beta_2$ instead depends on a different diagonal adaptive smoothness notion, which is defined as $L_{\mathrm{diag}}(f) = \min_{\boldsymbol{H} \in \mathcal{D}^d, -\boldsymbol{H} \preceq \nabla^2 f(\boldsymbol{x}) \preceq \boldsymbol{H}} \mathrm{Tr}(\boldsymbol{H})$ in Maladkar et al. (2024); Xie et al. (2025a). In particular, Adam with $\beta_1 = 0$ (a.k.a. RMSProp) for deterministic nonconvex functions admits the convergence rate $\min_{t \in [T]} \|\nabla f(\boldsymbol{x}_t)\|_1 = \tilde{O}(\sqrt{\Delta_0 L_{\mathrm{diag}}(f)/T})$ (Xie et al., 2024). Notably, this diagonal adaptive smoothness is always no smaller than $L_{\|\cdot\|_\infty}(f)$ (Balles et al., 2020). This suggests that though both SignGD and Adam admit convergence guarantees for the $\ell_1$ norm (the dual norm of $\|\cdot\|_\infty$) of the gradients, they achieve so under different smoothness notions. This distinction motivates the following question:

*How does the diagonal adaptive smoothness $L_{\mathrm{diag}}(f)$ emerge as an $\ell_\infty$ geometry?*

To address this question, let us consider the convergence rate of NSD under any norm $\|\cdot\|_{\boldsymbol{H}}$ for $\boldsymbol{H} \in \mathcal{H} = \mathcal{D}^d_+$ (see Theorem 4.5):

$$\min_{t \in [T]} \|\nabla f(\boldsymbol{x}_t)\|_{\boldsymbol{H}, *} = O\left( \sqrt{\frac{\Delta_0 L_{\|\cdot\|_{\boldsymbol{H}}}(f)}{T}} \right) \tag{2}$$

where $\|\cdot\|_{\boldsymbol{H}, *}$ is the dual norm of $\|\cdot\|_{\boldsymbol{H}}$. Minimizing both sides of (2) over $\boldsymbol{H} \in \mathcal{D}^d_+$ with $\mathrm{Tr}(\boldsymbol{H}) \leq 1$ yields

$$\inf_{\substack{\text{diagonal } \boldsymbol{H} \succeq 0 \\ \mathrm{Tr}(\boldsymbol{H}) \leq 1}} \min_{t \in [T]} \|\nabla f(\boldsymbol{x}_t)\|_{\boldsymbol{H}, *} = O\left( \sqrt{\frac{\Delta_0}{T} \inf_{\substack{\text{diagonal } \boldsymbol{H} \succeq 0 \\ \mathrm{Tr}(\boldsymbol{H}) \leq 1}} L_{\|\cdot\|_{\boldsymbol{H}}}(f)} \right) = O\left( \sqrt{\frac{\Delta_0 L_{\mathrm{diag}}(f)}{T}} \right) \tag{3}$$

where the equality can be checked by the definition of $L_{\mathrm{diag}}(f)$. Now the right-hand side matches the aforementioned convergence rate of Adam. The adaptivity of Adam is then demonstrated by its ability to automatically identify and adapt to the best diagonal matrix-induced norm for any given loss function, without the need of knowing $\boldsymbol{H}$.

Importantly, the left-hand side of (3) is closely related to the $\ell_1$ norm of the gradients because

$$\sup_{\text{diagonal } \boldsymbol{H} \succeq 0, \text{Tr}(\boldsymbol{H}) \leq 1} \| \cdot \|_{\boldsymbol{H}} = \| \cdot \|_\infty, \qquad \inf_{\text{diagonal } \boldsymbol{H} \succeq 0, \text{Tr}(\boldsymbol{H}) \leq 1} \| \cdot \|_{\boldsymbol{H},*} = \| \cdot \|_1. \qquad (4)$$

We illustrate this fact in Fig. 1. In words, this means that *the $\ell_\infty$ norm is the pointwise supremum of all weighted $\ell_2$ norms induced by diagonal matrices with unit trace, whereas its dual, the $\ell_1$ norm, is the pointwise infimum of all the corresponding dual norms.* Also, the unit $\ell_\infty$ ball is the intersection of all unit balls for those $\ell_2$ norms, and the unit $\ell_1$ ball is the union of all dual unit balls.

Indeed, the duality between supremum of a class of primal norms and infimum of the corresponding dual norms in (4) is not just a coincidence, but rather a special property induced by the structure of the preconditioner set $\mathcal{H} = \mathcal{D}_+^d$ for Adam. This property holds more generally for any well-structured preconditioner set and we discuss the corresponding adaptive smoothness in Section 2.2.

## 2.2 ADAPTIVE SMOOTHNESS ASSOCIATED WITH WELL-STRUCTURED PRECONDITIONER

The following definition of well-structured preconditioner sets is proposed by Xie et al. (2025b) to unify the analysis of a broad family of adaptive optimizers with structured preconditioners.

**Definition 2.1** (Well-structured preconditioner set). $\mathcal{H} \subseteq \mathcal{S}_+^d$ *is said to be a* well-structured precon­ditioner set *if* $\mathcal{H} = \mathcal{S}_+^d \cap \mathcal{K}$ *for some matrix subalgebra*[1] $\mathcal{K} \subseteq \mathcal{M}^d$ *with* $\boldsymbol{I}_d \in \mathcal{K}$.

As will be discussed in Section 3.1, many commonly used adaptive optimizers, including Adam, AdaGrad, and their variants, can be cast into the framework of a meta-algorithm (Algorithm 1) with well-structured preconditioner sets. A specific case is $\mathcal{H} = \mathcal{D}_+^d$, the set of all diagonal PSD matrices, which is the running example in the previous subsection. For any such well-structured pre­conditioner set $\mathcal{H}$, we have the duality between the supremum of the primal norms and the infimum of the corresponding dual norms, formalized in the following lemma.

**Lemma 2.2.** *Let* $\mathcal{H} \subseteq \mathcal{S}_+^d$ *be any well-structured preconditioner set. Recall that its induced norm is defined as* $\| \cdot \|_{\mathcal{H}} = \sup_{\boldsymbol{H} \in \mathcal{H}, \text{Tr}(\boldsymbol{H}) \leq 1} \| \cdot \|_{\boldsymbol{H}}$. *Then it holds that*

$$\| \cdot \|_{\mathcal{H},*} = \inf_{\boldsymbol{H} \in \mathcal{H}, \text{Tr}(\boldsymbol{H}) \leq 1} \| \cdot \|_{\boldsymbol{H},*} = \inf_{\boldsymbol{H} \in \mathcal{H}, \text{Tr}(\boldsymbol{H}) \leq 1} \| \cdot \|_{\boldsymbol{H}^{-1}}.$$

Based on this fact, we can generalize the discussion in Section 2.1 to any well-structured precondi­tioner set $\mathcal{H}$, showing that NSD and adaptive optimizers with preconditioner set $\mathcal{H}$ can exploit the geometry induced by $\| \cdot \| = \| \cdot \|_{\mathcal{H}}$ via two different smoothness notions, the former being the stan­dard smoothness under $\| \cdot \|_{\mathcal{H}}$ and the latter being the adaptive smoothness defined in Definition 2.4.

We proceed to introduce the adaptive smoothness associated with any well-structured preconditioner set $\mathcal{H}$. We first review the standard smoothness notion under a general norm $\|\cdot\|$.

**Definition 2.3.** *For a loss function* $f : \mathbb{R}^d \to \mathbb{R}$ *and any norm* $\|\cdot\|$, *we will use* $L_{\|\cdot\|}(f)$ *to denote the smoothness of* $f$ *with respect to* $\|\cdot\|$, *i.e., the smallest positive constant* $L$ *such that* $\|\nabla f(\boldsymbol{x}) - \nabla f(\boldsymbol{y})\|_* \leq L \|\boldsymbol{x} - \boldsymbol{y}\|$ *for any* $\boldsymbol{x}, \boldsymbol{y}$.

When $\| \cdot \| = \| \cdot \|_{\mathcal{H}}$ for some well-structured preconditioner set $\mathcal{H}$, $L_{\|\cdot\|_{\mathcal{H}}}(f)$ is then the standard smoothness of $f$ under the norm induced by $\mathcal{H}$. In contrast, the adaptive smoothness associated with $\mathcal{H}$ is defined as the smallest smoothness of $f$ under all norms $\| \cdot \|_{\boldsymbol{H}}$ induced by $\boldsymbol{H} \in \mathcal{H}$ with $\text{Tr}(\boldsymbol{H}) \leq 1$, as formalized below. This term is introduced as $\mathcal{H}$-smoothness in Xie et al. (2025b). We rename it to highlight this notion adapts to the structure of $\mathcal{H}$, in contrast to the standard smoothness.

**Definition 2.4** (Adaptive Smoothness, Xie et al. 2025b). *The adaptive smoothness of a function* $f$ *w.r.t. a well-structured preconditioner set* $\mathcal{H}$ *is defined as the smallest smoothness of* $f$ *under all* $\| \cdot \|_{\boldsymbol{H}}$ *for* $\boldsymbol{H} \in \mathcal{H}$ *with* $\text{Tr}(\boldsymbol{H}) \leq 1$, *that is,*

$$\Lambda_{\mathcal{H}}(f) := \min_{\substack{\boldsymbol{H} \in \mathcal{H} \\ \text{Tr}(\boldsymbol{H}) \leq 1}} L_{\|\cdot\|_{\boldsymbol{H}}}(f) = \min_{\substack{\boldsymbol{H} \in \mathcal{H} \\ \forall \boldsymbol{x}, -\boldsymbol{H} \preceq \nabla^2 f(\boldsymbol{x}) \preceq \boldsymbol{H}}} \text{Tr}(\boldsymbol{H}). \qquad (5)$$

---

[1] For a set of $d$-by-$d$ matrices $\mathcal{K} \subseteq \mathcal{M}^d$, we say that $\mathcal{K}$ is a *subalgebra* if it is closed under scalar mul­tiplication, matrix addition, and matrix multiplication. More concretely, we require that for any $\alpha \in \mathbb{R}$ and $\boldsymbol{A}, \boldsymbol{B} \in \mathcal{K}$, it holds that $\alpha \boldsymbol{A}, \boldsymbol{A}\boldsymbol{B}, \boldsymbol{A} + \boldsymbol{B} \in \mathcal{K}$.

---

**Algorithm 1** General Adaptive Optimization Algorithm

---

**Hyperparam:** $\epsilon \geq 0$, total steps $T$, learning rate $\eta$, convex cone $\mathcal{H} \subset \mathcal{S}_+$, decay factor $\beta$
**Input:** initialization $\boldsymbol{x}_0$, stochastic loss functions $\{f_t\}_{t=1}^T : \mathbb{R}^d \to \mathbb{R}$

  $\boldsymbol{M}_{-1} \leftarrow \boldsymbol{0}$
  **for** $t = 0, 1, \cdots, T-1:$
    $\boldsymbol{g}_t \leftarrow \nabla f_t(\boldsymbol{x}_t)$
$$\boldsymbol{M}_t \leftarrow \begin{cases} \boldsymbol{M}_{t-1} + \boldsymbol{g}_t\boldsymbol{g}_t^\top, & \text{Cumulative variant,} \\ \beta\,\boldsymbol{M}_{t-1} + (1-\beta)\,\boldsymbol{g}_t\boldsymbol{g}_t^\top, & \text{EMA variant,} \\ \beta\,\boldsymbol{M}_{t-1} + \boldsymbol{g}_t\boldsymbol{g}_t^\top, & \text{Weighted variant.} \end{cases}$$
    $\boldsymbol{V}_t \leftarrow \arg\min_{\boldsymbol{H} \in \mathcal{H}} \left\langle \boldsymbol{M}_t + \epsilon \boldsymbol{I}_d, \boldsymbol{H}^{-1} \right\rangle + \mathrm{Tr}(\boldsymbol{H})$
    $\boldsymbol{x}_{t+1} \leftarrow \boldsymbol{x}_t - \eta \boldsymbol{V}_t^{-1} \boldsymbol{g}_t$
  **return** $\boldsymbol{x}_T$

---

In the deterministic convex setting, it has been shown by Xie et al. (2025b) that the convergence rate of an adaptive optimizer with any well-structured preconditioner set $\mathcal{H}$ is of order $O(\Lambda_\mathcal{H}(f)\,\|\mathcal{X}\|_\mathcal{H}^2 / T)$. In Section 3, we extend such characterization to the nonconvex setting, demonstrating that the adaptive smoothness $\Lambda_\mathcal{H}(f)$ governs the convergence behavior of any adaptive optimizer with well-structured preconditioner set $\mathcal{H}$.

**Comparison between two smoothness notions.** For any $\boldsymbol{H} \in \mathcal{H}$ with $\mathrm{Tr}(\boldsymbol{H}) = 1$, it always holds $\|\boldsymbol{x} - \boldsymbol{y}\|_\mathcal{H} \geq \|\boldsymbol{x} - \boldsymbol{y}\|_{\boldsymbol{H}}$ and $\|\nabla f(\boldsymbol{x}) - \nabla f(\boldsymbol{y})\|_{\mathcal{H},*} \leq \|\nabla f(\boldsymbol{x}) - \nabla f(\boldsymbol{y})\|_{\boldsymbol{H},*}$. Therefore,

$$L_{\|\cdot\|_{\boldsymbol{H}}}(f) = \sup_{\boldsymbol{x},\boldsymbol{y}} \frac{\|\nabla f(\boldsymbol{x}) - \nabla f(\boldsymbol{y})\|_{\boldsymbol{H},*}}{\|\boldsymbol{x} - \boldsymbol{y}\|_{\boldsymbol{H}}} \geq \sup_{\boldsymbol{x},\boldsymbol{y}} \frac{\|\nabla f(\boldsymbol{x}) - \nabla f(\boldsymbol{y})\|_{\mathcal{H},*}}{\|\boldsymbol{x} - \boldsymbol{y}\|_\mathcal{H}} = L_{\|\cdot\|_\mathcal{H}}(f).$$

Minimizing over $\boldsymbol{H} \in \mathcal{H}$ with $\mathrm{Tr}(\boldsymbol{H}) = 1$ then yields $\Lambda_\mathcal{H}(f) = L_{\|\cdot\|_{\boldsymbol{H}}}(f) \geq L_{\|\cdot\|_\mathcal{H}}(f)$. In other words, as a condition, the adaptive smoothness is arguably stronger than the standard smoothness. But they can differ by at most a multiplicative factor of $d$, as summarized in Proposition 2.5.

**Proposition 2.5.** *For any well-structured preconditioner set $\mathcal{H} \subseteq \mathcal{S}_+^d$ and any loss function $f : \mathbb{R}^d \to \mathbb{R}$, it always holds that $L_{\|\cdot\|_\mathcal{H}}(f) \leq \Lambda_\mathcal{H}(f) \leq d \cdot L_{\|\cdot\|_\mathcal{H}}(f)$.*

## 3 UNIFIED ANALYSIS IN THE NONCONVEX SETTING

In the nonconvex setting, we establish a *unified* analysis that encompasses a broad family of adaptive optimization algorithms. Our result highlights how the convergence behavior of these methods depends critically on the notion of adaptive smoothness.

### 3.1 ADAPTIVE OPTIMIZERS WITH WELL-STRUCTURED PRECONDITIONER SETS

We adopt the framework in Gupta et al. (2017) and Xie et al. (2025b) to describe adaptive optimizers in a unified way, as displayed in Algorithm 1. This meta-algorithm is flexible in two aspects: the way of aggregating past gradients and the choice of preconditioner set $\mathcal{H}$. First, there are three different ways to aggregate the past gradients in Algorithm 1, each of which is presented in a separate algorithm block in Appendix D.1. The cumulative and EMA variants are the most common ways, and they are indeed equivalent to the weighted variant up to hyperparameter transformations. Therefore, it suffices to study the weighted variant, and the results for the other two variants follow as corollaries. Another flexibility of Algorithm 1 comes from the choice of convex cone $\mathcal{H}$. More specifically, Algorithm 1 recovers several standard optimizers by specifying $\mathcal{H}$ as follows:

- $\mathcal{H} = \{$all diagonal PSD matrices$\}$ recovers AdaGrad and Adam.
- $\mathcal{H} = \{c\,\boldsymbol{I}_d \mid c > 0\}$ recovers AdaGrad-Norm and AdaSGD (Wang & Wiens, 2020).
- $\mathcal{H} = \mathcal{S}_+^d$ recovers full-matrix AdaGrad (Duchi et al., 2011).
- $\mathcal{H} = \mathcal{S}_+^{d_L} \otimes \boldsymbol{I}_{d_R}$ yields one-sided Shampoo/ASGO recently proposed by (Xie et al., 2025a; An et al., 2025)

In particular, based on the notion of well-structured preconditioner sets defined in Definition 2.1, Xie et al. (2025b) develops a unified convergence analysis for Algorithm 4 in the convex setting, and the convergence rate depends on the adaptive smoothness with respect to $\mathcal{H}$ defined in Definition 2.4.

**Additional notations.** We define $P_{\mathcal{H}}(M) := \arg\min_{H\in\mathcal{H}} \langle M, H^{-1}\rangle + \mathrm{Tr}(H)$ for any $M \in \mathcal{S}^d_{++}$. Then in Algorithm 1, $V_t = P_{\mathcal{H}}(M_t)$ and Lemma B.4 will show that $P_{\mathcal{H}}(M)^2$ is the projection of $M$ onto $\mathcal{H}$. Specifically, when $\mathcal{H}$ contains all the PSD matrices, $V_t$ is $M_t^{\frac{1}{2}}$.

## 3.2 CONVERGENCE RATE IN THE DETERMINISTIC NONCONVEX SETTING

Here we only present results for the deterministic case to highlight the role of adaptive smoothness, and the complete results for the (stochastic) nonconvex setting and corresponding proofs can be found in Appendix D.2. We first present the convergence guarantee for the weighted variant of Algorithm 1 in the following theorem.

**Theorem 3.1.** *For any $\epsilon \geq 0$, $\beta \in (0,1]$, $\eta > 0$, and $T \in \mathbb{N}$, let $\{x_t\}_{t=0}^T$ be the iterates of Algorithm 1 with well-structured preconditioner set $\mathcal{H}$, where the update of $M_t$ follows the weighted version, i.e., $M_t = \beta M_{t-1} + g_t g_t^\top$ for all $t \in [T]$. Let $\Lambda_{\mathcal{H}}(f)$ be the adaptive smoothness of the loss $f$ according to Definition 2.4. Then when $f_t \equiv f$, it holds that*

$$\frac{1}{T}\sum_{t=0}^{T-1} \|\nabla f(x_t)\|_{\mathcal{H},*} \leq \frac{\sqrt{\sum_{i=0}^{T-1}\beta^{i/2}}}{T}\xi + \frac{\sqrt{d}\epsilon^{1/4}}{\sqrt{T}}\sqrt{\xi}.$$

*where $\xi = \frac{2\Delta_0}{\eta} + \eta \Lambda_{\mathcal{H}}(f)\|S_T\|_{\mathrm{op}}$ and $S_T = \mathbb{E}\sum_{t=0}^{T-1} V_t^{-1}(V_t^2 - \beta V_{t-1}^2)V_t^{-1}$.*

*For general well-structured preconditioner set, $\|S_T\|_{\mathrm{op}} = \tilde{O}(\log(d)[(1-\beta)T/\beta + \log(d)])$. When the preconditioner set only has diagonal matrices, $\|S_T\|_{\mathrm{op}} = (1-\beta)T + \tilde{O}(1)$.*

The above result for the weight variant can be converted to guarantees for the cumulative and EMA variants. Specifically, the cumulative variant is equivalent to weighted accumulation with $\beta = 1$ while the EMA variant with learning rate $\eta^E$ and stability constant $\epsilon^E$ produces identical iterates as weighted accumulation with $\eta^W = \eta^E/\sqrt{1-\beta}$ and $\epsilon^W = \epsilon^E/(1-\beta)$. Below we present the result for the cumulative variant in Theorem 3.2, and the result for the EMA variant is in Theorem D.8.

**Theorem 3.2.** *For any $\epsilon \geq 0$, $\eta > 0$, and $T \in \mathbb{N}$, let $\{x_t\}_{t=0}^T$ be the iterates of Algorithm 1 with well-structured preconditioner set $\mathcal{H}$, where the update of $M_t$ follows the cumulative version, i.e., $M_t = M_{t-1} + g_t g_t^\top$ for all $t \in [T]$. Let $\Lambda_{\mathcal{H}}(f)$ be the adaptive smoothness of the loss $f$ according to Definition 2.4. Then when $f_t \equiv f$, it holds that*

$$\frac{1}{T}\sum_{t=0}^{T-1} \|\nabla f(x_t)\|_{\mathcal{H},*} \leq \frac{1}{\sqrt{T}}\left(\xi + \sqrt{d}\epsilon^{1/4}\sqrt{\xi}\right)$$

*where $\xi = \tilde{O}\left(\frac{\Delta_0}{\eta} + \eta \cdot \Lambda_{\mathcal{H}}(f)\log^2 d\right)$. Moreover, when setting $\eta = \sqrt{\frac{\Delta_0}{\Lambda_{\mathcal{H}}(f)\log^2 d}}$, it holds that $\xi = \tilde{O}\left(\sqrt{\Delta_0 \cdot \Lambda_{\mathcal{H}}(f)}\log d\right)$.*

At a high level, Theorem 3.2 shows that, with appropriate hyperparameters, Algorithm 1 with any well-structured preconditioner set $\mathcal{H}$ achieves a convergence rate of order $\tilde{O}(\log d \cdot \sqrt{\Delta_0 \Lambda_{\mathcal{H}}(f)/T})$ on deterministic nonconvex functions where $\tilde{O}(\cdot)$ hides logarithmic factors in problem parameters other than the dimension $d$. This result illustrates that the adaptive smoothness $\Lambda_{\mathcal{H}}(f)$ governs the convergence rate of adaptive optimizers in the nonconvex setting, complementing previous results for the convex setting in Xie et al. (2025b). Moreover, we remark that when $\mathcal{H}$ contains only diagonal matrices, the $\log d$ factor disappears, recovering the bounds in Xie et al. (2025a).

It is worth noticing that the convergence guarantees in the above two theorems are concerned with $\|\nabla f(x_t)\|_{\mathcal{H},*}$ depending on specific $\mathcal{H}$ rather than $\|\nabla f(x_t)\|_2$. For the specific case of Adam where $\mathcal{H}$ is the set of all diagonal PSD, this becomes a guarantees in terms of $\ell_1$ norm of the gradients, as we discussed in Section 2.1. On the other hand, Pethick et al. (2025); Kovalev (2025a) show that NSD achieves $O\left((\frac{\Delta_0 L_{\|\cdot\|_{\mathcal{H}}}(f)}{T})^{\frac{1}{2}}\right)$ in the deterministic case. Taken together, these two rates suggest that adaptive optimizers and NSD exploit different smoothness notions in the nonconvex setting.

### 3.3 TECHNICAL CONTRIBUTION: A NOVEL MATRIX INEQUALITY

Previous theoretical results for one-sided Shampoo/ASGO (Algorithm 7) and other well-structured preconditioners primarily focus on convex objectives (Xie et al., 2025b; Kovalev, 2025a). In the nonconvex regime, existing convergence analyses apply essentially when the preconditioner set contains only diagonal matrices[2] (Xie et al., 2025a). In contrast, we show the first unified convergence analysis that applies to any general well-structured preconditioner set, well beyond the diagonal cases.

A central difficulty in our analysis is the extension from diagonal preconditioners to a general preconditioner set $\mathcal{H}$. In the diagonal case, the proof basically decomposes to entry-wise analyses, and scalar telescoping readily yields the desired bounds. However, for general $\mathcal{H}$, *noncommutativity prevents such simplification*, and bounding the second-order terms requires handling delicate matrix inequalities. Our resolution of this challenge yields a key technical contribution, formalized below.

**Lemma 3.3.** *Let $\epsilon \geq 0$ and $\beta \in (0,1]$. For any $T \in \mathbb{N}$, consider **any sequence of vectors** $\boldsymbol{g}_0, \ldots, \boldsymbol{g}_{T-1} \in \mathbb{R}^d$. Let $\boldsymbol{M}_{-1} = 0$, and recursively define $\boldsymbol{M}_t = \beta \boldsymbol{M}_{t-1} + \boldsymbol{g}_t \boldsymbol{g}_t^\top$ for $t = 0, \ldots, T-1$. For any well-structured preconditioner set $\mathcal{H}$, define $\boldsymbol{V}_t = \arg\min_{\boldsymbol{H} \in \mathcal{H}} \langle \boldsymbol{M}_t + \epsilon \boldsymbol{I}_d, \boldsymbol{H}^{-1} \rangle + \mathrm{Tr}(\boldsymbol{H})$ for each $t \in [T-1]$. Then for any $\boldsymbol{H} \in \mathcal{H} \cap \mathcal{S}_{++}^d$, it holds*

$$\sum_{t=0}^{T-1} \|\boldsymbol{V}_t^{-1} \boldsymbol{g}_t\|_{\boldsymbol{H}}^2 \leq \mathrm{Tr}(\boldsymbol{H}) \|\boldsymbol{S}_T\|_{\mathrm{op}} \quad \text{where } \boldsymbol{S}_T = \sum_{t=0}^{T-1} \boldsymbol{V}_t^{-1} \left(\boldsymbol{V}_t^2 - \beta \boldsymbol{V}_{t-1}^2\right) \boldsymbol{V}_t^{-1}.$$

*Moreover, there exists an absolute constant $C_1, C_2 > 0$, independent of $d$, $T$, $\epsilon$, $\beta$ and $\mathcal{H}$, such that*

$$\|\boldsymbol{S}_T\|_{\mathrm{op}} \leq C_1 \left(1 + \log\left(1 + \frac{d}{\epsilon}\sum_{t=0}^{T-1} \|\boldsymbol{g}_t\|_2^2 + d^2(1-\beta)T\right)\right) \left(\frac{1-\beta}{\beta}T + \log\|\boldsymbol{V}_{T-1}^2/\epsilon\|_{\mathrm{op}}\right) + C_2.$$

*In the special case when $\mathcal{H}$ is commutative, the above bound can be further improved to*

$$\|\boldsymbol{S}_T\|_{\mathrm{op}} \leq (1-\beta)T + \log\|\boldsymbol{V}_{T-1}^2/\epsilon\|_{\mathrm{op}}.$$

Specializing $\boldsymbol{g}_t$'s to be the gradients, Lemma 3.3 provides a general upper bound on the sum of second-order terms. It highlights the essential gap between diagonal and general preconditioner sets: noncommutativity introduces an additional $\log d$ factor, making the dependence strictly worse than in the diagonal case. Nevertheless, this is the first bound that applies to arbitrary well-structured preconditioner sets, and it plays a central role in extending convex analyses to the nonconvex setting.

The proof of Lemma 3.3 can be found in Appendix C. A key step is to establish a novel matrix inequality Lemma C.1 that relates the difference between two positive definite matrices to the difference between their logarithms, which may be of independent interest.

## 4 BENEFIT OF ADAPTIVE GEOMETRY

We have shown in Section 3 that the nonconvex convergence rate of adaptive optimizers relies critically on the adaptive smoothness of the loss, and the bound is worse than that of the corresponding NSD. This naturally raises the concern in Question 2: does the stronger adaptive smoothness lead to stronger results? In this section, we address this question from two complementary angles:

1. Under the adaptive smoothness assumption, adaptive optimizers can achieve faster convergence rates on convex functions via Nestrov acceleration.

2. The distinction between standard smoothness and adaptive smoothness mirrors a parallel separation in the assumptions on gradient noise.

At a high level, these two angles share the same underlying mechanism: *Under non-Euclidean geometry, averaging might not be effective in reducing the norm*, which we will explain below.

---

[2]This can be generalized to any commutative well-structured preconditioner set.

**Algorithm 2** Accelerated Adaptive Algorithm

**Hyperparam:** $\epsilon \geq 0$, total steps $T$, learning rate $\eta$, convex cone $\mathcal{H} \subseteq \mathcal{S}_+$

**Input:** initial $\boldsymbol{x}_0$, constants $\alpha_0, \ldots, \alpha_T \in (0, 1]$
$\quad \bar{\boldsymbol{x}}_0 \leftarrow \boldsymbol{x}_0, \boldsymbol{M}_{-1} \leftarrow \boldsymbol{0}$
$\quad$ **for** $t = 0, 1, \ldots, T - 1$ :
$\quad\quad \boldsymbol{g}_t \leftarrow \nabla f_t^{\alpha_t, \bar{\boldsymbol{x}}_t}(\boldsymbol{x}_t)$ where $f_t^{\alpha_t, \bar{\boldsymbol{x}}_t}$ is in (8)
$\quad\quad \boldsymbol{M}_t \leftarrow \boldsymbol{M}_{t-1} + \boldsymbol{g}_t \boldsymbol{g}_t^\top$
$\quad\quad \boldsymbol{V}_t \leftarrow \arg\min_{\boldsymbol{H} \in \mathcal{H}} \langle \boldsymbol{M}_t + \epsilon \boldsymbol{I}_d, \boldsymbol{H}^{-1} \rangle + \operatorname{Tr}(\boldsymbol{H})$
$\quad\quad \boldsymbol{x}_{t+1} \leftarrow \boldsymbol{x}_t - \eta \boldsymbol{V}_t^{-1} \boldsymbol{g}_t$
$\quad\quad \bar{\boldsymbol{x}}_{t+1} \leftarrow \alpha_t \boldsymbol{x}_{t+1} + (1 - \alpha_t) \bar{\boldsymbol{x}}_t$
$\quad$ **return** $\bar{\boldsymbol{x}}_T$

**Algorithm 3** NSD with momentum

**Hyperparam:** $\epsilon \geq 0$, total steps $T$, learning rate $\eta$, norm $\|\cdot\|$, averaging parameter $\alpha$

**Input:** initialization $\boldsymbol{x}_0$, initialization $\boldsymbol{m}_0$, stochastic loss functions $\{f_t\}_{t=1}^T$
$\quad$ **for** $t = 0, 1, \cdots, T - 1$ :
$\quad\quad \boldsymbol{g}_t \leftarrow \nabla f_t(\boldsymbol{x}_t)$
$\quad\quad \boldsymbol{m}_t \leftarrow (1 - \alpha) \boldsymbol{m}_{t-1} + \alpha \boldsymbol{g}_t$
$\quad\quad \boldsymbol{u}_t \leftarrow \arg\max_{\|\boldsymbol{u}\| \leq 1} \langle \boldsymbol{m}_t, \boldsymbol{u} \rangle$
$\quad\quad \boldsymbol{x}_{t+1} \leftarrow \boldsymbol{x}_t - \eta \boldsymbol{u}_t$
$\quad$ **return** $\boldsymbol{x}_T$

### 4.1 Adaptive Variance: An Analogue of Adaptive Smoothness

Our main results in this section are concerned about accelerated adaptive algorithms for convex functions and NSD in the nonconvex setting. Before presenting these results, we introduce a key technical ingredient: *adaptive variance*, a quantity that serves as the analogue of adaptive smoothness for gradient noise.

**Definition 4.1** (Standard and adaptive gradient variance). *For an index set $\mathcal{T}$, let $\{f_t\}_{t \in \mathcal{T}}$ be a set of stochastic loss functions where each $f_t : \mathbb{R}^d \to \mathbb{R}$.*

- *For any norm $\|\cdot\|$ on $\mathbb{R}^d$, the gradient variance of $\{f_t\}_{t \in \mathcal{T}}$ under $\|\cdot\|$ is defined as*

$$\sigma_{\|\cdot\|}(\{f_t\}_{t \in \mathcal{T}})^2 := \sup_{t \in \mathcal{T}, \boldsymbol{x} \in \mathbb{R}^d} \mathbb{E}\big[\big\|\nabla f_t(\boldsymbol{x}) - \mathbb{E}[\nabla f_t(\boldsymbol{x})]\big\|^2\big] \tag{6}$$

- *The adaptive gradient variance of $\{f_t\}_{t \in \mathcal{T}}$ w.r.t. any well-structured preconditioner set $\mathcal{H}$ is*

$$\sigma_{\mathcal{H}}(\{f_t\}_{t \in \mathcal{T}})^2 = \min_{\boldsymbol{H} \in \mathcal{H}, \operatorname{Tr}(\boldsymbol{H}) \leq 1} \sup_{t \in \mathcal{T}, \boldsymbol{x} \in \mathbb{R}^d} \mathbb{E}\big[\big\|\nabla f_t(\boldsymbol{x}) - \mathbb{E}[\nabla f_t(\boldsymbol{x})]\big\|_{\boldsymbol{H}^{-1}}^2\big]. \tag{7}$$

This adaptive variance is inspired by the noise assumption in Kovalev (2025a), both capturing the overall variation of gradient noise in the geometry induced by $\mathcal{H}$. Compared with the traditional noise assumption that only characterizes $\ell_2$ norm variance, adaptive variance provides a more informative measure. In addition, analogous to the comparison between $\Lambda_{\mathcal{H}}(f)$ and $L_{\|\cdot\|_{\mathcal{H}}}(f)$, the adaptive variance is always no smaller than $\|\cdot\|_{\mathcal{H},*}$-variance, as formalized in Proposition B.11.

Here we can compare Definition 4.1 with bounded covariance assumption in Xie et al. (2025b); An et al. (2025) that there exists $\boldsymbol{\Sigma} \succeq \boldsymbol{0}$ such that $\mathbb{E}[\nabla f(\boldsymbol{x}_t) - \nabla f_t(\boldsymbol{x}_t)][\nabla f(\boldsymbol{x}_t) - \nabla f_t(\boldsymbol{x}_t)]^\top \preceq \boldsymbol{\Sigma}$. When the covariance matrix is upper bounded by $\boldsymbol{\Sigma}$, $\sigma_{\mathcal{H}} \leq \operatorname{Tr}(P_{\mathcal{H}}(\boldsymbol{\Sigma}))$ for general $\mathcal{H}$ as shown in Proposition B.10. On the other hand, Definition 4.1 doesn't require the existence of $\boldsymbol{\Sigma}$ that can upper bound the covariance matrix everywhere. Therefore, Definition 4.1 is a weaker assumption than the bounded covariance assumption.

### 4.2 Acceleration on Convex Functions

We follow the framework in Kovalev (2025a) to formulate a unified class of adaptive optimizers with well-structured preconditioner sets with Nesterov acceleration in Algorithm 2. Through the perspective introduced by Kovalev & Borodich (2024), the idea is to interpret each step of Nesterov acceleration as a single step of standard gradient on a modified loss, i.e., $f^{\alpha_t, \bar{\boldsymbol{x}}_t}$ in Eq. 8. Here for a constant $\alpha \in (0, 1]$ and a reference point $\bar{\boldsymbol{x}}$, the corresponding modified loss is defined as

$$f^{\alpha, \bar{\boldsymbol{x}}}(\boldsymbol{x}) := \alpha^{-2} f(\alpha \boldsymbol{x} + (1 - \alpha) \bar{\boldsymbol{x}}). \tag{8}$$

For stochastic convex functions satisfying Assumption 4.2, we establish the convergence rate of Algorithm 2 in the following Theorem 4.3, whose proof is in Appendix E.

**Assumption 4.2.** *Let $f$ be a convex loss function. For all $t \in [T]$ and any $\boldsymbol{x}$, $\mathbb{E}[\nabla f_t(\boldsymbol{x})] = \nabla f(\boldsymbol{x})$.*

**Theorem 4.3.** *For a well-structured preconditioner set $\mathcal{H}$, let $f$ be a convex loss function whose $\mathcal{H}$-smoothness constant is $\Lambda_{\mathcal{H}}(f) \in (0, \infty)$ according to Definition 2.4. For $\epsilon > 0$, $T > 0$, consider Algorithm 2 with $\alpha_t = 2/(t + 2)$ for $t = 0, 1, \ldots, T - 1$. Suppose $\boldsymbol{x}^*$ is the global minima and $\max_{t=0,1,\ldots,T-1} \|\boldsymbol{x}_t - \boldsymbol{x}^*\|_{\mathcal{H}} \leq D$ for some $D > 0$ and Assumption 4.2 holds with adaptive gradient variance $\sigma_{\mathcal{H}}(\{f_t\}_{t=1}^T)^2 \leq \sigma_{\mathcal{H}}^2$ for some $\sigma_{\mathcal{H}} \in [0, \infty)$. Then it holds that*

$$\mathbb{E}[f(\bar{\boldsymbol{x}}_T) - f(\boldsymbol{x}^*)] \leq \frac{2D^2\epsilon}{\eta(T+1)^2}\mathbb{E}\operatorname{Tr}(\boldsymbol{V}_{T-1}^{-1}) + \left(\frac{D^2}{2\eta} - \frac{\eta}{2}\right)\mathbb{E}\frac{4}{(T+1)^2}\sum_{t=0}^{T-1}\boldsymbol{g}_t^\top\boldsymbol{V}_t^{-1}\boldsymbol{g}_t$$

$$+ \frac{2\eta^2}{(T+1)^2} \cdot \Lambda_{\mathcal{H}}(f) \cdot \tilde{O}(\log^2 d) + \frac{\eta}{T^{1/2}}\sigma_{\mathcal{H}} \cdot \tilde{O}(\log d).$$

*Moreover, when choosing learning rate $\eta = D$, the convergence rate becomes*

$$\mathbb{E}[f(\bar{\boldsymbol{x}}_T) - f(\boldsymbol{x}^*)] = \tilde{O}\left(\frac{\Lambda_{\mathcal{H}}(f) D^2 \log^2 d + d\sqrt{\epsilon}D}{T^2} + \frac{\sigma_{\mathcal{H}}D\log d}{\sqrt{T}}\right).$$

**Remark 4.4.** *A drawback of Algorithm 2 is that the optimal choice of learning rate $\eta$ in Theorem 4.3 depends on the unknown parameter $D = \max_t \|\boldsymbol{x}_t - \boldsymbol{x}^*\|_{\mathcal{H}}$. To circumvent this issue, we follow the approach in Kovalev (2025a) to introduce a projected variant (see Algorithm 8 and discussion in Appendix E.2), which ensures that all iterates remain inside a $\|\cdot\|_{\mathcal{H}}$-ball of radius $D$. The removes the requirement for prior knowledge of $D$, and we establish a same convergence rate in Theorem E.5.*

Our convergence guarantee attains a deterministic component of order $\tilde{O}(\Lambda_{\mathcal{H}}(f) D^2/T^2)$, an accelerated rate governed by the adaptive smoothness $\Lambda_{\mathcal{H}}(f)$. In comparison, Guzmán & Nemirovski (2015) shows that any first order optimizer can only achieve $\Omega(\frac{L_{\|\cdot\|_\infty}(f)}{T\log T})$ for the specific case of $\ell_\infty$ norm smoothness. Taken together, these results show that the adaptive smoothness is necessary to achieve the acceleration, which can't be replaced by the weaker non-Euclidean smoothness, highlighting its algorithmic benefit and answering Question 2 in the affirmative.

The analysis in Kovalev (2025a) yields similar results when $\mathcal{H}$ contains only diagonal matrices. However, their Assumption 4, which is critical for their analysis, imposes restrictive conditions on both the loss and the gradient noise for more general $\mathcal{H}$. By contrast, our approach avoids such requirements by leveraging Lemma 3.3.

## 4.3 NONCONVEX RESULTS FOR NSD UNDER ADAPTIVE NOISE ASSUMPTION

The ineffectiveness of averaging in the dual space also leads to difficulty in reducing gradient variance via averaging. To illustrate this, consider i.i.d. random vectors $\boldsymbol{x}_1, \ldots, \boldsymbol{x}_n$ with $\mathbb{E}[\boldsymbol{x}_i] = 0$ and $\mathbb{E}[\|\boldsymbol{x}_i\|_2^2] \leq \sigma^2$, we have $\mathbb{E}[\|\frac{1}{n}\sum_{i=1}^n\boldsymbol{x}_i\|_2^2] \leq \frac{\sigma^2}{n}$ while $\mathbb{E}[\|\frac{1}{n}\sum_{i=1}^n\boldsymbol{x}_i\|_1^2] \leq d\frac{\sigma^2}{n}$, and the extra $d$ factor in the latter bound can be tight. This causes the dimension-dependent convergence rate of NSD in the nonconvex setting, as shown in recent works (Pethick et al., 2025; Kovalev, 2025b).

In particular, the dimension-dependent factor $\rho = \sup_{\boldsymbol{x}} \frac{\|\boldsymbol{x}\|_{\mathcal{H},*}}{\|\boldsymbol{x}\|_2}$ captures the mismatch between $\|\cdot\|_{\mathcal{H},*}$ and $\|\cdot\|_2$. For diagonal $\mathcal{H}$, NSD reduces to SignGD with $\|\cdot\|_{\mathcal{H},*} = \|\cdot\|_1$, yielding $\rho = \Theta(\sqrt{d})$ and vacuous bounds when $T \ll d$. We avoid this by using the adaptive variance assumption. We prove the stochastic rate of NSD in Theorem 4.5 and the proof is deferred to Appendix F.1.

The concurrent work Kovalev & Borodich (2025) also leveraged the adaptive variance assumption to prove a dimension-free nonconvex convergence rate of NSD. However, they used a smoothness metric similar to adaptive smoothness while our Theorem 4.5 uses the standard smoothness. Our rate is strictly better because of the relationship between standard smoothness and adaptive smoothness.

**Theorem 4.5.** *Let $\mathcal{H}$ be a well-structured preconditioner set. For any $\epsilon \geq 0$, $\alpha \in (0, 1)$, $\eta > 0$, and $T \in \mathbb{N}$, let $\{\boldsymbol{x}_t\}_{t=0}^T$ be the iterates of Algorithm 3 with $\|\cdot\| = \|\cdot\|_{\mathcal{H}}$ and $\boldsymbol{m}_0 = \nabla f_0(\boldsymbol{x}_0)$. Let $L_{\|\cdot\|_{\mathcal{H}}}(f)$ be the smoothness of the loss $f$ w.r.t. $\|\cdot\|_{\mathcal{H}}$ according to Definition 2.3. Suppose Assumption 4.2 holds with adaptive gradient variance $\sigma_{\mathcal{H}}(\{f_t\}_{t=1}^T)^2 \leq \sigma_{\mathcal{H}}^2$. Then it holds that*

$$\mathbb{E}\frac{1}{T}\sum_{t=0}^{T-1}\|\nabla f(\boldsymbol{x}_t)\|_{\mathcal{H},*} \leq \frac{\Delta_0}{\eta T} + \frac{2\eta}{\alpha}L_{\|\cdot\|}(f) + \frac{2\sigma_{\mathcal{H}}}{\alpha T} + 2\sigma_{\mathcal{H}}\sqrt{\alpha}.$$

*Let $a_0 = \sqrt{\Delta_0 L_{\|\cdot\|}(f)}/\sigma_{\mathcal{H}}$. If $a_0 < 1$, then*

- *When $T < a_0^{-6}$, we choose $\alpha = T^{-2/3}$ and $\eta = \sqrt{\frac{\Delta_0}{L_{\|\cdot\|_{\mathcal{H}}}(f)}} T^{-5/12}$. The rate is $O\left(\sigma_{\mathcal{H}} T^{-1/3}\right)$.*

- *When $T \geq a_0^{-6}$, we choose $\alpha = \frac{a_0}{\sqrt{T}}$, $\eta = \frac{\Delta_0^{3/4} T^{-3/4}}{L_{\|\cdot\|_{\mathcal{H}}}(f)^{1/4} \sigma_{\mathcal{H}}^{1/2}}$. The rate is $O\left(\frac{(\Delta_0 L_{\|\cdot\|_{\mathcal{H}}}(f))^{1/4} \sqrt{\sigma_{\mathcal{H}}}}{T^{1/4}}\right)$.*

  *If $a_0 \geq 1$, then*

- *When $T \leq a_0^2$, we choose $\alpha = 1$ and $\eta = \sqrt{\frac{\Delta_0}{L_{\|\cdot\|_{\mathcal{H}}}(f)}} T^{-1/2}$. The rate is $O\left(\sqrt{\Delta_0 L_{\|\cdot\|_{\mathcal{H}}}(f)} T^{-1/2}\right)$.*

- *When $T \geq a_0^2$, we choose $\alpha = \frac{a_0}{\sqrt{T}}$ and $\eta = \frac{\Delta_0^{3/4} T^{-3/4}}{L_{\|\cdot\|_{\mathcal{H}}}(f)^{1/4} \sigma_{\mathcal{H}}^{1/2}}$. The rate is $O\left(\frac{(\Delta_0 L_{\|\cdot\|_{\mathcal{H}}}(f))^{1/4} \sqrt{\sigma_{\mathcal{H}}}}{T^{1/4}}\right)$.*

Theorem 4.5 shows that, with appropriate choices of the learning rate $\eta$ and averaging parameter $\alpha$, NSD achieves a dimension-free rate depending only on the standard smoothness $L_{\|\cdot\|_{\mathcal{H}}}(f)$ and the adaptive variance $\sigma_{\mathcal{H}}$, thereby avoiding the unfavorable $\rho$ factor. Next Theorem 4.6 shows that such a dimension-free upper bound is unattainable under the standard gradient variance assumption.

**Theorem 4.6.** *For any $\epsilon \geq 0$, $\alpha \in (0,1)$, $\eta > 0$, and $T \in \mathbb{N}$, let $\{x_t\}_{t=0}^T$ be the iterates of Algorithm 3 with any norm $\|\cdot\|$. Suppose Assumption 4.2 holds with the gradient variance $\sigma_{\|\cdot\|_*}(\{f_t\}_{t=1}^T)^2 \leq \sigma_{\|\cdot\|_*}^2$ for some $\sigma_{\|\cdot\|_*} \in [0,\infty)$. Then it holds that*

$$\mathbb{E} \frac{1}{T} \sum_{t=0}^{T-1} \|\nabla f(x_t)\|_* \leq \frac{\Delta_0}{\eta T} + \frac{2\eta}{\alpha} L_{\|\cdot\|}(f) + \frac{2}{\alpha T} \sigma_{\|\cdot\|_*} + 2\sigma_{\|\cdot\|_*} \cdot \min\left(1, \alpha^{1/2} \psi(\|\cdot\|_*, \|\cdot\|_2)\right)$$

*where $\psi(\|\cdot\|_*, \|\cdot\|_2) = \sup_x \frac{\|x\|_*}{\|x\|_2} \cdot \sup_x \frac{\|x\|_2}{\|x\|_*}$ measures the distortion between the two norms.*

Here the norm distortion $\psi(\|\cdot\|_*, \|\cdot\|_2)$ can grow with the dimension $d$. Consequently, Theorem 4.6 gives two kinds of upper bound for the convergence rate of NSD:

- When $T$ is small (e.g. $T < d$), the constant term $\sigma_{\|\cdot\|_*}$ dominates.
- For sufficiently large $T$ and small $\alpha$ (for which the four terms are balanced), the last term in the upper bound becomes $2\sigma_{\|\cdot\|_*} \alpha^{1/2} \psi(\|\cdot\|_*, \|\cdot\|_2)$, which depends on $d$.

Moreover, such a dependence on $d$ is inevitable in the worst case as shown by Theorem 4.7.

**Theorem 4.7.** *For any fixed $\Delta_0, L, \sigma^2, d, T$, learning rate $\eta$, and any averaging parameter $\alpha$, there exists a loss function $f : \mathbb{R}^d \to \mathbb{R}$, a sequence of stochastic iid loss functions $f_0, f_1, \cdots, f_{T-1}$ and an initialization $x_0$ satisfying the following conditions:(1) $f(x_0) - \inf_x f(x) = \Delta_0$ and $L_{\|\cdot\|_\infty}(f) \leq L$; (2) For any $x \in \mathbb{R}^d$, it holds that $\mathbb{E}[\nabla f_t(x)] = \nabla f(x)$ and $\mathbb{E}[\|\nabla f_t(x) - \nabla f(x)\|_1^2] \leq \sigma^2$.*

*When running Algorithm 3 with $\|\cdot\| = \|\cdot\|_\infty$, learning rate $\eta$, averaging parameter $\alpha$ and initialization $x_0 = 0$, it holds that*

$$\mathbb{E}\left[\min_{t \in [T]} \|\nabla f(x_t)\|_1\right] = \min\{e^{-2} 5^{-\frac{1}{4}} (dL\Delta_0 \sigma^2)^{\frac{1}{4}} T^{-\frac{1}{2}}, e^{-2} 5^{-\frac{1}{2}} \sigma\}$$

Theorem 4.7 also shows two kinds of lower bound we can achieve on signGD with momentum:

- When $T$ is not large enough, we can achieve the lower bound $\Omega(\sigma)$, which shows the hardness induced by the stochasticity and matches the first upper bound in Theorem 4.6.
- On the other hand, if we want to achieve the error $\epsilon < e^{-2} 5^{-\frac{1}{2}} \sigma$, we require the number of steps $T = \Omega(\epsilon^{-2} (dL\Delta_0 \sigma^2)^{\frac{1}{2}})$, whose dependence on dimension $d$ is $\Omega(d^{\frac{1}{2}})$.

In conclusion, under the standard gradient variance assumption with $\|\cdot\| = \|\cdot\|_\infty$ and $\|\cdot\|_* = \|\cdot\|_1$, the $d$-dependent rate in Theorem 4.6 is unavoidable. In contrast, Theorem 4.5 attains a dimension-free rate under the adaptive gradient variance assumption, highlighting a fundamental gap.

## 5 CONCLUSION

We extend the unified analysis of adaptive optimizers to nonconvex functions, establishing convergence rate depending on the adaptive smoothness. It strengthens the comparison between smoothnesses that adaptive optimizers and NSD use in the convex settings. We further show the benefit of adaptive smoothness by showing the accelerated rate of adaptive optimizers with Nesterov momentum. The benefit of adaptive geometry is also justified by comparing two kinds of noise.

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

CONTENTS

## A    RELATED WORK

**Adaptive optimizers.**    Matrix-valued preconditioning methods such as Shampoo (Gupta et al., 2018) provide structure-aware updates and a growing body of work refines their preconditioners and practice (e.g., improved analyses/implementations and Adam–Shampoo hybrids) (Morwani et al., 2024; Vyas et al., 2024; Lau et al., 2025; Si et al., 2025). Variants like one-sided Shampoo/ASGO achieve stronger convex rates than earlier methods (Xie et al., 2025b; An et al., 2025), but the nonconvex guarantees are missing which we show in Section 3. The recent work Frans et al. (2025) views adaptive optimizers as a modified matrix whitening technique and points out it can outperform accurate spectral normalization (NSD in our context). They highlight the variance adaptation is previously overlooked while we also want to understand the benefit of adaptive optimizers/geometry.

**NSD.**    Cutkosky & Mehta (2020) proves the convergence rate $O(1/T^{3.5})$ of normalized gradient descent with momentum on nonconvex functions. Pethick et al. (2025); Kovalev (2025b) extend the results to any normalized steepest descent while the smoothness metric becomes smoothness w.r.t. the specific norm NSD uses. Many works analyze popular optimizers by showing the equivalence to NSD under some norm, e.g. Lion is NSD under $\ell_\infty$-norm (Sfyraki & Wang, 2025) and Muon is NSD under spectral norm (Chen et al., 2025), and thus apply the classic optimization results. Jiang et al. (2025) achieves a similar result as our Theorem 4.5 but their noise assumption is stronger than ours and only states the result for specific SignGD. The difference between the two kinds of smoothness is also discussed in Balles et al. (2020) in the context for SignGD only.

**Acceleration.**    Nesterov acceleration can be viewed as the linear coupling between gradient descent and mirror descent (Kelner et al., 2014; Allen-Zhu & Orecchia, 2014; Cheng et al., 2018). Through this lens, Cutkosky (2019); Kavis et al. (2019); Joulani et al. (2020); Ene et al. (2021) applied it on adaptive optimizers with specific structures like diagonal/coordinate-wise and achieve deterministic $O(1/T^2)$ convergence rate.

## B    AUXILIARY RESULTS

Recall that we define $P_{\mathcal{H}}(\boldsymbol{M}) = \arg\min_{\boldsymbol{H} \in \mathcal{H}} \langle \boldsymbol{M}, \boldsymbol{H}^{-1} \rangle + \mathrm{Tr}(\boldsymbol{H})$ for any $\boldsymbol{M} \in \mathcal{S}_{++}^d$ in Section 2. In this section, we will prove several important properties of $P_{\mathcal{H}}$ which will be useful in proving the convergence rate for general well-structured $\mathcal{H}$.

It will help interpret these results when keeping in mind that $P_{\mathcal{H}}(\boldsymbol{M}) = \boldsymbol{M}^{1/2}$ when $\mathcal{H}$ is the PSD matrix cone, i.e., there is no constraint on the structure.

We will first state Lemma A.2 in Xie et al. (2025b), which will be used a lot in our proof.

**Lemma B.1** (Lemma A.2 in Xie et al. (2025b)).    *For any $\boldsymbol{A}, \boldsymbol{B} \in \mathcal{H}$, if $\langle \boldsymbol{A}, \boldsymbol{H} \rangle = \langle \boldsymbol{B}, \boldsymbol{H} \rangle$ holds for all $\boldsymbol{H} \in \mathcal{H}$, then it must be $\boldsymbol{A} = \boldsymbol{B}$.*

**Lemma B.2.**    *For any $\boldsymbol{M} \in \mathcal{S}_{++}^d$ and any $\boldsymbol{H} \in \mathcal{H}$, $\lambda \geq 0$, it always holds that $\langle \boldsymbol{M}, \boldsymbol{H} \rangle = \langle P_{\mathcal{H}}(\boldsymbol{M})^2, \boldsymbol{H} \rangle$ and $P_{\mathcal{H}}(\boldsymbol{M} + \lambda \boldsymbol{I}_d)^2 = P_{\mathcal{H}}(\boldsymbol{M})^2 + \lambda \boldsymbol{I}_d$.*

*Proof.*    By property (c) in Proposition A.3 in Xie et al. (2025b), we know that for any $\boldsymbol{M} \succ 0$,
$$\langle \boldsymbol{M} - P_{\mathcal{H}}(\boldsymbol{M})^2, P_{\mathcal{H}}(\boldsymbol{M})^{-1} \boldsymbol{H} P_{\mathcal{H}}(\boldsymbol{M})^{-1} - P_{\mathcal{H}}(\boldsymbol{M})^{-1} \rangle = 0.$$
Note that $\boldsymbol{H} \mapsto P_{\mathcal{H}}(\boldsymbol{M})^{-1} \boldsymbol{H} P_{\mathcal{H}}(\boldsymbol{M})^{-1}$ is a bijection from $\mathcal{H}$ to $\mathcal{H}$ by lemma A.1 in Xie et al. (2025b), so we have $\langle \boldsymbol{M} - P_{\mathcal{H}}(\boldsymbol{M})^2, \boldsymbol{H} - P_{\mathcal{H}}(\boldsymbol{M})^{-1} \rangle = 0$ for all $\boldsymbol{H} \in \mathcal{H}$. $\boldsymbol{H} \mapsto \boldsymbol{H} - P_{\mathcal{H}}(\boldsymbol{M})^{-1}$ is also a bijection from $\mathcal{H}$ to $\mathcal{H}$. So we have that $\langle \boldsymbol{M} - P_{\mathcal{H}}(\boldsymbol{M})^2, \boldsymbol{H} \rangle = 0$ for all $\boldsymbol{H} \in \mathcal{H}$.

For any $\lambda \geq 0$ and any $\boldsymbol{H} \in \mathcal{H}$, we have that
$$\begin{aligned}
\langle P_{\mathcal{H}}(\boldsymbol{M} + \lambda \boldsymbol{I}_d)^2, \boldsymbol{H} \rangle &= \langle \boldsymbol{M} + \lambda \boldsymbol{I}_d, \boldsymbol{H} \rangle \\
&= \langle \boldsymbol{M}, \boldsymbol{H} \rangle + \langle \lambda \boldsymbol{I}_d, \boldsymbol{H} \rangle \\
&= \langle P_{\mathcal{H}}(\boldsymbol{M})^2, \boldsymbol{H} \rangle + \langle \lambda \boldsymbol{I}_d, \boldsymbol{H} \rangle \\
&= \langle P_{\mathcal{H}}(\boldsymbol{M})^2 + \lambda \boldsymbol{I}_d, \boldsymbol{H} \rangle.
\end{aligned}$$
Note that $P_{\mathcal{H}}(\boldsymbol{M} + \lambda \boldsymbol{I}_d)^2$ and $P_{\mathcal{H}}(\boldsymbol{M})^2 + \lambda \boldsymbol{I}_d$ are both in $\mathcal{H}$. We conclude $P_{\mathcal{H}}(\boldsymbol{M} + \lambda \boldsymbol{I}_d)^2 = P_{\mathcal{H}}(\boldsymbol{M})^2 + \lambda \boldsymbol{I}_d$ according to Lemma B.1. $\square$

**Lemma B.3.** *For any $M_1, M_2 \in \mathcal{S}_{++}^d$, $P_{\mathcal{H}}(M_1 + M_2)^2 = P_{\mathcal{H}}(M_1)^2 + P_{\mathcal{H}}(M_2)^2$. For any $c \geq 0$, $P_{\mathcal{H}}(cM)^2 = cP_{\mathcal{H}}(M)^2$. For any $0 \preceq A \preceq B$, it always holds $P_{\mathcal{H}}(A) \preceq P_{\mathcal{H}}(B)$.*

*Proof.* For any $H \in \mathcal{H}$, we have that

$$
\begin{aligned}
\langle P_{\mathcal{H}}(M_1 + M_2)^2, H \rangle &= \langle M_1 + M_2, H \rangle \\
&= \langle M_1, H \rangle + \langle M_2, H \rangle \\
&= \langle P_{\mathcal{H}}(M_1)^2, H \rangle + \langle P_{\mathcal{H}}(M_2)^2, H \rangle \\
&= \langle P_{\mathcal{H}}(M_1)^2 + P_{\mathcal{H}}(M_2)^2, H \rangle.
\end{aligned}
$$

Then we have that $P_{\mathcal{H}}(M_1 + M_2)^2 = P_{\mathcal{H}}(M_1)^2 + P_{\mathcal{H}}(M_2)^2$ from Lemma B.1. For any $c \geq 0$, it holds that

$$
\begin{aligned}
\langle P_{\mathcal{H}}(cM)^2, H \rangle &= \langle cM, H \rangle \\
&= c \langle M, H \rangle \\
&= c \langle P_{\mathcal{H}}(M)^2, H \rangle \\
&= \langle cP_{\mathcal{H}}(M)^2, H \rangle.
\end{aligned}
$$

So it also holds that $P_{\mathcal{H}}(cM)^2 = cP_{\mathcal{H}}(M)^2$ from Lemma B.1.

Moreover, when $0 \preceq A \preceq B$, $P_{\mathcal{H}}(B)^2 = P_{\mathcal{H}}(A)^2 + P_{\mathcal{H}}(B - A)^2 \succeq P_{\mathcal{H}}(A)^2$. Then $P_{\mathcal{H}}(B) \succeq P_{\mathcal{H}}(A)$ because $X^{1/2}$ is a monotone function. $\square$

**Lemma B.4.** *If we define $\mathrm{proj}_{\mathcal{H}}(M) = \arg\min_{H \in \mathcal{H}} \|M - H\|_{\mathrm{F}}^2$, then it holds that $\mathrm{proj}_{\mathcal{H}}(M) = P_{\mathcal{H}}(M)^2$.*

*Proof.* We define $d_M(H) = \|M - H\|_{\mathrm{F}}^2 = \mathrm{Tr}((M - H)(M - H)^{\top})$. Then $\nabla_H d_M(H) = 2(H - M)$. Because $\mathcal{H}$ is a cone, by the optimality of $\mathrm{proj}_{\mathcal{H}}(M)$, it always holds for any $H \in \mathcal{H}$ that

$$
0 = \langle \nabla_H d_M(\mathrm{proj}_{\mathcal{H}}(M)), H - \mathrm{proj}_{\mathcal{H}}(M) \rangle = 2 \langle \mathrm{proj}_{\mathcal{H}}(M) - M, H - \mathrm{proj}_{\mathcal{H}}(M) \rangle.
$$

For any $H' \in \mathcal{H}$, we can always choose $H = H' + \mathrm{proj}_{\mathcal{H}}(M)$. $H \in \mathcal{H}$ because $\mathcal{H}$ is closed under matrix addition. Then it always holds that $\langle \mathrm{proj}_{\mathcal{H}}(M) - M, H' \rangle = 0$. On the other hand, Lemma B.2 shows it always holds that $\langle P_{\mathcal{H}}(M)^2 - M, H' \rangle = 0$ for any $H' \in \mathcal{H}$. Then we can get $\mathrm{proj}_{\mathcal{H}}(M) = P_{\mathcal{H}}(M)^2$ by Lemma B.1. $\square$

**Lemma B.5.** *For any $A, B \in \mathcal{S}_{++}^d$ and any well-structured preconditioner set $\mathcal{H}$,*

$$
\|P_{\mathcal{H}}(B) - P_{\mathcal{H}}(A)\|_{\mathrm{op}} \leq \|B - A\|_{\mathrm{op}}^{\frac{1}{2}}
$$

*Proof.* We have that

$$
\begin{aligned}
P_{\mathcal{H}}(B) = P_{\mathcal{H}}(A + B - A) &\preceq P_{\mathcal{H}}(A + \|B - A\|_{\mathrm{op}} I) \\
&= \left( P_{\mathcal{H}}(A)^2 + \|B - A\|_{\mathrm{op}} I \right)^{\frac{1}{2}} \preceq P_{\mathcal{H}}(A) + \|B - A\|_{\mathrm{op}}^{\frac{1}{2}} I
\end{aligned}
$$

The last step is by considering $\sqrt{\lambda_i^2 + \|B - A\|_{\mathrm{op}}} \leq \lambda_i + \sqrt{\|B - A\|_{\mathrm{op}}}$ for every eigenvalue $\lambda_i$ of $P_{\mathcal{H}}(A)$. Similarly we can also show $P_{\mathcal{H}}(A) \preceq P_{\mathcal{H}}(B) + \|B - A\|_{\mathrm{op}}^{\frac{1}{2}} I$, which finishes the proof. $\square$

**Lemma B.6.** $\mathrm{Tr}[P_{\mathcal{H}}(X)]$ *is a concave function.*

*Proof.* For any psd matrices $\boldsymbol{A}$ and $\boldsymbol{B}$, we have that

$$
\begin{aligned}
2\operatorname{Tr}\left[P_{\mathcal{H}}(\frac{\boldsymbol{A}+\boldsymbol{B}}{2})\right] &= \min_{\boldsymbol{H}\in\mathcal{H}}\left[\left\langle\frac{\boldsymbol{A}+\boldsymbol{B}}{2},\boldsymbol{H}^{-1}\right\rangle + \operatorname{Tr}(\boldsymbol{H})\right] \\
&= \min_{\boldsymbol{H}\in\mathcal{H}}\left[\frac{\langle\boldsymbol{A},\boldsymbol{H}^{-1}\rangle + \operatorname{Tr}(\boldsymbol{H})}{2} + \frac{\langle\boldsymbol{B},\boldsymbol{H}^{-1}\rangle + \operatorname{Tr}(\boldsymbol{H})}{2}\right] \\
&\geq \min_{\boldsymbol{H}\in\mathcal{H}}\frac{\langle\boldsymbol{A},\boldsymbol{H}^{-1}\rangle + \operatorname{Tr}(\boldsymbol{H})}{2} + \min_{\boldsymbol{H}\in\mathcal{H}}\frac{\langle\boldsymbol{B},\boldsymbol{H}^{-1}\rangle + \operatorname{Tr}(\boldsymbol{H})}{2} \\
&= \operatorname{Tr}[P_{\mathcal{H}}(\boldsymbol{A})] + \operatorname{Tr}[P_{\mathcal{H}}(\boldsymbol{B})].
\end{aligned}
$$

$\square$

**Lemma B.7.** *Let $\boldsymbol{A}\succeq 0$ and $\boldsymbol{x},\boldsymbol{y}\in\mathbb{R}^d$. It holds*
$$
\left\|P_{\mathcal{H}}(\boldsymbol{A}+\boldsymbol{x}\boldsymbol{x}^\top) - P_{\mathcal{H}}(\boldsymbol{A}+\boldsymbol{y}\boldsymbol{y}^\top)\right\|_{\mathrm{op}} \leq \sqrt{2}\,\|\boldsymbol{x}-\boldsymbol{y}\|_2.
$$

*Proof.* Set $F(\boldsymbol{X},\boldsymbol{Y}) := \operatorname{tr}\left(\sqrt{\boldsymbol{X}}\,\sqrt{\boldsymbol{Y}}\right)$ for $\boldsymbol{X},\boldsymbol{Y}\succeq 0$. By Lieb's concavity theorem Lieb (1973), $F$ is jointly concave and positively homogeneous, hence superadditive:
$$
F(\boldsymbol{X}_1+\boldsymbol{X}_2,\boldsymbol{Y}_1+\boldsymbol{Y}_2) \geq F(\boldsymbol{X}_1,\boldsymbol{Y}_1) + F(\boldsymbol{X}_2,\boldsymbol{Y}_2).
$$

Define $\boldsymbol{\Pi}_{\mathcal{H}}(\boldsymbol{H}) = P_{\mathcal{H}}(\boldsymbol{H})^2$. Then $\boldsymbol{\Pi}_{\mathcal{H}}$ is the pinching onto a unital $*$-subalgebra $\mathcal{H}$.

For any $\boldsymbol{A},\boldsymbol{P},\boldsymbol{Q}\succeq 0$,
$$
\|P_{\mathcal{H}}(\boldsymbol{A}+\boldsymbol{P}) - P_{\mathcal{H}}(\boldsymbol{A}+\boldsymbol{Q})\|_{\mathrm{F}}^2 = \operatorname{tr}(\boldsymbol{A}+\boldsymbol{P}) + \operatorname{tr}(\boldsymbol{A}+\boldsymbol{Q}) - 2\operatorname{tr}\left(P_{\mathcal{H}}(\boldsymbol{A}+\boldsymbol{P})P_{\mathcal{H}}(\boldsymbol{A}+\boldsymbol{Q})\right).
$$

Using $P_{\mathcal{H}}(\boldsymbol{Z})^2 = \boldsymbol{\Pi}_{\mathcal{H}}(\boldsymbol{Z})$, the cross term equals $F(\boldsymbol{\Pi}_{\mathcal{H}}(\boldsymbol{A}) + \boldsymbol{\Pi}_{\mathcal{H}}(\boldsymbol{P}), \boldsymbol{\Pi}_{\mathcal{H}}(\boldsymbol{A}) + \boldsymbol{\Pi}_{\mathcal{H}}(\boldsymbol{Q}))$, which by superadditivity is at least $F(\boldsymbol{\Pi}_{\mathcal{H}}(\boldsymbol{A}),\boldsymbol{\Pi}_{\mathcal{H}}(\boldsymbol{A})) + F(\boldsymbol{\Pi}_{\mathcal{H}}(\boldsymbol{P}),\boldsymbol{\Pi}_{\mathcal{H}}(\boldsymbol{Q})) = \operatorname{tr}\boldsymbol{A} + \operatorname{tr}\left(P_{\mathcal{H}}(\boldsymbol{P})P_{\mathcal{H}}(\boldsymbol{Q})\right)$. Hence
$$
\|P_{\mathcal{H}}(\boldsymbol{A}+\boldsymbol{P}) - P_{\mathcal{H}}(\boldsymbol{A}+\boldsymbol{Q})\|_{\mathrm{F}} \leq \|P_{\mathcal{H}}(\boldsymbol{P}) - P_{\mathcal{H}}(\boldsymbol{Q})\|_{\mathrm{F}}. \tag{9}
$$

Write $r = \|\boldsymbol{x}\|_2$, $s = \|\boldsymbol{y}\|_2$, $\boldsymbol{u} = \boldsymbol{x}/r$, $\boldsymbol{v} = \boldsymbol{y}/s$, and $\cos\theta = \boldsymbol{u}^\top\boldsymbol{v}$. For the unstructured square root,
$$
\left\|\sqrt{\boldsymbol{x}\boldsymbol{x}^\top} - \sqrt{\boldsymbol{y}\boldsymbol{y}^\top}\right\|_{\mathrm{F}}^2 = r^2 + s^2 - 2rs\cos^2\theta \leq 2\left(r^2 + s^2 - 2rs\cos\theta\right) = 2\|\boldsymbol{x}-\boldsymbol{y}\|_2^2.
$$
Pinching monotonicity of $F$ yields
$$
\|P_{\mathcal{H}}(\boldsymbol{x}\boldsymbol{x}^\top) - P_{\mathcal{H}}(\boldsymbol{y}\boldsymbol{y}^\top)\|_{\mathrm{F}}^2 = \|\sqrt{\boldsymbol{\Pi}_{\mathcal{H}}(\boldsymbol{x}\boldsymbol{x}^\top)} - \sqrt{\boldsymbol{\Pi}_{\mathcal{H}}(\boldsymbol{y}\boldsymbol{y}^\top)}\|_{\mathrm{F}}^2 \leq \|\sqrt{\boldsymbol{x}\boldsymbol{x}^\top} - \sqrt{\boldsymbol{y}\boldsymbol{y}^\top}\|_{\mathrm{F}}^2 \leq 2\|\boldsymbol{x}-\boldsymbol{y}\|_2^2.
$$

Apply Eq. 9 with $\boldsymbol{P} = \boldsymbol{x}\boldsymbol{x}^\top$, $\boldsymbol{Q} = \boldsymbol{y}\boldsymbol{y}^\top$,
$$
\|P_{\mathcal{H}}(\boldsymbol{A}+\boldsymbol{x}\boldsymbol{x}^\top) - P_{\mathcal{H}}(\boldsymbol{A}+\boldsymbol{y}\boldsymbol{y}^\top)\|_{\mathrm{op}} \leq \|P_{\mathcal{H}}(\boldsymbol{x}\boldsymbol{x}^\top) - P_{\mathcal{H}}(\boldsymbol{y}\boldsymbol{y}^\top)\|_{\mathrm{F}} \leq \sqrt{2}\,\|\boldsymbol{x}-\boldsymbol{y}\|_2.
$$

$\square$

**Lemma B.8.** *For any well-structured preconditioner set $\mathcal{H}$ and any vector $\boldsymbol{x}\in\mathbb{R}^d$, it holds that*
$$
\|\boldsymbol{x}\|_{\mathcal{H},*} = \operatorname{Tr}[P_{\mathcal{H}}(\boldsymbol{x}\boldsymbol{x}^\top)].
$$

*Proof.* We have that
$$
\begin{aligned}
\operatorname{Tr}[P_{\mathcal{H}}(\boldsymbol{x}\boldsymbol{x}^\top)] &= \frac{1}{2}\min_{\boldsymbol{H}\in\mathcal{H}}\boldsymbol{x}^\top\boldsymbol{H}^{-1}\boldsymbol{x} + \operatorname{Tr}(\boldsymbol{H}) \\
&= \frac{1}{2}\min_{\boldsymbol{H}\in\mathcal{H},\operatorname{Tr}(\boldsymbol{H})\leq 1}\min_{c>0}c^{-1}\boldsymbol{x}^\top\boldsymbol{H}^{-1}\boldsymbol{x} + c\operatorname{Tr}(\boldsymbol{H}) \\
&= \min_{\boldsymbol{H}\in\mathcal{H},\operatorname{Tr}(\boldsymbol{H})\leq 1}\sqrt{\boldsymbol{x}^\top\boldsymbol{H}^{-1}\boldsymbol{x}}.
\end{aligned}
$$

Xie et al. (2025b) shows that $\|\boldsymbol{x}\|_{\mathcal{H},*} = \min_{\boldsymbol{H}\in\mathcal{H},\operatorname{Tr}(\boldsymbol{H})\leq 1}\sqrt{\boldsymbol{x}^\top\boldsymbol{H}^{-1}\boldsymbol{x}}$ in their proof of lemma 3.3, which finishes the proof.

$\square$

**Lemma B.9.**

$$P_{\mathcal{H}}(\boldsymbol{g}_t\boldsymbol{g}_t^\top)^{-1}\boldsymbol{g}_t = \arg\min_{\|\boldsymbol{x}\|_{\mathcal{H}}\leq 1}\langle\boldsymbol{x},\boldsymbol{g}_t\rangle$$

*Proof.* First we show that it satisfies that $\left\|P_{\mathcal{H}}(\boldsymbol{g}_t\boldsymbol{g}_t^\top)^{-1}\boldsymbol{g}_t\right\|_{\mathcal{H}} \leq 1$. For any $\boldsymbol{H} \in \mathcal{H}$ and $\mathrm{Tr}(\boldsymbol{H}) \leq 1$, we have that

$$
\begin{aligned}
\left\|P_{\mathcal{H}}(\boldsymbol{g}_t\boldsymbol{g}_t^\top)^{-1}\boldsymbol{g}_t\right\|_{\boldsymbol{H}}^2 &= \boldsymbol{g}_t^\top P_{\mathcal{H}}(\boldsymbol{g}_t\boldsymbol{g}_t^\top)^{-1}\boldsymbol{H}P_{\mathcal{H}}(\boldsymbol{g}_t\boldsymbol{g}_t^\top)^{-1}\boldsymbol{g}_t \\
&= \mathrm{Tr}(P_{\mathcal{H}}(\boldsymbol{g}_t\boldsymbol{g}_t^\top)^{-1}\boldsymbol{H}P_{\mathcal{H}}(\boldsymbol{g}_t\boldsymbol{g}_t^\top)^{-1}\boldsymbol{g}_t\boldsymbol{g}_t^\top) \\
&= \mathrm{Tr}(P_{\mathcal{H}}(\boldsymbol{g}_t\boldsymbol{g}_t^\top)^{-1}\boldsymbol{H}P_{\mathcal{H}}(\boldsymbol{g}_t\boldsymbol{g}_t^\top)^{-1}P_{\mathcal{H}}(\boldsymbol{g}_t\boldsymbol{g}_t^\top)^2) \\
&= \mathrm{Tr}(\boldsymbol{H}) \leq 1
\end{aligned}
$$

Then $\left\|P_{\mathcal{H}}(\boldsymbol{g}_t\boldsymbol{g}_t^\top)^{-1}\boldsymbol{g}_t\right\|_{\mathcal{H}} = \sup_{\boldsymbol{H}\in\mathcal{H},\mathrm{Tr}(\boldsymbol{H})\leq 1}\left\|P_{\mathcal{H}}(\boldsymbol{g}_t\boldsymbol{g}_t^\top)^{-1}\boldsymbol{g}_t\right\|_{\boldsymbol{H}} \leq 1$.

Next, we can employ Lemma B.8 to show that

$$\left\langle P_{\mathcal{H}}(\boldsymbol{g}_t\boldsymbol{g}_t^\top)^{-1}\boldsymbol{g}_t, \boldsymbol{g}_t\right\rangle = \mathrm{Tr}(P_{\mathcal{H}}(\boldsymbol{g}_t\boldsymbol{g}_t^\top)^{-1}\boldsymbol{g}_t\boldsymbol{g}_t^\top) = \mathrm{Tr}(P_{\mathcal{H}}(\boldsymbol{g}_t\boldsymbol{g}_t^\top)) = \|\boldsymbol{g}_t\|_{\mathcal{H},*}.$$

Because $\min_{\|\boldsymbol{x}\|_{\mathcal{H}}\leq 1}\langle\boldsymbol{x},\boldsymbol{g}_t\rangle = \|\boldsymbol{g}_t\|_{\mathcal{H},*}$ and there is only one unique vector achieving this optimality, we finish showing the statement. $\qquad\square$

**Proposition B.10.** *For any well-structured preconditioner set $\mathcal{H}$, let $\sigma_{\mathcal{H}}(\{f_t\}_{t\in\mathcal{T}})$ be adaptive gradient variance defined in Definition 4.1. It always hold $\sigma_{\mathcal{H}}(\{f_t\}_{t\in\mathcal{T}}) \leq \mathrm{Tr}(P_{\mathcal{H}}(\boldsymbol{\Sigma}))$ when we assume $\mathbb{E}[\nabla f(\boldsymbol{x}) - \nabla f_t(\boldsymbol{x})][\nabla f(\boldsymbol{x}) - \nabla f_t(\boldsymbol{x})]^\top \preceq \boldsymbol{\Sigma}$ for any $t \in \mathcal{T}$.*

*Proof of Proposition B.10.* From Definition 4.1, we can have that

$$
\begin{aligned}
\sigma_{\mathcal{H}}(\{f_t\}_{t\in\mathcal{T}})^2 &= \min_{\boldsymbol{H}\in\mathcal{H},\mathrm{Tr}(\boldsymbol{H})\leq 1}\sup_{\boldsymbol{x},t}\mathbb{E}\|\nabla f(\boldsymbol{x}) - \nabla f_t(\boldsymbol{x}\|_{\boldsymbol{H}^{-1}}^2 \\
&= \min_{\boldsymbol{H}\in\mathcal{H},\mathrm{Tr}(\boldsymbol{H})\leq 1}\sup_{\boldsymbol{x},t}\mathbb{E}\left\langle[\nabla f(\boldsymbol{x}) - \nabla f_t(\boldsymbol{x})][\nabla f(\boldsymbol{x}) - \nabla f_t(\boldsymbol{x})]^\top, \boldsymbol{H}^{-1}\right\rangle \\
&\leq \min_{\boldsymbol{H}\in\mathcal{H},\mathrm{Tr}(\boldsymbol{H})\leq 1}\left\langle\boldsymbol{\Sigma}, \boldsymbol{H}^{-1}\right\rangle.
\end{aligned}
$$

By the definition of $P_{\mathcal{H}}(\boldsymbol{\Sigma})$, we know that $\left\langle\boldsymbol{\Sigma}, P_{\mathcal{H}}(\boldsymbol{\Sigma})^{-1}\right\rangle = \mathrm{Tr}(P_{\mathcal{H}}(\boldsymbol{\Sigma}))$ and

$$
\begin{aligned}
\mathrm{Tr}(P_{\mathcal{H}}(\boldsymbol{\Sigma})) &= \frac{1}{2}\min_{\boldsymbol{H}\in\mathcal{H}}\left\langle\boldsymbol{\Sigma}, \boldsymbol{H}^{-1}\right\rangle + \mathrm{Tr}(\boldsymbol{H}) \\
&\geq \min_{\boldsymbol{H}\in\mathcal{H}}\sqrt{\left\langle\boldsymbol{\Sigma}, \boldsymbol{H}^{-1}\right\rangle\mathrm{Tr}(\boldsymbol{H})} \\
&= \min_{\boldsymbol{H}\in\mathcal{H}}\sqrt{\left\langle\boldsymbol{\Sigma}, (\boldsymbol{H}/\mathrm{Tr}(\boldsymbol{H}))^{-1}\right\rangle} \\
&= \min_{\boldsymbol{H}\in\mathcal{H},\mathrm{Tr}(\boldsymbol{H})\leq 1}\sqrt{\left\langle\boldsymbol{\Sigma}, \boldsymbol{H}^{-1}\right\rangle} \\
&\geq \sigma_{\mathcal{H}}(\{f_t\}_{t\in\mathcal{T}}),
\end{aligned}
$$

which finishes the proof. $\qquad\square$

### B.1 COMPARISON ON ADAPTIVE GEOMETRY

**Proposition 2.5.** *For any well-structured preconditioner set $\mathcal{H} \subseteq \mathcal{S}_+^d$ and any loss function $f : \mathbb{R}^d \to \mathbb{R}$, it always holds that $L_{\|\cdot\|_{\mathcal{H}}}(f) \leq \Lambda_{\mathcal{H}}(f) \leq d \cdot L_{\|\cdot\|_{\mathcal{H}}}(f)$.*

*Proof of Proposition 2.5.* For any $\boldsymbol{H} \in \mathcal{H}$ with $\mathrm{Tr}(\boldsymbol{H}) \leq 1$, by the definition of $\|\cdot\|_{\mathcal{H}}$ and $\|\cdot\|_{\mathcal{H},*}$, it holds for any $\boldsymbol{x} \in \mathbb{R}^d$ that $\|\boldsymbol{x}\|_{\mathcal{H}} \geq \|\boldsymbol{x}\|_{\boldsymbol{H}}$ and $\|\boldsymbol{x}\|_{\mathcal{H},*} \leq \|\boldsymbol{x}\|_{\boldsymbol{H},*}$. Then by the definition of $L_{\|\cdot\|}(f)$ in Definition 2.4,

$$L_{\|\cdot\|_{\mathcal{H}}}(f) = \sup_{\boldsymbol{x},\boldsymbol{y}}\frac{\|\nabla f(\boldsymbol{x}) - \nabla f(\boldsymbol{y})\|_{\mathcal{H},*}}{\|\boldsymbol{x} - \boldsymbol{y}\|_{\mathcal{H}}} \leq \sup_{\boldsymbol{x},\boldsymbol{y}}\frac{\|\nabla f(\boldsymbol{x}) - \nabla f(\boldsymbol{y})\|_{\boldsymbol{H},*}}{\|\boldsymbol{x} - \boldsymbol{y}\|_{\boldsymbol{H}}} = L_{\|\cdot\|_{\boldsymbol{H}}}(f).$$

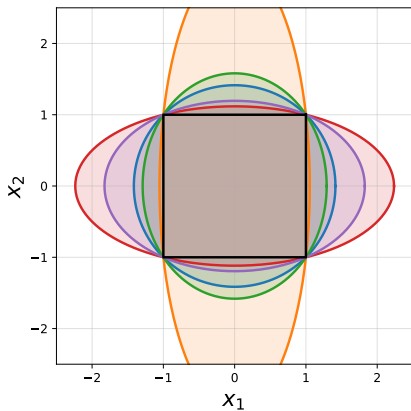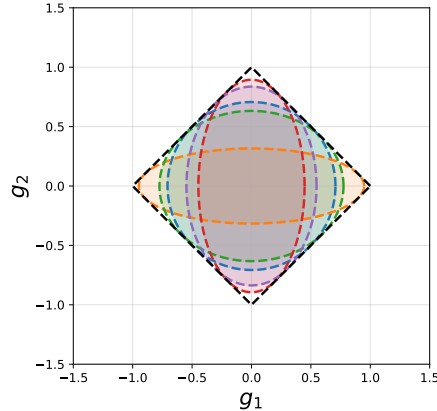

Figure 1: Here we demonstrate the duality between the supremum of the primal norms and the infimum of the corresponding dual norms for any well-structured preconditioner set $\mathcal{H}$. In particular, we consider $\mathcal{H} = \{\text{all diagonal PSD matrices}\}$, in which case $\|\cdot\|_{\mathcal{H}} = \|\cdot\|_{\infty}$ and $\|\cdot\|_{\mathcal{H},*} = \|\cdot\|_1$. **Left figure**: the $\|\cdot\|_{\infty}$-unit ball (black square) in the primal space is the intersection of all $\|\cdot\|_{H}$-unit ball (colored ellipses) for $H \in \mathcal{H}$ with $\text{Tr}(H) \leq 1$, that is, $\|\cdot\|_{\infty}$ is the supremum of all such primal $\|\cdot\|_{H}$ norms. **Right figure**: the $\|\cdot\|_1$-unit ball (dashed black square) in the dual space is the union of all $\|\cdot\|_{H,*}$-unit balls (dashed ellipses) for $H \in \mathcal{H}$ with $\text{Tr}(H) \leq 1$, that is, $\|\cdot\|_1$ is the infimum of all such dual $\|\cdot\|_{H,*}$ norms.

Therefore, further minimizing both sides over $H \in \mathcal{H}$ with $\text{Tr}(H) \leq 1$, we obtain the desired result.

On the other hand, we can choose $H^* = \frac{1}{d} I_d \in \mathcal{H}$ and Xie et al. (2025b) shows that $\|\cdot\|_{H^*} = \frac{1}{\sqrt{d}} \|\cdot\|_2$. Then we have that $\Lambda_{\mathcal{H}}(f) \leq L_{\|\cdot\|_{H^*}}(f)$.

We can also choose $\mathcal{H}^*$ to be all PSD matrices so that we know $\mathcal{H} \subseteq \mathcal{H}^*$. Then we know $\|x\|_{\mathcal{H}} = \sup_{H \in \mathcal{H}, \text{Tr}(H) \leq 1} \|x\|_{H} \leq \sup_{H \in \mathcal{H}^*, \text{Tr}(H) \leq 1} \|x\|_{H} = \|x\|_{\mathcal{H}^*}$ and $\|x\|_{\mathcal{H},*} = \inf_{H \in \mathcal{H}, \text{Tr}(H) \leq 1} \|x\|_{H^{-1}} \geq \inf_{H \in \mathcal{H}^*, \text{Tr}(H) \leq 1} = \|x\|_{\mathcal{H}^*,*}$. Xie et al. (2025b) also shows that $\|x\|_{\mathcal{H}^*} = \|x\|_{\mathcal{H}^*,*} = \|x\|_2$. So we have that

$$
\begin{aligned}
L_{\|\cdot\|_{\mathcal{H}}}(f) &= \sup_{x,y} \frac{\|\nabla f(x) - \nabla f(y)\|_{\mathcal{H},*}}{\|x - y\|_{\mathcal{H}}} \\
&\geq \sup_{x,y} \frac{\|\nabla f(x) - \nabla f(y)\|_2}{\|x - y\|_2} \\
&= \frac{1}{d} \sup_{x,y} \frac{\sqrt{d} \|\nabla f(x) - \nabla f(y)\|_2}{\frac{1}{\sqrt{d}} \|x - y\|_2} \\
&= \frac{1}{d} \sup_{x,y} \frac{\|\nabla f(x) - \nabla f(y)\|_{H^{*-1}}}{\|x - y\|_{H^*}} \\
&\geq \frac{1}{d} \Lambda_{\mathcal{H}}(f).
\end{aligned}
$$

$\square$

**Proposition B.11.** *For any set of stochastic loss functions $\{f_t\}_{t \in \mathcal{T}}$ and any well-structured preconditioner set $\mathcal{H}$, it always holds $\sigma_{\|\cdot\|_{\mathcal{H},*}}(\{f_t\}_{t \in \mathcal{T}})^2 \leq \sigma_{\mathcal{H}}(\{f_t\}_{t \in \mathcal{T}})^2 \leq d \cdot \sigma_{\|\cdot\|_{\mathcal{H},*}}(\{f_t\}_{t \in \mathcal{T}})^2$.*

*Proof.* For any $H \in \mathcal{H}$ with $\text{Tr}(H) \leq 1$, it always holds that $\|u\|_{\mathcal{H},*}^2 \leq \|u\|_{H^{-1}}^2$ for any $u \in \mathbb{R}^d$, and thus

$$
\sup_{t \in \mathcal{T}, x \in \mathbb{R}^d} \mathbb{E}\big[\|\nabla f_t(x) - \mathbb{E}[\nabla f_t(x)]\|_{\mathcal{H},*}^2\big] \leq \sup_{t \in \mathcal{T}, x \in \mathbb{R}^d} \mathbb{E}\big[\|\nabla f_t(x) - \mathbb{E}[\nabla f_t(x)]\|_{H^{-1}}^2\big].
$$

By minimizing over $\boldsymbol{H} \in \mathcal{H}$ with $\mathrm{Tr}(\boldsymbol{H}) \leq 1$, we conclude that $\sigma_{\|\cdot\|_{\mathcal{H},*}}(\{f_t\}_{t \in \mathcal{T}})^2 \leq \sigma_{\mathcal{H}}(\{f_t\}_{t \in \mathcal{T}})^2$.

Similar with the proof of Proposition 2.5, we can show that $\sigma_{\mathcal{H}}(\{f_t\}_{t \in \mathcal{T}})^2 \leq \sup_{\boldsymbol{x},t \in [T]} \mathbb{E} d \|\nabla f(\boldsymbol{x}) - \nabla f_t(\boldsymbol{x})\|_2^2$ when choosing $\boldsymbol{H}^* = \frac{1}{d} \boldsymbol{I}_d$ . We can also show that $\sup_{t \in \mathcal{T}, \boldsymbol{x} \in \mathbb{R}^d} \mathbb{E} \|\nabla f_t(\boldsymbol{x}) - \mathbb{E}[\nabla f_t(\boldsymbol{x})]\|_{\mathcal{H},*}^2 \geq \sup_{t \in \mathcal{T}, \boldsymbol{x} \in \mathbb{R}^d} \mathbb{E} \|\nabla f_t(\boldsymbol{x}) - \mathbb{E}[\nabla f_t(\boldsymbol{x})]\|_2^2$. Then combining them, we have that $d \sup_{t \in \mathcal{T}, \boldsymbol{x} \in \mathbb{R}^d} \mathbb{E} \|\nabla f_t(\boldsymbol{x}) - \mathbb{E}[\nabla f_t(\boldsymbol{x})]\|_{\mathcal{H},*}^2 \geq \sigma_{\mathcal{H}}(\{f_t\}_{t \in \mathcal{T}})^2$. $\quad\square$

## C  PROOF FOR THE MATRIX INEQUALITY

**Lemma 3.3.** *Let $\epsilon \geq 0$ and $\beta \in (0,1]$. For any $T \in \mathbb{N}$, consider **any sequence of vectors** $\boldsymbol{g}_0, \ldots, \boldsymbol{g}_{T-1} \in \mathbb{R}^d$. Let $\boldsymbol{M}_{-1} = 0$, and recursively define $\boldsymbol{M}_t = \beta \boldsymbol{M}_{t-1} + \boldsymbol{g}_t \boldsymbol{g}_t^\top$ for $t = 0, \ldots, T-1$. For any well-structured preconditioner set $\mathcal{H}$, define $\boldsymbol{V}_t = \arg\min_{\boldsymbol{H} \in \mathcal{H}} \langle \boldsymbol{M}_t + \epsilon \boldsymbol{I}_d, \boldsymbol{H}^{-1} \rangle + \mathrm{Tr}(\boldsymbol{H})$ for each $t \in [T-1]$. Then for any $\boldsymbol{H} \in \mathcal{H} \cap \mathcal{S}_{++}^d$, it holds*

$$\sum_{t=0}^{T-1} \|\boldsymbol{V}_t^{-1} \boldsymbol{g}_t\|_{\boldsymbol{H}}^2 \leq \mathrm{Tr}(\boldsymbol{H}) \|\boldsymbol{S}_T\|_{\mathrm{op}} \quad \text{where } \boldsymbol{S}_T = \sum_{t=0}^{T-1} \boldsymbol{V}_t^{-1} (\boldsymbol{V}_t^2 - \beta \boldsymbol{V}_{t-1}^2) \boldsymbol{V}_t^{-1}.$$

*Moreover, there exists an absolute constant $C_1, C_2 > 0$, independent of $d, T, \epsilon, \beta$ and $\mathcal{H}$, such that*

$$\|\boldsymbol{S}_T\|_{\mathrm{op}} \leq C_1 \left(1 + \log\left(1 + \frac{d}{\epsilon} \sum_{t=0}^{T-1} \|\boldsymbol{g}_t\|_2^2 + d^2(1-\beta)T\right)\right) \left(\frac{1-\beta}{\beta}T + \log \|\boldsymbol{V}_{T-1}^2/\epsilon\|_{\mathrm{op}}\right) + C_2.$$

*In the special case when $\mathcal{H}$ is commutative, the above bound can be further improved to*

$$\|\boldsymbol{S}_T\|_{\mathrm{op}} \leq (1-\beta)T + \log \|\boldsymbol{V}_{T-1}^2/\epsilon\|_{\mathrm{op}}.$$

*Proof of Lemma 3.3.* For each $t = 0, \ldots, T-1$, by the definition of $\{\boldsymbol{M}_t\}_{t=0}^{T-1}$, we have

$$\|\boldsymbol{V}_t^{-1} \boldsymbol{g}_t\|_{\boldsymbol{H}}^2 = \langle \boldsymbol{V}_t^{-1} \boldsymbol{H} \boldsymbol{V}_t^{-1}, \boldsymbol{g}_t \boldsymbol{g}_t^\top \rangle = \langle \boldsymbol{V}_t^{-1} \boldsymbol{H} \boldsymbol{V}_t^{-1}, \boldsymbol{M}_t + \epsilon \boldsymbol{I}_d \rangle - \beta \langle \boldsymbol{V}_t^{-1} \boldsymbol{H} \boldsymbol{V}_t^{-1}, \boldsymbol{M}_{t-1} + \epsilon \boldsymbol{I}_d \rangle.$$

Since $\boldsymbol{V}_t^{-1} \boldsymbol{H} \boldsymbol{V}_t^{-1} \in \mathcal{H}$, it follows from Lemma B.2 and the definition of $\{\boldsymbol{V}_t\}_{t=0}^{T-1}$ that

$$\begin{aligned}
\|\boldsymbol{V}_t^{-1} \boldsymbol{g}_t\|_{\boldsymbol{H}}^2 &= \langle \boldsymbol{V}_t^{-1} \boldsymbol{H} \boldsymbol{V}_t^{-1}, P_{\mathcal{H}}(\boldsymbol{M}_t + \epsilon \boldsymbol{I}_d)^2 \rangle - \beta \langle \boldsymbol{V}_t^{-1} \boldsymbol{H} \boldsymbol{V}_t^{-1}, P_{\mathcal{H}}(\boldsymbol{M}_{t-1} + \epsilon \boldsymbol{I}_d)^2 \rangle \\
&= \langle \boldsymbol{V}_t^{-1} \boldsymbol{H} \boldsymbol{V}_t^{-1}, \boldsymbol{V}_t^2 \rangle - \beta \langle \boldsymbol{V}_t^{-1} \boldsymbol{H} \boldsymbol{V}_t^{-1}, \boldsymbol{V}_{t-1}^2 \rangle \\
&= \langle \boldsymbol{H}, \boldsymbol{V}_t^{-1} (\boldsymbol{V}_t^2 - \beta \boldsymbol{V}_{t-1}^2) \boldsymbol{V}_t^{-1} \rangle
\end{aligned}$$

Then summing over $t = 0, \ldots, T-1$ gives

$$\begin{aligned}
\sum_{t=0}^{T-1} \|\boldsymbol{V}_t^{-1} \boldsymbol{g}_t\|_{\boldsymbol{H}}^2 &= \sum_{t=0}^{T-1} \langle \boldsymbol{H}, \boldsymbol{V}_t^{-1} (\boldsymbol{V}_t^2 - \beta \boldsymbol{V}_{t-1}^2) \boldsymbol{V}_t^{-1} \rangle \\
&\leq \mathrm{Tr}(\boldsymbol{H}) \left\| \sum_{t=0}^{T-1} \boldsymbol{V}_t^{-1} (\boldsymbol{V}_t^2 - \beta \boldsymbol{V}_{t-1}^2) \boldsymbol{V}_t^{-1} \right\|_{\mathrm{op}}.
\end{aligned}$$

This shows the first part of the lemma. We next bound the spectral norm of $\sum_{t=0}^{T-1} \boldsymbol{V}_t^{-1} (\boldsymbol{V}_t^2 - \beta \boldsymbol{V}_{t-1}^2) \boldsymbol{V}_t^{-1}$.

Note that $\beta \boldsymbol{V}_{t-1}^2 = \beta P_{\mathcal{H}}(\boldsymbol{M}_{t-1} + \epsilon \boldsymbol{I}_d)^2 = P_{\mathcal{H}}(\beta(\boldsymbol{M}_{t-1} + \epsilon \boldsymbol{I}_d))^2$ by Lemma B.3. Since $\boldsymbol{M}_t + \epsilon \boldsymbol{I}_d \succeq \beta(\boldsymbol{M}_{t-1} + \epsilon \boldsymbol{I}_d)$, we further have $\boldsymbol{V}_t^2 \succeq \beta \boldsymbol{V}_{t-1}^2 \succ 0$ by Lemma B.3. Therefore, since $\lambda_{\min}(\boldsymbol{V}_t^2) \geq \epsilon$,

we can apply Lemma C.1 to get that for any $C \geq c > 0$,

$$
\boldsymbol{V}_t^{-1}(\boldsymbol{V}_t^2 - \beta \boldsymbol{V}_{t-1}^2)\boldsymbol{V}_t^{-1} \preceq \frac{3(\log C - \log c)}{\pi^2}(\log(\boldsymbol{V}_t^2) - \log(\beta \boldsymbol{V}_{t-1}^2)) + \left(\frac{12cd}{\pi^2 \epsilon^2}\right.
$$

$$
\left. + \frac{12C^{-1}d}{\pi^2}\right) \mathrm{Tr}(\boldsymbol{V}_t^2 - \beta \boldsymbol{V}_{t-1}^2)\boldsymbol{I}_d.
$$

$$
= \frac{3(\log C - \log c)}{\pi^2} \log \frac{1}{\beta} \cdot \boldsymbol{I}_d + \frac{3(\log C - \log c)}{\pi^2}(\log(\boldsymbol{V}_t^2) - \log(\boldsymbol{V}_{t-1}^2))
$$

$$
+ \left(\frac{12cd}{\pi^2 \epsilon^2} + \frac{12C^{-1}d}{\pi^2}\right) \mathrm{Tr}[\boldsymbol{M}_t + \epsilon \boldsymbol{I}_d - \beta(\boldsymbol{M}_{t-1} + \epsilon \boldsymbol{I}_d)]\boldsymbol{I}_d
$$

$$
= \frac{3(\log C - \log c)}{\pi^2} \log \frac{1}{\beta} \cdot \boldsymbol{I}_d + \frac{3(\log C - \log c)}{\pi^2}(\log(\boldsymbol{V}_t^2) - \log(\boldsymbol{V}_{t-1}^2))
$$

$$
+ \left(\frac{12cd}{\pi^2 \epsilon^2} + \frac{12C^{-1}d}{\pi^2}\right) \mathrm{Tr}[\boldsymbol{g}_t \boldsymbol{g}_t^\top + (1-\beta)\epsilon \boldsymbol{I}_d]\boldsymbol{I}_d
$$

where the first equality holds by Lemma B.2. Further summing over $t = 0, \ldots, T-1$ gives

$$
\sum_{t=0}^{T-1} \boldsymbol{V}_t^{-1}(\boldsymbol{V}_t^2 - \beta \boldsymbol{V}_{t-1}^2)\boldsymbol{V}_t^{-1} \preceq \frac{3(\log C - \log c)}{\pi^2} T \log \frac{1}{\beta} \cdot \boldsymbol{I}_d + \frac{3(\log C - \log c)}{\pi^2}(\log(\boldsymbol{V}_{T-1}^2) - \log(\boldsymbol{V}_{-1}^2))
$$

$$
+ \left(\frac{12cd}{\pi^2 \epsilon^2} + \frac{12C^{-1}d}{\pi^2}\right) \sum_{t=0}^{T-1} \mathrm{Tr}[\boldsymbol{g}_t \boldsymbol{g}_t^\top + (1-\beta)\epsilon \boldsymbol{I}_d]\boldsymbol{I}_d
$$

$$
= \frac{3(\log C - \log c)}{\pi^2}\left(T \log \frac{1}{\beta} \cdot \boldsymbol{I}_d + \log(\boldsymbol{V}_{T-1}^2/\epsilon)\right)
$$

$$
+ \left(\frac{12cd}{\pi^2 \epsilon^2} + \frac{12C^{-1}d}{\pi^2}\right)\left(d(1-\beta)\epsilon T + \sum_{t=0}^{T-1} \|\boldsymbol{g}_t\|_2^2\right)\boldsymbol{I}_d
$$

where the equality holds as $\boldsymbol{V}_{-1} = P_{\mathcal{H}}(\epsilon \boldsymbol{I}_d) = \sqrt{\epsilon}\boldsymbol{I}_d$. Note that $\log(1/\beta) = \log(1 + (1-\beta)/\beta) \leq (1-\beta)/\beta$ for any $\beta \in (0, 1]$. Then by triangle inequality for the spectral norm, we have

$$
\left\|\sum_{t=0}^{T-1} \boldsymbol{V}_t^{-1}(\boldsymbol{V}_t^2 - \beta \boldsymbol{V}_{t-1}^2)\boldsymbol{V}_t^{-1}\right\|_{\mathrm{op}} \leq \frac{3(\log C - \log c)}{\pi^2}\left(\frac{1-\beta}{\beta}T + \log \left\|\boldsymbol{V}_{T-1}^2/\epsilon\right\|_{\mathrm{op}}\right)
$$

$$
+ \left(\frac{12cd}{\pi^2 \epsilon^2} + \frac{12C^{-1}d}{\pi^2}\right)\left(d(1-\beta)T\epsilon + \sum_{t=0}^{T-1} \|\boldsymbol{g}_t\|_2^2\right).
$$

$$
(10)
$$

In particular, we choose

$$
c = \frac{(1-\beta)T/\beta + \log \left\|\boldsymbol{V}_{T-1}^2/\epsilon\right\|_{\mathrm{op}}}{4d(d(1-\beta)T\epsilon + \sum_{t=0}^{T-1} \|\boldsymbol{g}_t\|_2^2)}\epsilon^2, \quad C = \max(\epsilon^2/c, c). \quad (11)
$$

With this, the second term on the right-hand side of (10) satisfies that

$$
\left(\frac{12cd}{\pi^2 \epsilon^2} + \frac{12C^{-1}d}{\pi^2}\right)\left(d(1-\beta)T\epsilon + \sum_{t=0}^{T-1} \|\boldsymbol{g}_t\|_2^2\right) \leq \frac{6}{\pi^2}\left(\frac{1-\beta}{\beta}T + \log \left\|\boldsymbol{V}_{T-1}^2/\epsilon\right\|_{\mathrm{op}}\right).
$$

Plugging this back into (10), we obtain

$$
\left\|\sum_{t=0}^{T-1} \boldsymbol{V}_t^{-1}(\boldsymbol{V}_t^2 - \beta \boldsymbol{V}_{t-1}^2)\boldsymbol{V}_t^{-1}\right\|_{\mathrm{op}} \leq \frac{6 + 3\log(C/c)}{\pi^2}\left(\frac{1-\beta}{\beta}T + \log \left\|\boldsymbol{V}_{T-1}^2/\epsilon\right\|_{\mathrm{op}}\right).
$$

Moreover, by the choice of $C, c$ in (11),

$$
\log \frac{C}{c} = \log(\max(\epsilon^2/c^2, 1)) = 2\log\left(\max\left(\frac{4d(d(1-\beta)T\epsilon + \sum_{t=0}^{T-1} \|\boldsymbol{g}_t\|_2^2)}{(1-\beta)T/\beta + \log \|\boldsymbol{V}_{T-1}^2/\epsilon\|_{\mathrm{op}}}, 1\right)\right).
$$

For convenience, denote $A = (1-\beta)T/\beta + \log\|\boldsymbol{V}_{T-1}^2/\epsilon\|_{\mathrm{op}}$ and $B = 4d(d(1-\beta)T\epsilon + \sum_{T=0}^{T-1}\|\boldsymbol{g}_t\|_2^2$. Then combining the above two displays, we get

$$\left\|\sum_{t=0}^{T-1}\boldsymbol{V}_t^{-1}(\boldsymbol{V}_t^2 - \beta\boldsymbol{V}_{t-1}^2)\boldsymbol{V}_t^{-1}\right\|_{\mathrm{op}} \leq \frac{6}{\pi^2}A + \frac{3}{\pi^2}A\log\left(\max\left(\frac{B}{A}, 1\right)\right).$$

When $A > 1$, we always have $\max(B/A, 1) \leq 1 + B$, which gives rise to the following bound

$$\left\|\sum_{t=0}^{T-1}\boldsymbol{V}_t^{-1}(\boldsymbol{V}_t^2 - \beta\boldsymbol{V}_{t-1}^2)\boldsymbol{V}_t^{-1}\right\|_{\mathrm{op}} \leq \frac{6}{\pi^2}\big(1 + \log(1 + B)\big)A.$$

When $A < 1$, using the fact that $x\log(1/x) \leq 1/e$, we have

$$A\log\left(\max\left(\frac{B}{A}, 1\right)\right) \leq A\log(1 + B) - A\log A \leq A\log(1 + B) + 1/e.$$

In this case, the bound becomes

$$\left\|\sum_{t=0}^{T-1}\boldsymbol{V}_t^{-1}(\boldsymbol{V}_t^2 - \beta\boldsymbol{V}_{t-1}^2)\boldsymbol{V}_t^{-1}\right\|_{\mathrm{op}} \leq \frac{6}{\pi^2}\Big[\big(1 + \log(1 + B)\big)A + 1/e\Big].$$

Combining the above two case, we can conclude that there exists universal constants $C_1, C_2 > 0$ such that

$$\left\|\sum_{t=0}^{T-1}\boldsymbol{V}_t^{-1}(\boldsymbol{V}_t^2 - \beta\boldsymbol{V}_{t-1}^2)\boldsymbol{V}_t^{-1}\right\|_{\mathrm{op}} \leq C_1\big(1 + \log(1 + B)\big)A + C_2.$$

This completes the proof for general well-structured preconditioner set $\mathcal{H}$.

In this special case where matrices in $\mathcal{H}$ are diagonal, we have that

$$\sum_{t=0}^{T-1}\boldsymbol{V}_t^{-1}(\boldsymbol{V}_t^2 - \beta\boldsymbol{V}_{t-1}^2)\boldsymbol{V}_t^{-1} = (1-\beta)T\boldsymbol{I}_d + \beta\sum_{t=0}^{T-1}(\boldsymbol{I}_d - \boldsymbol{V}_t^{-1}\boldsymbol{V}_{t-1}^2\boldsymbol{V}_t^{-1})$$

$$\preceq (1-\beta)T\boldsymbol{I}_d + \beta\sum_{t=0}^{T-1}\log(\boldsymbol{V}_t\boldsymbol{V}_{t-1}^{-2}\boldsymbol{V}_t)$$

where the inequality follows from the fact that $\boldsymbol{I}_d - \boldsymbol{A}^{-1} \preceq \log\boldsymbol{A}$ for any $\boldsymbol{A} \succeq 0$. Further note that as $\boldsymbol{V}_{t-1}, \boldsymbol{V}_t$ are diagonal matrices, they commute with each other. Then using the fact that $\log(\boldsymbol{A}\boldsymbol{B}) = \log\boldsymbol{A} + \log\boldsymbol{B}$ when $\boldsymbol{A}$ and $\boldsymbol{B}$ commute, we obtain

$$\sum_{t=0}^{T-1}\boldsymbol{V}_t^{-1}(\boldsymbol{V}_t^2 - \beta\boldsymbol{V}_{t-1}^2)\boldsymbol{V}_t^{-1} \preceq (1-\beta)T\boldsymbol{I}_d + \beta\sum_{t=0}^{T-1}[\log(\boldsymbol{V}_t^2) - \log(\boldsymbol{V}_{t-1}^2)]$$

$$= (1-\beta)T\boldsymbol{I}_d + \log\left(\boldsymbol{V}_{T-1}^2/\epsilon\right).$$

Therefore, in this case it holds that

$$\left\|\sum_{t=0}^{T-1}\boldsymbol{V}_t^{-1}(\boldsymbol{V}_t^2 - \beta\boldsymbol{V}_{t-1}^2)\boldsymbol{V}_t^{-1}\right\|_{\mathrm{op}} \leq (1-\beta)T + \log\|\boldsymbol{V}_{T-1}^2/\epsilon\|_{\mathrm{op}}.$$

This completes the proof. $\qquad\square$

**Lemma C.1.** *For any positive definite matrices $\boldsymbol{X}, \boldsymbol{Y} \in \mathbb{R}^{d\times d}$ such that $\boldsymbol{Y} \preceq \boldsymbol{X}$, it holds for any $0 \leq c \leq C$ that*

$$\boldsymbol{X}^{-1/2}(\boldsymbol{X} - \boldsymbol{Y})\boldsymbol{X}^{-1/2} \preceq \frac{3(\log C - \log c)}{\pi^2}(\log\boldsymbol{X} - \log\boldsymbol{Y}) + \left(\frac{12cd}{\pi^2\lambda_{\min}(\boldsymbol{X})^2} + \frac{12C^{-1}d}{\pi^2}\right)\mathrm{Tr}(\boldsymbol{X} - \boldsymbol{Y})\cdot\boldsymbol{I}_d.$$

*Proof of Lemma C.1.* For any $\delta \in (0, 1)$, since the matrix logarithm is operator concave, it holds that

$$\log((1 - \delta)\boldsymbol{X} + \delta\boldsymbol{Y}) \succeq (1 - \delta)\log\boldsymbol{X} + \delta\log\boldsymbol{Y}.$$

Reorganizing the above inequality yields that for all $\delta \in (0, 1)$,

$$\log\boldsymbol{X} - \log\boldsymbol{Y} \succeq -\frac{\log(\boldsymbol{X} + \delta(\boldsymbol{Y} - \boldsymbol{X})) - \log\boldsymbol{X}}{\delta}.$$

Taking the limit $\delta \to 0$, we obtain

$$\log\boldsymbol{X} - \log\boldsymbol{Y} \succeq \partial\log(\boldsymbol{X})[\boldsymbol{X} - \boldsymbol{Y}].$$

The proof is completed by further applying Lemma C.2 with $\boldsymbol{A} = \boldsymbol{X} - \boldsymbol{Y}$. $\qquad\square$

**Lemma C.2.** *For any positive definite matrix $\boldsymbol{X} \in \mathbb{R}^{d \times d}$ and any positive semi-definite matrix $\boldsymbol{A} \in \mathbb{R}^{d \times d}$, it holds for any $0 \le c \le C$ that*

$$\boldsymbol{X}^{-1/2}\boldsymbol{A}\boldsymbol{X}^{-1/2} \preceq \frac{3(\log C - \log c)}{\pi^2}\partial\log(\boldsymbol{X})[\boldsymbol{A}] + \left(\frac{12cd}{\pi^2\lambda_{\min}(\boldsymbol{X})^2} + \frac{12C^{-1}d}{\pi^2}\right)\operatorname{Tr}(\boldsymbol{A})\cdot\boldsymbol{I}_d. \tag{12}$$

*Proof of Lemma C.2.* We consider the following expansion of the matrix logarithm:

$$\log(\boldsymbol{X} + \delta\boldsymbol{A}) = \log\boldsymbol{X} + \int_0^\infty (\boldsymbol{X} + z\boldsymbol{I})^{-1}\cdot\delta\boldsymbol{A}\cdot(\boldsymbol{X} + z\boldsymbol{I})^{-1}\mathrm{d}z + O(\delta^2).$$

This implies that

$$\partial\log(\boldsymbol{X})[\boldsymbol{A}] = \lim_{\delta\to 0}\frac{\log(\boldsymbol{X} + \delta\boldsymbol{A}) - \log\boldsymbol{X}}{\delta} = \int_0^\infty \frac{\boldsymbol{I}}{\boldsymbol{X} + z\boldsymbol{I}}\cdot\boldsymbol{A}\cdot\frac{\boldsymbol{I}}{\boldsymbol{X} + z\boldsymbol{I}}\mathrm{d}z.$$

Let $\boldsymbol{v} \in \mathbb{R}^d$ be an arbitrary unit vector. Note that both $\boldsymbol{X}^{-1/2}\boldsymbol{A}\boldsymbol{X}^{-1/2}$ and $\partial\log(\boldsymbol{X})[\boldsymbol{A}]$ are linear in $\boldsymbol{A}$, and thus we first consider $\boldsymbol{A} = \boldsymbol{u}\boldsymbol{u}^\top$ for any unit vector $\boldsymbol{u} \in \mathbb{R}^d$. We define the following two quantities

$$f(\boldsymbol{u}, \boldsymbol{v}) = \boldsymbol{v}^\top\partial\log(\boldsymbol{X})[\boldsymbol{u}\boldsymbol{u}^\top]\boldsymbol{v} = \boldsymbol{v}^\top\int_0^\infty (\boldsymbol{X} + z\boldsymbol{I}_d)^{-1}\boldsymbol{u}\boldsymbol{u}^\top(\boldsymbol{X} + z\boldsymbol{I}_d)^{-1}\mathrm{d}z\boldsymbol{v}, \tag{13}$$

$$g(\boldsymbol{u}, \boldsymbol{v}) = \boldsymbol{v}^\top\boldsymbol{X}^{-1/2}\boldsymbol{u}\boldsymbol{u}^\top\boldsymbol{X}^{-1/2}\boldsymbol{v}. \tag{14}$$

Now suppose that the SVD of $\boldsymbol{X}$ is $\boldsymbol{X} = \boldsymbol{U}\operatorname{diag}(\lambda_1, \ldots, \lambda_d)\boldsymbol{U}^\top$ where $\boldsymbol{U} \in \mathbb{R}^{d \times d}$ is an orthogonal matrix. Then $\boldsymbol{X}^{-1/2} = \boldsymbol{U}\operatorname{diag}(\lambda_1^{-1/2}, \ldots, \lambda_d^{-1/2})\boldsymbol{U}^\top$ and $(\boldsymbol{X} + z\boldsymbol{I})^{-1} = \boldsymbol{U}\operatorname{diag}(1/(\lambda_1 + z), \ldots, 1/(\lambda_d + z))\boldsymbol{U}^\top$. Writing $\tilde{\boldsymbol{u}} = \boldsymbol{U}\boldsymbol{u}$ and $\tilde{\boldsymbol{v}} = \boldsymbol{U}\boldsymbol{v}$, then $f(\boldsymbol{u}, \boldsymbol{v})$ becomes

$$f(\boldsymbol{u}, \boldsymbol{v}) = \int_0^\infty \tilde{\boldsymbol{v}}^\top\operatorname{diag}\left(\frac{1}{\lambda_1 + z}, \ldots, \frac{1}{\lambda_d + z}\right)\tilde{\boldsymbol{u}}\tilde{\boldsymbol{u}}^\top\operatorname{diag}\left(\frac{1}{\lambda_1 + z}, \ldots, \frac{1}{\lambda_d + z}\right)\tilde{\boldsymbol{v}}\mathrm{d}z$$

$$= \int_0^\infty \langle\tilde{\boldsymbol{u}}\odot\tilde{\boldsymbol{v}}, (1/(\lambda_1 + z), \ldots, 1/(\lambda_d + z))\rangle^2\mathrm{d}z.$$

Similarly, $g(\boldsymbol{u}, \boldsymbol{v})$ becomes

$$g(\boldsymbol{u}, \boldsymbol{v}) = \tilde{\boldsymbol{v}}^\top\operatorname{diag}(\lambda_1^{-1/2}, \ldots, \lambda_d^{-1/2})\tilde{\boldsymbol{u}}\tilde{\boldsymbol{u}}^\top\operatorname{diag}(\lambda_1^{-1/2}, \ldots, \lambda_d^{-1/2})\tilde{\boldsymbol{v}}$$

$$= \left\langle\tilde{\boldsymbol{u}}\odot\tilde{\boldsymbol{v}}, (1/\lambda_1^{-1/2}, \ldots, 1/\lambda_d^{-1/2})\right\rangle^2.$$

Now applying the fact that $\lambda^{-1/2} = \frac{1}{\pi}\int_0^\infty \frac{z^{-1/2}}{\lambda + z}\mathrm{d}z$ for any $\lambda > 0$, we have

$$g(\boldsymbol{u}, \boldsymbol{v}) = \left(\frac{1}{\pi}\int_0^\infty \left\langle\tilde{\boldsymbol{u}}\odot\tilde{\boldsymbol{v}}, (z^{-1/2}/(\lambda_1 + z), \ldots, z^{-1/2}/(\lambda_d + z))\right\rangle\mathrm{d}z\right)^2.$$

To further bound $g(\boldsymbol{u}, \boldsymbol{v})$, we split the integral into three parts: $[0, c]$, $[c, C]$, and $[C, \infty)$ where $c > 0$ and $C > 0$ are constants to be determined later. That is,

$$g(\boldsymbol{u}, \boldsymbol{v}) \leq \frac{3}{\pi^2} \left( \int_0^c \left\langle \tilde{\boldsymbol{u}} \odot \tilde{\boldsymbol{v}}, (z^{-1/2}/(\lambda_1 + z), \ldots, z^{-1/2}/(\lambda_d + z)) \right\rangle \mathrm{d}z \right)^2$$

$$+ \frac{3}{\pi^2} \left( \int_c^C \left\langle \tilde{\boldsymbol{u}} \odot \tilde{\boldsymbol{v}}, (z^{-1/2}/(\lambda_1 + z), \ldots, z^{-1/2}/(\lambda_d + z)) \right\rangle \mathrm{d}z \right)^2$$

$$+ \frac{3}{\pi^2} \left( \int_C^\infty \left\langle \tilde{\boldsymbol{u}} \odot \tilde{\boldsymbol{v}}, (z^{-1/2}/(\lambda_1 + z), \ldots, z^{-1/2}/(\lambda_d + z)) \right\rangle \mathrm{d}z \right)^2$$

$$=: g_1(\boldsymbol{u}, \boldsymbol{v}) + g_2(\boldsymbol{u}, \boldsymbol{v}) + g_3(\boldsymbol{u}, \boldsymbol{v})$$

where we apply the triangle inequality and denote the three terms on the right-hand side by $g_1(\boldsymbol{u}, \boldsymbol{v}), g_2(\boldsymbol{u}, \boldsymbol{v}), g_3(\boldsymbol{u}, \boldsymbol{v})$ respectively. We control each term separately.

First, for $g_1(\boldsymbol{u}, \boldsymbol{v})$,

$$g_1(\boldsymbol{u}, \boldsymbol{v}) = \frac{3}{\pi^2} \left( \int_0^c \langle \tilde{\boldsymbol{u}} \odot \tilde{\boldsymbol{v}}, (1/(\lambda_1 + z), \ldots, 1/(\lambda_d + z)) \rangle z^{-1/2} \mathrm{d}z \right)^2$$

$$\leq \frac{3}{\pi^2} \left( \int_0^c \|\tilde{\boldsymbol{u}} \odot \tilde{\boldsymbol{v}}\|_2 \left( \sum_{i=1}^d \frac{1}{(\lambda_i + z)^2} \right)^{1/2} z^{-1/2} \mathrm{d}z \right)^2$$

$$\leq \frac{3}{\pi^2} \left( \frac{\|\tilde{\boldsymbol{u}} \odot \tilde{\boldsymbol{v}}\|_2 \sqrt{d}}{\min_{i \in [d]} \lambda_i} \int_0^c z^{-1/2} \mathrm{d}z \right)^2$$

$$= \frac{12 \|\tilde{\boldsymbol{u}} \odot \tilde{\boldsymbol{v}}\|_2^2 \cdot cd}{\pi^2 \cdot \min_{i \in [d]} \lambda_i^2}$$

where the first inequality follows from Cauchy-Schwarz inequality and in the second inequality we apply $\frac{1}{\lambda_i + z} \leq \frac{1}{\lambda_i}$ as each $\lambda_i$ is positive. Next, for the integral over $[c, C]$, we have

$$g_2(\boldsymbol{u}, \boldsymbol{v}) = \frac{3}{\pi^2} \left( \int_c^C \langle \tilde{\boldsymbol{u}} \odot \tilde{\boldsymbol{v}}, (1/(\lambda_1 + z), \ldots, 1/(\lambda_d + z)) \rangle z^{-1/2} \mathrm{d}z \right)^2$$

$$\leq \frac{3}{\pi^2} \left( \int_c^C \langle \tilde{\boldsymbol{u}} \odot \tilde{\boldsymbol{v}}, (1/(\lambda_1 + z), \ldots, 1/(\lambda_d + z)) \rangle^2 \mathrm{d}z \right) \left( \int_c^C z^{-1} \mathrm{d}z \right)$$

$$\leq \frac{3(\log C - \log c)}{\pi^2} \left( \int_0^\infty \langle \tilde{\boldsymbol{u}} \odot \tilde{\boldsymbol{v}}, (1/(\lambda_1 + z), \ldots, 1/(\lambda_d + z)) \rangle^2 \mathrm{d}z \right)$$

$$= \frac{3(\log C - \log c)}{\pi^2} f(\boldsymbol{u}, \boldsymbol{v})$$

where the first inequality follows from Cauchy-Schwarz inequality, and in the second inequality we relax the domain of the integral to $[0, \infty)$, which exactly gives us $f(\boldsymbol{u}, \boldsymbol{v})$. Finally, for $g_3(\boldsymbol{u}, \boldsymbol{v})$,

$$g_3(\boldsymbol{u}, \boldsymbol{v}) = \frac{3}{\pi^2} \left( \int_C^\infty \langle \tilde{\boldsymbol{u}} \odot \tilde{\boldsymbol{v}}, (1/(\lambda_1 + z), \ldots, 1/(\lambda_d + z)) \rangle z^{-1/2} \mathrm{d}z \right)^2$$

$$\leq \frac{3}{\pi^2} \left( \int_C^\infty \|\tilde{\boldsymbol{u}} \odot \tilde{\boldsymbol{v}}\|_2 \left( \sum_{i=1}^d \frac{1}{(\lambda_i + z)^2} \right)^{1/2} z^{-1/2} \mathrm{d}z \right)^2$$

$$\leq \frac{3}{\pi^2} \left( \int_C^\infty \|\tilde{\boldsymbol{u}} \odot \tilde{\boldsymbol{v}}\|_2 \sqrt{d} z^{-3/2} \mathrm{d}z \right)^2$$

$$= \frac{12 \|\tilde{\boldsymbol{u}} \odot \tilde{\boldsymbol{v}}\|_2^2 \cdot C^{-1} d}{\pi^2}$$

where the first inequality follows from Cauchy-Schwarz inequality and in the second inequality we apply $\frac{1}{\lambda_i + z} \leq \frac{1}{z}$ as each $\lambda_i$ is positive. Collecting the above bounds, we obtain

$$g(\boldsymbol{u}, \boldsymbol{v}) \leq \frac{3(\log C - \log c)}{\pi^2} f(\boldsymbol{u}, \boldsymbol{v}) + \frac{12 \|\tilde{\boldsymbol{u}} \odot \tilde{\boldsymbol{v}}\|_2^2 \cdot cd}{\pi^2 \cdot \min_{i \in [d]} \lambda_i^2} + \frac{12 \|\tilde{\boldsymbol{u}} \odot \tilde{\boldsymbol{v}}\|_2^2 \cdot C^{-1} d}{\pi^2}. \tag{15}$$

Now for any general positive semi-definite matrix $\boldsymbol{A} \in \mathbb{R}^{d \times d}$ with eigendecomposition $\boldsymbol{A} = \sum_{i=1}^{d} \alpha_i \boldsymbol{u}_i \boldsymbol{u}_i^{\top}$, we apply the bound (15) to each $\boldsymbol{u}_i$ in the eigendecomposition and sum over all $i$ to get

$$\sum_{i=1}^{d} \alpha_i g(\boldsymbol{u}_i, \boldsymbol{v}) \leq \frac{3(\log C - \log c)}{\pi^2} \sum_{i=1}^{d} \alpha_i f(\boldsymbol{u}_i, \boldsymbol{v}) + \frac{12cd}{\pi^2 \cdot \min_{i \in [d]} \lambda_i^2} \sum_{i=1}^{d} \alpha_i \|\tilde{\boldsymbol{u}}_i \odot \tilde{\boldsymbol{v}}\|_2^2$$
$$+ \frac{12C^{-1}d}{\pi^2} \sum_{i=1}^{d} \alpha_i \|\tilde{\boldsymbol{u}}_i \odot \tilde{\boldsymbol{v}}\|_2^2.$$

Note that $\|\tilde{\boldsymbol{u}}_i \odot \tilde{\boldsymbol{v}}\|_2^2 \leq \|\tilde{\boldsymbol{u}}\|_2^2 \|\tilde{\boldsymbol{v}}\|_2^2 = 1$ as both $\tilde{\boldsymbol{u}}$ and $\tilde{\boldsymbol{v}}$ are unit vectors. Therefore, we further have

$$\sum_{i=1}^{d} \alpha_i g(\boldsymbol{u}_i, \boldsymbol{v}) \leq \frac{3(\log C - \log c)}{\pi^2} \sum_{i=1}^{d} \alpha_i f(\boldsymbol{u}_i, \boldsymbol{v}) + \frac{12cd}{\pi^2 \cdot \min_{i \in [d]} \lambda_i^2} \sum_{i=1}^{d} \alpha_i + \frac{12C^{-1}d}{\pi^2} \sum_{i=1}^{d} \alpha_i$$
$$= \frac{3(\log C - \log c)}{\pi^2} \sum_{i=1}^{d} \alpha_i f(\boldsymbol{u}_i, \boldsymbol{v}) + \frac{12cd}{\pi^2 \cdot \min_{i \in [d]} \lambda_i^2} \operatorname{Tr}(\boldsymbol{A}) + \frac{12C^{-1}d}{\pi^2} \operatorname{Tr}(\boldsymbol{A}).$$

Recall the definition of $f(\boldsymbol{u}, \boldsymbol{v})$ and $g(\boldsymbol{u}, \boldsymbol{v})$ in (13) and (14). Then we have shown that for any unit vector $\boldsymbol{v} \in \mathbb{R}^d$,

$$\boldsymbol{v}^{\top} \boldsymbol{X}^{-1/2} \boldsymbol{A} \boldsymbol{X}^{-1/2} \boldsymbol{v} \leq \frac{3(\log C - \log c)}{\pi^2} \boldsymbol{v}^{\top} \partial \log(\boldsymbol{X})[\boldsymbol{A}] \boldsymbol{v} + \frac{12cd}{\pi^2 \cdot \min_{i \in [d]} \lambda_i^2} \operatorname{Tr}(\boldsymbol{A}) + \frac{12C^{-1}d}{\pi^2} \operatorname{Tr}(\boldsymbol{A}).$$

Therefore, we conclude that

$$\boldsymbol{X}^{-1/2} \boldsymbol{A} \boldsymbol{X}^{-1/2} \preceq \frac{3(\log C - \log c)}{\pi^2} \partial \log(\boldsymbol{X})[\boldsymbol{A}] + \left( \frac{12cd}{\pi^2 \lambda_{\min}(\boldsymbol{X})^2} + \frac{12C^{-1}d}{\pi^2} \right) \operatorname{Tr}(\boldsymbol{A}) \cdot \boldsymbol{I}_d.$$

$\square$

# D UNIFIED PROOF FOR ADAPTIVE ALGORITHMS

## D.1 RELATIONSHIP BETWEEN ADAPTIVE ALGORITHMS

- *Cumulative accumulation* (Algorithm 4) maintains the plain sum of past squared gradients, thereby giving equal weight to the entire gradient history. A famous example in this category is AdaGrad.

- *EMA accumulation* (Algorithm 5) computes an exponential moving average of past gradients, which yields a stationary estimate of recent gradient statistics. Notable examples include Adam and RMSProp.

- *Weighted accumulation* (Algorithm 6) applies geometrically decaying weights to past gradients, which differs from EMA accumulation in that it does not normalize the weights to sum to one.

---

**Algorithm 4** General Adaptive Cumulative Optimization Algorithm

---

**Hyperparam:** $\epsilon \geq 0$, total steps $T$, learning rate $\eta$, convex cone $\mathcal{H} \subset \mathcal{S}_+$, $\beta$
**Input:** initialization $\boldsymbol{x}_0$, stochastic loss functions $\{f_t\}_{t=1}^{T} : \mathbb{R}^{d_L \times d_R} \to \mathbb{R}$
  $\boldsymbol{M}_0 \leftarrow \boldsymbol{0}$
  **for** $t = 1, 2, \cdots, T$ :
   $\boldsymbol{g}_t \leftarrow \nabla f_t(\boldsymbol{x}_{t-1})$
   $\boldsymbol{M}_t \leftarrow \boldsymbol{M}_{t-1} + \boldsymbol{g}_t \boldsymbol{g}_t^{\top}$
   $\boldsymbol{V}_t \leftarrow \arg\min_{\boldsymbol{H} \in \mathcal{H}} \left\langle \boldsymbol{M}_t + \epsilon \boldsymbol{I}_d, \boldsymbol{H}^{-1} \right\rangle + \operatorname{Tr}(\boldsymbol{H})$
   $\boldsymbol{x}_t \leftarrow \boldsymbol{x}_{t-1} - \eta \boldsymbol{V}_t^{-1} \boldsymbol{g}_t$
  **return** $\boldsymbol{x}_T$

---

---

**Algorithm 5** General Adaptive EMA Optimization Algorithm

---

**Hyperparam:** $\epsilon \geq 0$, total steps $T$, learning rate $\eta$, convex cone $\mathcal{H} \subset \mathcal{S}_+$, $\beta$
**Input:** initialization $\boldsymbol{x}_0$, stochastic loss functions $\{f_t\}_{t=1}^T : \mathbb{R}^{d_L \times d_R} \to \mathbb{R}$
  $\boldsymbol{M}_0 \leftarrow \boldsymbol{0}$
  **for** $t = 1, 2, \cdots, T$ :
   $\boldsymbol{g}_t \leftarrow \nabla f_t(\boldsymbol{x}_{t-1})$
   $\boldsymbol{M}_t \leftarrow \beta \boldsymbol{M}_{t-1} + (1 - \beta)\boldsymbol{g}_t\boldsymbol{g}_t^\top$
   $\boldsymbol{V}_t \leftarrow \arg\min_{\boldsymbol{H} \in \mathcal{H}} \langle \boldsymbol{M}_t + \epsilon \boldsymbol{I}_d, \boldsymbol{H}^{-1} \rangle + \mathrm{Tr}(\boldsymbol{H})$
   $\boldsymbol{x}_t \leftarrow \boldsymbol{x}_{t-1} - \eta \boldsymbol{V}_t^{-1} \boldsymbol{g}_t$
  **return** $\boldsymbol{x}_T$

---

**Algorithm 6** General Adaptive Weighted Optimization Algorithm

---

**Hyperparam:** $\epsilon \geq 0$, total steps $T$, learning rate $\eta$, convex cone $\mathcal{H} \subset \mathcal{S}_+$, $\beta$
**Input:** initialization $\boldsymbol{x}_0$, stochastic loss functions $\{f_t\}_{t=1}^T : \mathbb{R}^{d_L \times d_R} \to \mathbb{R}$
  $\boldsymbol{M}_0 \leftarrow \boldsymbol{0}$
  **for** $t = 1, 2, \cdots, T$ :
   $\boldsymbol{g}_t \leftarrow \nabla f_t(\boldsymbol{x}_{t-1})$
   $\boldsymbol{M}_t \leftarrow \beta \boldsymbol{M}_{t-1} + \boldsymbol{g}_t\boldsymbol{g}_t^\top$
   $\boldsymbol{V}_t \leftarrow \arg\min_{\boldsymbol{H} \in \mathcal{H}} \langle \boldsymbol{M}_t + \epsilon \boldsymbol{I}_d, \boldsymbol{H}^{-1} \rangle + \mathrm{Tr}(\boldsymbol{H})$
   $\boldsymbol{x}_t \leftarrow \boldsymbol{x}_{t-1} - \eta \boldsymbol{V}_t^{-1} \boldsymbol{g}_t$
  **return** $\boldsymbol{x}_T$

---

We also put the definition of one-sided Shampoo (Xie et al., 2025b) in Algorithm 7.

---

**Algorithm 7** One-sided Shampoo

---

**Hyperparam:** $\epsilon \geq 0$, total steps $T$, learning rate $\eta$, initial $\boldsymbol{M}_0, \boldsymbol{L}_0 = \boldsymbol{0}$
**Input:** initialization $\boldsymbol{x}_0$, stochastic loss functions $\{f_t\}_{t=1}^T : \mathbb{R}^{d_L \times d_R} \to \mathbb{R}$
  **for** $t = 1, 2, \cdots, T$ :
   $\boldsymbol{G}_t \leftarrow \nabla f_t(\boldsymbol{X}_{t-1})$
   $\boldsymbol{L}_t \leftarrow \boldsymbol{L}_{t-1} + \boldsymbol{G}_t\boldsymbol{G}_t^\top$
   $\boldsymbol{X}_t \leftarrow \boldsymbol{X}_{t-1} - \eta(\boldsymbol{L}_t + \epsilon \boldsymbol{I}_{d_L})^{-\frac{1}{2}} \boldsymbol{G}_t$
  **return** $\boldsymbol{x}_T$

---

## D.2 PROOF FOR WEIGHTED ALGORITHM

Before presenting the main theorems, we first clarify our gradient noise assumption below. Assumption D.1 is stronger than the conventional noise assumption when it assumes the condition holds almost surely. It is required in proving Lemma D.3. Lemma D.5 can still hold when we only assume the covariance matrix is bounded.

**Assumption D.1.** *For any $t \in [T]$ and any $\boldsymbol{x} \in \mathbb{R}^d$, $\mathbb{E}[\nabla f_t(\boldsymbol{x})] = \nabla f(\boldsymbol{x})$ where the expectation is taken with respect to the randomness in the loss $f_t$. Moreover, there exists $\boldsymbol{\Sigma} \succeq 0$ such that for all $t \geq 0$, $-\boldsymbol{\Sigma} \preceq \nabla f(\boldsymbol{x})\nabla f(\boldsymbol{x})^\top - \nabla f_t(\boldsymbol{x})\nabla f_t(\boldsymbol{x})^\top \preceq \boldsymbol{\Sigma}$.*

With Assumption D.1 in place, Theorem D.2 presents the general result for weighted adaptive algorithms on stochastic nonconvex functions, which is then specialized to the cumulative and EMA variants in Theorem D.7 and D.8, respectively. When plugging $\boldsymbol{\Sigma} = \boldsymbol{0}$ in the stochastic rate, we can get the deterministic result in Section 3.2.

In this section, we will define $\boldsymbol{H}^* = \arg\min_{\boldsymbol{H} \in \mathcal{H}, \mathrm{Tr}(\boldsymbol{H}) \leq 1} L_{\|\cdot\|_{\boldsymbol{H}}}(f)$. For all the $\boldsymbol{M}_t$ and $\boldsymbol{V}_t$ in this section except the proof of Theorem D.7 and Theorem D.8, they are defined as in Algorithm 6, i.e., $\boldsymbol{M}_t = \beta \boldsymbol{M}_{t-1} + \boldsymbol{g}_t\boldsymbol{g}_t^\top$. We will define $\bar{\boldsymbol{g}}_t = \nabla f(\boldsymbol{x}_t)$ and

$$\tilde{\boldsymbol{M}}_t = \beta \boldsymbol{M}_{t-1} + \bar{\boldsymbol{g}}_t\bar{\boldsymbol{g}}_t^\top + \boldsymbol{\Sigma}, \quad \tilde{\boldsymbol{V}}_t = P_{\mathcal{H}}(\tilde{\boldsymbol{M}}_t + \epsilon \boldsymbol{I}_d). \tag{16}$$

Then it always holds $M_t \preceq \tilde{M}_t$ because of Assumption D.1. By Lemma B.3, it also holds $V_t \preceq \tilde{V}_t$.

**Theorem D.2.** *For any $\epsilon \geq 0$, $\beta \in (0, 1]$, $\eta > 0$, and $T \in \mathbb{N}$, let $\{x_t\}_{t=0}^T$ be the iterates of Algorithm 1 with well-structured preconditioner set $\mathcal{H}$, where the update of $M_t$ follows the weighted version, i.e., $M_t = \beta M_{t-1} + g_t g_t^\top$ for all $t \in [T]$. Let $\Lambda_\mathcal{H}(f)$ be the adaptive smoothness of the loss $f$ according to Definition 2.4. Then under Assumption D.1, it holds that*

$$\mathbb{E}\left[\frac{1}{T}\sum_{t=0}^{T-1} \|\nabla f(x_t)\|_{\mathcal{H},*}\right] \leq \frac{\sqrt{\sum_{i=0}^{T-1} \beta^{i/2}}}{T}\xi + \frac{\sqrt{\text{Tr}\left[P_\mathcal{H}\left((\sum_{i=0}^{T-1} \beta^i)\Sigma + \epsilon I_d\right)\right]}}{\sqrt{T}}\sqrt{\xi}.$$

*where $\xi$ is given by*

$$\xi = \frac{2\Delta_0}{\eta} + \eta\Lambda_\mathcal{H}(f)\|S_T\|_{\text{op}} + \sqrt{2}d\|\Sigma\|_{\text{op}}^{1/2}\|S_T\|_{\text{op}} \tag{17}$$

*and $S_T = \mathbb{E}\sum_{t=0}^{T-1} V_t^{-1}(V_t^2 - \beta V_{t-1}^2)V_t^{-1}$.*

*For general well-structured preconditioner set, $\|S_T\|_{\text{op}} = \tilde{O}\left(\log(d)[(1-\beta)T/\beta + \log(d)]\right)$. When the preconditioner set only has diagonal matrices, $\|S_T\|_{\text{op}} = (1-\beta)T + \tilde{O}(1)$.*

*Proof of Theorem D.2.* With the shorthand $\bar{g}_t = \nabla f(x_t)$, we need to bound

$$\mathbb{E}\left[\frac{1}{T}\sum_{t=0}^{T-1} \|\nabla f(x_t)\|_{\mathcal{H},*}\right] = \mathbb{E}\left[\frac{1}{T}\sum_{t=0}^{T-1} \|\bar{g}_t\|_{\mathcal{H},*}\right] = \mathbb{E}\left[\frac{1}{T}\sum_{t=0}^{T-1} \text{Tr}[P_\mathcal{H}(\bar{g}_t\bar{g}_t^\top)]\right] \tag{18}$$

where the second equality holds by Lemma B.8. By employing Lemma D.4 and Lemma D.5, we have that

$$\left(\mathbb{E}\sum_{t=0}^{T-1} \text{Tr}[P_\mathcal{H}(\bar{g}_t\bar{g}_t^\top)]\right)^2 \tag{19}$$

$$\leq \left(\mathbb{E}\sum_{t=0}^{T-1} \text{Tr}[\tilde{V}_t^{-1}\bar{g}_t\bar{g}_t^\top]\right)\left(\mathbb{E}\sum_{t=0}^{T-1} \text{Tr}[\tilde{V}_t]\right)$$

$$\leq \left(\mathbb{E}\sum_{t=0}^{T-1} \text{Tr}[\tilde{V}_t^{-1}\bar{g}_t\bar{g}_t^\top]\right)\left(T\cdot\text{Tr}\left[P_\mathcal{H}\left(\left(\sum_{i=0}^{T-1}\beta^i\right)\Sigma + \epsilon I_d\right)\right] + \left(\sum_{i=0}^{T-1}\beta^{i/2}\right)\mathbb{E}\sum_{t=0}^{T-1}\text{Tr}[\bar{g}_t\bar{g}_t^\top\tilde{V}_t^{-1}]\right). \tag{20}$$

It then suffices to bound the sum of $\text{Tr}[\tilde{V}_t^{-1}\bar{g}_t\bar{g}_t^\top]$. To this end, we apply Lemma D.3 to get

$$\mathbb{E}\sum_{t=0}^{T-1} \text{Tr}[\tilde{V}_t^{-1}\bar{g}_t\bar{g}_t^\top] \leq 2\mathbb{E}\sum_{t=0}^{T-1}\text{Tr}[V_t^{-1}g_t\bar{g}_t^\top] + \sqrt{2\|\Sigma\|_{\text{op}}}\mathbb{E}\sum_{t=0}^{T-1}\text{Tr}[V_t^{-2}g_tg_t^\top]. \tag{21}$$

Here, the second term on the right-hand side can be controlled by applying Lemma 3.3. For the first term on the right-hand side, we need to apply the descent lemma. Specifically, recall that $H^* = \arg\min_{H \in \mathcal{H}, \text{Tr}(H) \leq 1} L_{\|\cdot\|_H}(f)$, and then by second-order Taylor expansion,

$$f(x_{t+1}) \leq f(x_t) + \langle\nabla f(x_t), x_{t+1} - x_t\rangle + \frac{\Lambda_\mathcal{H}(f)}{2}\|x_{t+1} - x_t\|_{H^*}^2$$

$$= f(x_t) - \eta\langle\bar{g}_t, V_t^{-1}g_t\rangle + \frac{\eta^2\Lambda_\mathcal{H}(f)}{2}\|V_t^{-1}g_t\|_{H^*}^2.$$

By taking expectation on both sides and summing over $t = 0, 1, \ldots, T-1$, we get

$$\mathbb{E}[f(x_T) - f(x_0)] \leq -\eta\mathbb{E}\sum_{t=0}^{T-1}\text{Tr}[V_t^{-1}g_t\bar{g}_t^\top] + \frac{\eta^2\Lambda_\mathcal{H}(f)}{2}\mathbb{E}\sum_{t=0}^{T-1}\|V_t^{-1}g_t\|_{H^*}^2.$$

Rearranging the above inequality, we have

$$\mathbb{E}\sum_{t=0}^{T-1}\operatorname{Tr}[\boldsymbol{V}_t^{-1}\boldsymbol{g}_t\bar{\boldsymbol{g}}_t^\top] \le \frac{\mathbb{E}[f(\boldsymbol{x}_0)-f(\boldsymbol{x}_T)]}{\eta} + \frac{\eta\Lambda_{\mathcal{H}}(f)}{2}\mathbb{E}\sum_{t=0}^{T-1}\left\|\boldsymbol{V}_t^{-1}\boldsymbol{g}_t\right\|_{\boldsymbol{H}^*}^2$$

$$\le \frac{\Delta_0}{\eta} + \frac{\eta\Lambda_{\mathcal{H}}(f)}{2}\mathbb{E}\sum_{t=0}^{T-1}\left\|\boldsymbol{V}_t^{-1}\boldsymbol{g}_t\right\|_{\boldsymbol{H}^*}^2 \tag{22}$$

where the second inequality holds by the definition $\Delta_0 = f(\boldsymbol{x}_0)-\min_{\boldsymbol{x}} f(\boldsymbol{x})$. Therefore, combining (21) and (22) yields

$$\mathbb{E}\sum_{t=0}^{T-1}\operatorname{Tr}[\tilde{\boldsymbol{V}}_t^{-1}\bar{\boldsymbol{g}}_t\bar{\boldsymbol{g}}_t^\top] \le \frac{2\Delta_0}{\eta} + \eta\Lambda_{\mathcal{H}}(f)\mathbb{E}\sum_{t=0}^{T-1}\left\|\boldsymbol{V}_t^{-1}\boldsymbol{g}_t\right\|_{\boldsymbol{H}^*}^2 + \sqrt{2\left\|\boldsymbol{\Sigma}\right\|_{\mathrm{op}}}\mathbb{E}\sum_{t=0}^{T-1}\operatorname{Tr}[\boldsymbol{V}_t^{-2}\boldsymbol{g}_t\boldsymbol{g}_t^\top]. \tag{23}$$

Now, we apply Lemma 3.3 to both $\mathbb{E}\sum_{t=0}^{T-1}\left\|\boldsymbol{V}_t^{-1}\boldsymbol{g}_t\right\|_{\boldsymbol{H}^*}^2$ and $\mathbb{E}\sum_{t=0}^{T-1}\operatorname{Tr}[\boldsymbol{V}_t^{-2}\boldsymbol{g}_t\boldsymbol{g}_t^\top] = \mathbb{E}\sum_{t=0}^{T-1}\left\|\boldsymbol{V}_t^{-1}\boldsymbol{g}_t\right\|_2^2$, which gives

$$\mathbb{E}\sum_{t=0}^{T-1}\left\|\boldsymbol{V}_t^{-1}\boldsymbol{g}_t\right\|_{\boldsymbol{H}^*}^2 \le \operatorname{Tr}[\boldsymbol{H}^*]\cdot\|\boldsymbol{S}_T\|_{\mathrm{op}} = \|\boldsymbol{S}_T\|_{\mathrm{op}},$$

$$\mathbb{E}\sum_{t=0}^{T-1}\left\|\boldsymbol{V}_t^{-1}\boldsymbol{g}_t\right\|_2^2 \le d\|\boldsymbol{S}_T\|_{\mathrm{op}}$$

where $\boldsymbol{S}_T = \mathbb{E}\sum_{t=0}^{T-1}\boldsymbol{V}_t^{-1}(\boldsymbol{V}_t^2 - \beta\boldsymbol{V}_{t-1}^2)\boldsymbol{V}_t^{-1}$. Based on this, we further define

$$\xi = \frac{2}{\eta}[f(\boldsymbol{x}_0) - \min f(\boldsymbol{x})] + \eta\Lambda_{\mathcal{H}}(f)\|\boldsymbol{S}_T\|_{\mathrm{op}} + \sqrt{2\left\|\boldsymbol{\Sigma}\right\|_{\mathrm{op}}d}\,\|\boldsymbol{S}_T\|_{\mathrm{op}}. \tag{24}$$

Applying these bounds to (23) gives

$$\mathbb{E}\sum_{t=0}^{T-1}\operatorname{Tr}[\tilde{\boldsymbol{V}}_t^{-1}\bar{\boldsymbol{g}}_t\bar{\boldsymbol{g}}_t^\top] \le \xi. \tag{25}$$

Now we can plug (25) into (20), and hence it follows from (18) that

$$\mathbb{E}\left[\frac{1}{T}\sum_{t=0}^{T-1}\|\nabla f(\boldsymbol{x}_t)\|_{\mathcal{H},*}\right] = \frac{1}{T}\mathbb{E}\sum_{t=0}^{T-1}\operatorname{Tr}[P_{\mathcal{H}}(\bar{\boldsymbol{g}}_t\bar{\boldsymbol{g}}_t^\top)]$$

$$\le \frac{\sqrt{\sum_{i=0}^{T-1}\beta^{i/2}}}{T}\xi + T^{-1/2}\operatorname{Tr}\left[P_{\mathcal{H}}\left(\left(\sum_{i=0}^{T-1}\beta^i\right)\boldsymbol{\Sigma} + \epsilon\boldsymbol{I}_d\right)\right]^{\frac{1}{2}}\sqrt{\xi}.$$

It remains to provide a concrete bound for $\|\boldsymbol{S}_T\|_{\mathrm{op}}$, which follows from Lemma 3.3 and Lemma D.6. Specifically, we know from Lemma 3.3 that

$$\|\boldsymbol{S}_T\|_{\mathrm{op}} \le C_1\left(\log\left[\frac{d}{\epsilon}\sum_{t=0}^{T-1}\|\boldsymbol{g}_t\|_2^2 + d^2(1-\beta)T\right] + 1\right)\left(\frac{(1-\beta)T}{\beta} + \log\left\|\frac{\boldsymbol{V}_{T-1}^2}{\epsilon}\right\|_{\mathrm{op}}\right) + C_2. \tag{26}$$

Moreover, by Lemma D.6, we know that

$$\sum_{t=0}^{T-1}\|\boldsymbol{g}_t\|_2^2 = \operatorname{poly}(T, \|\boldsymbol{\Sigma}\|_{\mathrm{op}}, \|\nabla f(\boldsymbol{x}_0)\|_2, \Lambda_{\mathcal{H}}(f), d, \eta)$$

$$\|\boldsymbol{V}_{T-1}\|_{\mathrm{op}} = \operatorname{poly}(T, \|\boldsymbol{\Sigma}\|_{\mathrm{op}}, \|\nabla f(\boldsymbol{x}_0)\|_2, \Lambda_{\mathcal{H}}(f), d, \eta, \epsilon).$$

Then the first term in Eq. 26 satisfies

$$\log\left[\frac{d}{\epsilon}\sum_{t=0}^{T-1}\|\boldsymbol{g}_t\|_2^2 + d^2(1-\beta)T\right] + 1 = 1 + \log\left[\text{poly}(T, \|\boldsymbol{\Sigma}\|_{\text{op}}, \|\nabla f(\boldsymbol{x}_0)\|_2, \Lambda_{\mathcal{H}}(f), d, \eta, 1-\beta, \epsilon^{-1})\right]$$

$$= \tilde{O}(\log d)$$

where $\tilde{O}(\cdot)$ hides logarithm dependence on all problem parameters except $d$. Similarly, the second term in Eq. 26 satisfies

$$\frac{(1-\beta)T}{\beta} + \log\left\|\frac{\boldsymbol{V}_{T-1}^2}{\epsilon}\right\|_{\text{op}} = \frac{(1-\beta)T}{\beta} + \tilde{O}(\log d)$$

Plugging these two bounds into Eq. 26 yields that for any well-structured $\mathcal{H}$,

$$\|\boldsymbol{S}_T\|_{\text{op}} = \tilde{O}\left(\log d \cdot \left(\frac{(1-\beta)T}{\beta} + \log d\right)\right) \tag{27}$$

When $\mathcal{H}$ only has diagonal matrices, we can improve Lemma D.6 with results in Xie et al. (2025a). The dependence on $d$ can be improved such that $\log(\boldsymbol{V}_{T-1}^2/\epsilon) = \tilde{O}(1)$. Then Lemma 3.3 gives us

$$\|\boldsymbol{S}_T\|_{\text{op}} \leq (1-\beta)T + \log\left\|\frac{\boldsymbol{V}_{T-1}^2}{\epsilon}\right\|_{\text{op}} = (1-\beta)T + \tilde{O}(1).$$

This completes the proof. $\square$

**Lemma D.3.** *Under the setting of Theorem D.7, let $\tilde{\boldsymbol{V}}_t$ be as defined in Eq. 16. Then it holds that*

$$\sum_{t=0}^{T-1}\mathbb{E}_t\,\text{Tr}[\boldsymbol{V}_t^{-1}\boldsymbol{g}_t\bar{\boldsymbol{g}}_t^\top] \geq \frac{1}{2}\sum_{t=0}^{T-1}\mathbb{E}_t\,\text{Tr}[\tilde{\boldsymbol{V}}_t^{-1}\bar{\boldsymbol{g}}_t\bar{\boldsymbol{g}}_t^\top] - \frac{\sqrt{2\|\boldsymbol{\Sigma}\|_{\text{op}}}}{2}\sum_{t=0}^{T-1}\mathbb{E}_t\,\text{Tr}[\boldsymbol{V}_t^{-2}\boldsymbol{g}_t\boldsymbol{g}_t^\top].$$

*Proof.* First we can compute the gap by replacing $\boldsymbol{V}_t$ with $\tilde{\boldsymbol{V}}_t$ as follows

$$\left|\text{Tr}\left[(\boldsymbol{V}_t^{-1} - \tilde{\boldsymbol{V}}_t^{-1})\boldsymbol{g}_t\bar{\boldsymbol{g}}_t^\top\right]\right| = \left|\text{Tr}\left[\tilde{\boldsymbol{V}}_t^{-1}(\tilde{\boldsymbol{V}}_t - \boldsymbol{V}_t)\boldsymbol{V}_t^{-1}\boldsymbol{g}_t\bar{\boldsymbol{g}}_t^\top\right]\right|$$

$$\leq \frac{1}{2}\text{Tr}\left[\tilde{\boldsymbol{V}}_t^{-1}\bar{\boldsymbol{g}}_t\bar{\boldsymbol{g}}_t^\top\right] + \frac{1}{2}\text{Tr}\left[(\tilde{\boldsymbol{V}}_t - \boldsymbol{V}_t)\tilde{\boldsymbol{V}}_t^{-1}(\tilde{\boldsymbol{V}}_t - \boldsymbol{V}_t)\boldsymbol{V}_t^{-1}\boldsymbol{g}_t\boldsymbol{g}_t^\top\boldsymbol{V}_t^{-1}\right], \tag{28}$$

where the second inequality follows from the fact that $\text{Tr}[\boldsymbol{A}^\top\boldsymbol{B}] \leq \frac{1}{2}\text{Tr}[\boldsymbol{A}\boldsymbol{A}^\top] + \frac{1}{2}\text{Tr}[\boldsymbol{B}\boldsymbol{B}^\top]$ with $\boldsymbol{A} = \tilde{\boldsymbol{V}}_t^{-\frac{1}{2}}\bar{\boldsymbol{g}}_t$ and $\boldsymbol{B} = \tilde{\boldsymbol{V}}_t^{-\frac{1}{2}}(\tilde{\boldsymbol{V}}_t - \boldsymbol{V}_t)\boldsymbol{V}_t^{-1}\boldsymbol{g}_t$. Note that since $\tilde{\boldsymbol{V}}_t \succeq \boldsymbol{V}_t$, it holds that $0 \preceq \tilde{\boldsymbol{V}}_t - \boldsymbol{V}_t \preceq \tilde{\boldsymbol{V}}_t$. Then we can upper bound the second term on the right-hand side of (28) as

$$\text{Tr}\left[(\tilde{\boldsymbol{V}}_t - \boldsymbol{V}_t)\tilde{\boldsymbol{V}}_t^{-1}(\tilde{\boldsymbol{V}}_t - \boldsymbol{V}_t)\boldsymbol{V}_t^{-1}\boldsymbol{g}_t\boldsymbol{g}_t^\top\boldsymbol{V}_t^{-1}\right] \leq \text{Tr}\left[(\tilde{\boldsymbol{V}}_t - \boldsymbol{V}_t)\boldsymbol{V}_t^{-1}\boldsymbol{g}_t\boldsymbol{g}_t^\top\boldsymbol{V}_t^{-1}\right]$$

$$\leq \|\tilde{\boldsymbol{V}}_t - \boldsymbol{V}_t\|_{\text{op}}\,\text{Tr}[\boldsymbol{V}_t^{-2}\boldsymbol{g}_t\boldsymbol{g}_t^\top]$$

$$= \|P_{\mathcal{H}}(\tilde{\boldsymbol{M}}_t) - P_{\mathcal{H}}(\boldsymbol{M}_t)\|_{\text{op}}\,\text{Tr}[\boldsymbol{V}_t^{-2}\boldsymbol{g}_t\boldsymbol{g}_t^\top]$$

Further applying Lemma B.5 yields

$$\text{Tr}\left[(\tilde{\boldsymbol{V}}_t - \boldsymbol{V}_t)\tilde{\boldsymbol{V}}_t^{-1}(\tilde{\boldsymbol{V}}_t - \boldsymbol{V}_t)\boldsymbol{V}_t^{-1}\boldsymbol{g}_t\boldsymbol{g}_t^\top\boldsymbol{V}_t^{-1}\right] \leq \|\tilde{\boldsymbol{M}}_t - \boldsymbol{M}_t\|_{\text{op}}^{1/2}\,\text{Tr}[\boldsymbol{V}_t^{-2}\boldsymbol{g}_t\boldsymbol{g}_t^\top]$$

$$\leq \sqrt{2\|\boldsymbol{\Sigma}\|_{\text{op}}}\,\text{Tr}[\boldsymbol{V}_t^{-2}\boldsymbol{g}_t\boldsymbol{g}_t^\top] \tag{29}$$

where the second inequality holds because $\boldsymbol{0} \preceq \tilde{\boldsymbol{M}}_t - \boldsymbol{M}_t = \bar{\boldsymbol{g}}_t\bar{\boldsymbol{g}}_t^\top + \boldsymbol{\Sigma} - \boldsymbol{g}_t\boldsymbol{g}_t^\top \preceq 2\boldsymbol{\Sigma}$ based on Assumption D.1. Then plugging (29) back into (28), we get

$$\sum_{t=0}^{T-1}\text{Tr}[\boldsymbol{V}_t^{-1}\boldsymbol{g}_t\bar{\boldsymbol{g}}_t^\top] \geq \sum_{t=0}^{T-1}\text{Tr}[\tilde{\boldsymbol{V}}_t^{-1}\boldsymbol{g}_t\bar{\boldsymbol{g}}_t^\top] - \sum_{t=0}^{T-1}\left|\text{Tr}\left[\left(\boldsymbol{V}_t^{-1} - \tilde{\boldsymbol{V}}_t^{-1}\right)\boldsymbol{g}_t\bar{\boldsymbol{g}}_t^\top\right]\right|$$

$$\geq \frac{1}{2}\sum_{t=0}^{T-1}\text{Tr}[\tilde{\boldsymbol{V}}_t^{-1}\bar{\boldsymbol{g}}_t\bar{\boldsymbol{g}}_t^\top] - \frac{\sqrt{2\|\boldsymbol{\Sigma}\|_{\text{op}}}}{2}\sum_{t=0}^{T-1}\text{Tr}[\boldsymbol{V}_t^{-2}\boldsymbol{g}_t\boldsymbol{g}_t^\top].$$

$\square$

**Lemma D.4.** *Under the setting of Theorem D.7, let $\tilde{V}_t$ be as defined in Eq. 16. Then it holds that*

$$\left( \mathbb{E} \sum_{t=0}^{T-1} \mathrm{Tr}[\tilde{V}_t^{-1} \bar{g}_t \bar{g}_t^\top] \right) \left( \mathbb{E} \sum_{t=0}^{T-1} \mathrm{Tr}[\tilde{V}_t] \right) \geq \left( \mathbb{E} \sum_{t=0}^{T-1} \mathrm{Tr}[P_{\mathcal{H}}(\bar{g}_t \bar{g}_t^\top)] \right)^2.$$

*Proof of Lemma D.4.* For convenience, denote $\boldsymbol{A}_t = P_{\mathcal{H}}(\bar{g}_t \bar{g}_t^\top)$ and $\boldsymbol{B}_t = \tilde{V}_t^{\frac{1}{2}}$. Then

$$\sum_{t=0}^{T-1} \mathrm{Tr}[P_{\mathcal{H}}(\bar{g}_t \bar{g}_t^\top)] = \sum_{t=0}^{T-1} \mathrm{Tr}[\boldsymbol{A}_t] = \sum_{t=0}^{T-1} \langle \boldsymbol{A}_t, \boldsymbol{I} \rangle = \sum_{t=0}^{T-1} \langle \boldsymbol{B}_t^{-1} \boldsymbol{A}_t, \boldsymbol{B}_t \rangle.$$

Now taking expectation on both sides and applying Cauchy-Schwarz inequality, we get

$$\mathbb{E} \sum_{t=0}^{T-1} \mathrm{Tr}[P_{\mathcal{H}}(\bar{g}_t \bar{g}_t^\top)] \leq \sum_{t=0}^{T-1} \mathbb{E}[\|\boldsymbol{B}_t^{-1} \boldsymbol{A}_t\|_F \|\boldsymbol{B}_t\|_F] \leq \sum_{t=0}^{T-1} \left( \mathbb{E}\|\boldsymbol{B}_t^{-1} \boldsymbol{A}_t\|_F^2 \right)^{1/2} \left( \mathbb{E}\|\boldsymbol{B}_t\|_F^2 \right)^{1/2}$$

Applying Cauchy-Schwarz inequality again, we further have

$$\begin{aligned}
\mathbb{E} \sum_{t=0}^{T-1} \mathrm{Tr}[P_{\mathcal{H}}(\bar{g}_t \bar{g}_t^\top)] &\leq \left( \sum_{t=0}^{T-1} \mathbb{E}\|\boldsymbol{B}_t^{-1} \boldsymbol{A}_t\|_F^2 \right)^{1/2} \left( \sum_{t=0}^{T-1} \mathbb{E}\|\boldsymbol{B}_t\|_F^2 \right)^{1/2} \\
&= \left( \mathbb{E} \sum_{t=0}^{T-1} \mathrm{Tr}[\tilde{V}_t^{-1} P_{\mathcal{H}}(\bar{g}_t \bar{g}_t^\top)^2] \right)^{1/2} \left( \mathbb{E} \sum_{t=0}^{T-1} \mathrm{Tr}[\tilde{V}_t] \right)^{1/2} \\
&= \left( \mathbb{E} \sum_{t=0}^{T-1} \mathrm{Tr}[\tilde{V}_t^{-1} \bar{g}_t \bar{g}_t^\top] \right)^{1/2} \left( \mathbb{E} \sum_{t=0}^{T-1} \mathrm{Tr}[\tilde{V}_t] \right)^{1/2}
\end{aligned}$$

where the second equality follows from Lemma B.2 as $\tilde{V}_t \in \mathcal{H}$. This completes the proof.

$\square$

**Lemma D.5.** *Under the setting of Theorem D.7, let $\tilde{V}_t$ be as defined in Eq. 16. Then it holds that*

$$\mathbb{E} \sum_{t=0}^{T-1} \mathrm{Tr}[\tilde{V}_t] \leq T \cdot \mathrm{Tr}\left[ P_{\mathcal{H}}\left( \left( \sum_{i=0}^{T-1} \beta^i \right) \boldsymbol{\Sigma} + \epsilon \boldsymbol{I}_d \right) \right] + \left( \sum_{i=0}^{T-1} \beta^{i/2} \right) \mathbb{E} \sum_{t=0}^{T-1} \mathrm{Tr}[\bar{g}_t \bar{g}_t^\top \tilde{V}_t^{-1}]. \quad (30)$$

*Proof of Lemma D.5.* Recall that $\tilde{V}_t = P_{\mathcal{H}}(\tilde{M}_t + \epsilon \boldsymbol{I}_d) \in \mathcal{H}$. Then applying Lemma B.2, we get

$$\mathrm{Tr}[\tilde{V}_t] = \mathrm{Tr}[\tilde{V}_t^2 \tilde{V}_t^{-1}] = \mathrm{Tr}[(P_{\mathcal{H}}(\tilde{M}_t + \epsilon \boldsymbol{I}_d))^2 \tilde{V}_t^{-1}] = \mathrm{Tr}[(\tilde{M}_t + \epsilon \boldsymbol{I}_d) \tilde{V}_t^{-1}]. \quad (31)$$

Further plugging in $\tilde{M}_t = \beta M_{t-1} + \bar{g}_t \bar{g}_t^\top + \boldsymbol{\Sigma}$, we have

$$\begin{aligned}
\mathrm{Tr}[\tilde{V}_t] &= \mathrm{Tr}[(\bar{g}_t \bar{g}_t^\top + \boldsymbol{\Sigma} + \beta M_{t-1} + \epsilon \boldsymbol{I}_d) \tilde{V}_t^{-1}] \\
&= \mathrm{Tr}[(\boldsymbol{\Sigma} + \beta M_{t-1} + \epsilon \boldsymbol{I}_d) \tilde{V}_t^{-1}] + \mathrm{Tr}[\bar{g}_t \bar{g}_t^\top \tilde{V}_t^{-1}]
\end{aligned}$$

Applying Lemma B.2 again, we further have

$$\begin{aligned}
\mathrm{Tr}[\tilde{V}_t] &= \mathrm{Tr}[P_{\mathcal{H}}(\boldsymbol{\Sigma} + \beta M_{t-1} + \epsilon \boldsymbol{I}_d)^2 \tilde{V}_t^{-1}] + \mathrm{Tr}[\bar{g}_t \bar{g}_t^\top \tilde{V}_t^{-1}] \\
&\leq \mathrm{Tr}[P_{\mathcal{H}}(\boldsymbol{\Sigma} + \beta M_{t-1} + \epsilon \boldsymbol{I}_d)] + \mathrm{Tr}[\bar{g}_t \bar{g}_t^\top \tilde{V}_t^{-1}]
\end{aligned} \quad (32)$$

where the inequality holds because $\tilde{V}_t = P_{\mathcal{H}}(\bar{g}_t \bar{g}_t^\top + \boldsymbol{\Sigma} + \beta M_{t-1} + \epsilon \boldsymbol{I}_d) \succeq P_{\mathcal{H}}(\boldsymbol{\Sigma} + \beta M_{t-1} + \epsilon \boldsymbol{I}_d)$. Next, we will further control the right-hand side of the above inequality by recursive expansion.

For notational convenience, for any $1 \leq s < t$, denote

$$\boldsymbol{A}_s = \left( \sum_{i=0}^{s-1} \beta^i \right) \boldsymbol{\Sigma} + \beta^s M_{t-s} + \epsilon \boldsymbol{I}_d.$$

Then for any $1 \leq s < t$, we have

$$\mathbb{E}\operatorname{Tr}[P_{\mathcal{H}}(\boldsymbol{A}_s)] = \mathbb{E}\operatorname{Tr}\left[P_{\mathcal{H}}\left(\left(\sum_{i=0}^{s-1}\beta^i\right)\boldsymbol{\Sigma} + \beta^{s+1}\boldsymbol{M}_{t-s-1} + \beta^s\boldsymbol{g}_{t-s}\boldsymbol{g}_{t-s}^\top + \epsilon\boldsymbol{I}_d\right)\right]$$

$$\leq \mathbb{E}\operatorname{Tr}\left[P_{\mathcal{H}}\left(\mathbb{E}_{t-s-1}\left(\sum_{i=0}^{s-1}\beta^i\right)\boldsymbol{\Sigma} + \beta^{s+1}\boldsymbol{M}_{t-s-1} + \beta^s\boldsymbol{g}_{t-s}\boldsymbol{g}_{t-s}^\top + \epsilon\boldsymbol{I}_d\right)\right]$$

where the inequality holds because $\operatorname{Tr}[P_{\mathcal{H}}(\boldsymbol{X})]$ is concave in $\boldsymbol{X}$ by Lemma B.6. Then since $\mathbb{E}\boldsymbol{g}_{t-s}\boldsymbol{g}_{t-s}^\top \preceq \bar{\boldsymbol{g}}_{t-s}\bar{\boldsymbol{g}}_{t-s}^\top + \boldsymbol{\Sigma}$ by Assumption D.1, we further have

$$\mathbb{E}\operatorname{Tr}[P_{\mathcal{H}}(\boldsymbol{A}_s)] \leq \mathbb{E}\operatorname{Tr}\left[P_{\mathcal{H}}\left(\left(\sum_{i=0}^{s}\beta^i\right)\boldsymbol{\Sigma} + \beta^{s+1}\boldsymbol{M}_{t-s-1} + \beta^s\bar{\boldsymbol{g}}_{t-s}\bar{\boldsymbol{g}}_{t-s}^\top + \epsilon\boldsymbol{I}_d\right)\right]$$

$$= \mathbb{E}\operatorname{Tr}\left[P_{\mathcal{H}}(\boldsymbol{A}_{s+1} + \beta^s\bar{\boldsymbol{g}}_{t-s}\bar{\boldsymbol{g}}_{t-s}^\top)\right].$$

Applying the same trick as in (31) to $\boldsymbol{A}_{s+1} + \beta^s\bar{\boldsymbol{g}}_{t-s}\bar{\boldsymbol{g}}_{t-s}^\top$, we obtain

$$\mathbb{E}\operatorname{Tr}\left[P_{\mathcal{H}}(\boldsymbol{A}_s)\right] \leq \mathbb{E}\operatorname{Tr}\left[P_{\mathcal{H}}(\boldsymbol{A}_{s+1} + \beta^s\bar{\boldsymbol{g}}_{t-s}\bar{\boldsymbol{g}}_{t-s}^\top)^2 P_{\mathcal{H}}(\boldsymbol{A}_{s+1} + \beta^s\bar{\boldsymbol{g}}_{t-s}\bar{\boldsymbol{g}}_{t-s}^\top)^{-1}\right]$$

$$= \mathbb{E}\operatorname{Tr}\left[(\boldsymbol{A}_{s+1} + \beta^s\bar{\boldsymbol{g}}_{t-s}\bar{\boldsymbol{g}}_{t-s}^\top)P_{\mathcal{H}}(\boldsymbol{A}_{s+1} + \beta^s\bar{\boldsymbol{g}}_{t-s}\bar{\boldsymbol{g}}_{t-s}^\top)^{-1}\right]$$

$$= \mathbb{E}\operatorname{Tr}\left[P_{\mathcal{H}}(\boldsymbol{A}_{s+1})^2 P_{\mathcal{H}}(\boldsymbol{A}_{s+1} + \beta^s\bar{\boldsymbol{g}}_{t-s}\bar{\boldsymbol{g}}_{t-s}^\top)^{-1}\right]$$

$$+ \mathbb{E}\operatorname{Tr}\left[\beta^s\bar{\boldsymbol{g}}_{t-s}\bar{\boldsymbol{g}}_{t-s}^\top P_{\mathcal{H}}(\boldsymbol{A}_{s+1} + \beta^s\bar{\boldsymbol{g}}_{t-s}\bar{\boldsymbol{g}}_{t-s}^\top)^{-1}\right]$$

$$\leq \mathbb{E}\operatorname{Tr}\left[P_{\mathcal{H}}(\boldsymbol{A}_{s+1})\right] + \mathbb{E}\operatorname{Tr}\left[\beta^s\bar{\boldsymbol{g}}_{t-s}\bar{\boldsymbol{g}}_{t-s}^\top P_{\mathcal{H}}(\boldsymbol{A}_{s+1} + \beta^s\bar{\boldsymbol{g}}_{t-s}\bar{\boldsymbol{g}}_{t-s}^\top)^{-1}\right] \quad (33)$$

where the second equality is again by Lemma B.2 and the second inequality follows from the fact that $P_{\mathcal{H}}(\boldsymbol{A}_{s+1} + \beta^s\bar{\boldsymbol{g}}_{t-s}\bar{\boldsymbol{g}}_{t-s}^\top) \succeq P_{\mathcal{H}}(\boldsymbol{A}_{s+1})$. In addition, note that

$$\boldsymbol{A}_{s+1} + \beta^s\bar{\boldsymbol{g}}_{t-s}\bar{\boldsymbol{g}}_{t-s}^\top = \beta^s\left(\frac{\sum_{i=0}^{s}\beta^i}{\beta^s}\boldsymbol{\Sigma} + \beta\boldsymbol{M}_{t-s-1} + \bar{\boldsymbol{g}}_{t-s}\bar{\boldsymbol{g}}_{t-s}^\top\right) + \epsilon\boldsymbol{I}_d \succeq \beta^s(\tilde{\boldsymbol{M}}_{t-s} + \epsilon\boldsymbol{I}_d).$$

Thus it follows from Lemma B.3 that

$$P_{\mathcal{H}}(\boldsymbol{A}_{s+1} + \beta^s\bar{\boldsymbol{g}}_{t-s}\bar{\boldsymbol{g}}_{t-s}^\top) \succeq P_{\mathcal{H}}(\beta^s(\tilde{\boldsymbol{M}}_{t-s} + \epsilon\boldsymbol{I}_d)) = \beta^{s/2}P_{\mathcal{H}}(\tilde{\boldsymbol{M}}_{t-s} + \epsilon\boldsymbol{I}_d) = \beta^{s/2}\tilde{\boldsymbol{V}}_{t-s}. \quad (34)$$

Now combining (33) and (34) yields

$$\mathbb{E}\operatorname{Tr}\left[P_{\mathcal{H}}(\boldsymbol{A}_s)\right] \leq \mathbb{E}\operatorname{Tr}\left[P_{\mathcal{H}}(\boldsymbol{A}_{s+1})\right] + \beta^{s/2}\mathbb{E}\operatorname{Tr}[\bar{\boldsymbol{g}}_{t-s}\bar{\boldsymbol{g}}_{t-s}^\top\tilde{\boldsymbol{V}}_{t-s}^{-1}].$$

Further telescoping over $s = 1, \ldots, t-1$, we obtain

$$\mathbb{E}\operatorname{Tr}\left[P_{\mathcal{H}}(\boldsymbol{\Sigma} + \beta\boldsymbol{M}_{t-1} + \epsilon\boldsymbol{I}_d)\right] = \mathbb{E}\operatorname{Tr}\left[P_{\mathcal{H}}(\boldsymbol{A}_1)\right]$$

$$\leq \mathbb{E}\operatorname{Tr}\left[P_{\mathcal{H}}(\boldsymbol{A}_t)\right] + \sum_{s=1}^{t-1}\beta^{s/2}\mathbb{E}\operatorname{Tr}\left[\bar{\boldsymbol{g}}_{t-s}\bar{\boldsymbol{g}}_{t-s}^\top\tilde{\boldsymbol{V}}_{t-s}^{-1}\right]$$

$$= \operatorname{Tr}\left[P_{\mathcal{H}}\left(\left(\sum_{i=0}^{t-1}\beta^i\right)\boldsymbol{\Sigma} + \epsilon\boldsymbol{I}_d\right)\right] + \sum_{s=1}^{t-1}\beta^{s/2}\mathbb{E}\operatorname{Tr}\left[\bar{\boldsymbol{g}}_{t-s}\bar{\boldsymbol{g}}_{t-s}^\top\tilde{\boldsymbol{V}}_{t-s}^{-1}\right]$$

$$(35)$$

Now, plugging (35) back into (32), we obtain

$$\mathbb{E}\operatorname{Tr}[\tilde{\boldsymbol{V}}_t] \leq \operatorname{Tr}\left[P_{\mathcal{H}}\left(\left(\sum_{i=0}^{t-1}\beta^i\right)\boldsymbol{\Sigma} + \epsilon\boldsymbol{I}_d\right)\right] + \sum_{s=0}^{t-1}\beta^{s/2}\mathbb{E}\operatorname{Tr}\left[\bar{\boldsymbol{g}}_{t-s}\bar{\boldsymbol{g}}_{t-s}^\top\tilde{\boldsymbol{V}}_{t-s}^{-1}\right]$$

$$= \operatorname{Tr}\left[P_{\mathcal{H}}\left(\left(\sum_{i=0}^{t-1}\beta^i\right)\boldsymbol{\Sigma} + \epsilon\boldsymbol{I}_d\right)\right] + \sum_{s=1}^{t}\beta^{(t-s)/2}\mathbb{E}\operatorname{Tr}\left[\bar{\boldsymbol{g}}_s\bar{\boldsymbol{g}}_s^\top\tilde{\boldsymbol{V}}_s^{-1}\right].$$

Finally, summing over $t = 0, 1, \ldots, T-1$, we get

$$\mathbb{E}\sum_{t=0}^{T-1}\operatorname{Tr}[\tilde{\boldsymbol{V}}_t] \leq \sum_{t=0}^{T-1}\operatorname{Tr}\left[P_{\mathcal{H}}\left(\left(\sum_{i=0}^{t-1}\beta^i\right)\boldsymbol{\Sigma} + \epsilon\boldsymbol{I}_d\right)\right] + \sum_{t=0}^{T-1}\sum_{s=1}^{t}\beta^{(t-s)/2}\mathbb{E}\operatorname{Tr}[\bar{\boldsymbol{g}}_s\bar{\boldsymbol{g}}_s^\top\tilde{\boldsymbol{V}}_s^{-1}]$$

$$\leq T \cdot \operatorname{Tr}\left[P_{\mathcal{H}}\left(\left(\sum_{i=0}^{T-1}\beta^i\right)\boldsymbol{\Sigma} + \epsilon\boldsymbol{I}_d\right)\right] + \left(\sum_{i=0}^{T-1}\beta^{i/2}\right)\mathbb{E}\sum_{t=0}^{T-1}\operatorname{Tr}[\bar{\boldsymbol{g}}_t\bar{\boldsymbol{g}}_t^\top\tilde{\boldsymbol{V}}_t^{-1}].$$

This completes the proof. $\qquad\square$

**Lemma D.6.** *Under the setting of Theorem D.2, for any fixed initialization $\boldsymbol{x}_0$, let $\boldsymbol{g}_0, \ldots, \boldsymbol{g}_{T-1}$ and $\boldsymbol{V}_0, \ldots, \boldsymbol{V}_{T-1}$ be given by Algorithm 6. Then the following hold with probability 1:*

$$\sum_{t=0}^{T-1} \|\boldsymbol{g}_t\|_2^2 \leq 2(\|\boldsymbol{\Sigma}\|_{\mathrm{op}} + \|\nabla f(\boldsymbol{x}_0)\|_2^2 + \Lambda_{\mathcal{H}}(f)^2 d\eta^2)T^3,$$

$$\|\boldsymbol{V}_{T-1}\|_{\mathrm{op}} \leq \sqrt{2(\|\boldsymbol{\Sigma}\|_{\mathrm{op}} + \|\nabla f(\boldsymbol{x}_0)\|_2^2 + \Lambda_{\mathcal{H}}(f)^2 d\eta^2)T^3 + \epsilon d}.$$

*Proof.* In this proof, we will define $\boldsymbol{H}^* \in \arg\min_{\boldsymbol{H} \in \mathcal{H}, -\boldsymbol{H} \preceq \nabla^2 f(\boldsymbol{x}) \preceq \boldsymbol{H}} \mathrm{Tr}(\boldsymbol{H})$.

We first control $\sum_{t=0}^{T-1} \|\boldsymbol{g}_t\|_2^2$. According to Assumption D.1, we have

$$\sum_{t=0}^{T-1} \|\boldsymbol{g}_t\|_2^2 = \sum_{t=0}^{T-1} \|\boldsymbol{g}_t \boldsymbol{g}_t^\top\|_{\mathrm{op}} \leq \sum_{t=0}^{T-1} \left( \|\boldsymbol{\Sigma}\|_{\mathrm{op}} + \|\bar{\boldsymbol{g}}_t \bar{\boldsymbol{g}}_t^\top\|_{\mathrm{op}} \right) = T \|\boldsymbol{\Sigma}\|_{\mathrm{op}} + \sum_{t=0}^{T-1} \|\bar{\boldsymbol{g}}_t\|_2^2 \qquad (36)$$

So it suffices to control the sum of $\|\bar{\boldsymbol{g}}_t\|_2^2$. By triangle inequality, $\|\bar{\boldsymbol{g}}_t\|_2^2 \leq 2\|\bar{\boldsymbol{g}}_0\|_2^2 + 2\|\bar{\boldsymbol{g}}_t - \bar{\boldsymbol{g}}_0\|_2^2$, and thus we only need to bound the distance $\bar{\boldsymbol{g}}_t - \bar{\boldsymbol{g}}_0$. To this end, since $-\boldsymbol{H}^* \preceq \nabla^2 f(\boldsymbol{x}) \preceq \boldsymbol{H}^*$ for all $\boldsymbol{x}$, we have

$$\|\bar{\boldsymbol{g}}_t - \bar{\boldsymbol{g}}_0\|_2 = \|\nabla f(\boldsymbol{x}_t) - \nabla f(\boldsymbol{x}_0)\|_2 \leq \|\boldsymbol{H}^*\|_{\mathrm{op}} \|\boldsymbol{x}_t - \boldsymbol{x}_0\|_2 \leq \Lambda_{\mathcal{H}}(f) \sum_{s=0}^{t-1} \|\boldsymbol{x}_{s+1} - \boldsymbol{x}_s\|_2. \quad (37)$$

Moreover, we can control each $\|\boldsymbol{x}_{s+1} - \boldsymbol{x}_s\|_2$ as follows:

$$\begin{aligned}
\|\boldsymbol{x}_{s+1} - \boldsymbol{x}_s\|_2^2 &= \eta^2 \|\boldsymbol{V}_s^{-1} \boldsymbol{g}_s\|_2^2 = \eta^2 \mathrm{Tr}(\boldsymbol{g}_s^\top \boldsymbol{V}_s^{-2} \boldsymbol{g}_s) \\
&= \eta^2 \mathrm{Tr}(P_{\mathcal{H}}(\boldsymbol{M}_s)^{-2} \boldsymbol{g}_s \boldsymbol{g}_s^\top) \\
&= \eta^2 \mathrm{Tr}(P_{\mathcal{H}}(\boldsymbol{M}_s)^{-2} P_{\mathcal{H}}(\boldsymbol{g}_s \boldsymbol{g}_s^\top)^2)
\end{aligned}$$

where the last equality holds by Lemma B.2. Then since $\boldsymbol{M}_s \succeq \boldsymbol{g}_s \boldsymbol{g}_s^\top$, we have $P_{\mathcal{H}}(\boldsymbol{M}_s) \succeq P_{\mathcal{H}}(\boldsymbol{g}_s \boldsymbol{g}_s^\top)$ by Lemma B.3, and it follows that

$$\|\boldsymbol{x}_{s+1} - \boldsymbol{x}_s\|_2^2 \leq \eta^2 \mathrm{Tr}(\boldsymbol{I}_d) = d\eta^2.$$

Plugging this back into Eq. 37, we get $\|\bar{\boldsymbol{g}}_t - \bar{\boldsymbol{g}}_0\|_2 \leq \eta t \Lambda_{\mathcal{H}}(f)\sqrt{d}$, so $\|\bar{\boldsymbol{g}}_t\|_2^2 \leq 2\|\bar{\boldsymbol{g}}_0\|_2^2 + 2\eta^2 t^2 \Lambda_{\mathcal{H}}(f)^2 d$. Now applying this to Eq. 36, we obtain

$$\begin{aligned}
\sum_{t=0}^{T-1} \|\boldsymbol{g}_t\|_2^2 &\leq T \|\boldsymbol{\Sigma}\|_{\mathrm{op}} + \sum_{t=0}^{T-1} (2\|\bar{\boldsymbol{g}}_0\|_2^2 + 2\eta^2 t^2 \Lambda_{\mathcal{H}}(f)^2 d) \\
&\leq 2(\|\boldsymbol{\Sigma}\|_{\mathrm{op}} + \|\bar{\boldsymbol{g}}_0\|_2^2 + \Lambda_{\mathcal{H}}(f)^2 d\eta^2)T^3. \qquad (38)
\end{aligned}$$

Finally, we can bound $\|V_{T-1}\|_{\mathrm{op}}$ as follows:

$$\|\boldsymbol{V}_{T-1}\|_{\mathrm{op}}^2 = \|\boldsymbol{V}_{T-1}^2\|_{\mathrm{op}} = \|P_{\mathcal{H}}(\boldsymbol{M}_{T-1})^2\|_{\mathrm{op}} \leq \mathrm{Tr}(P_{\mathcal{H}}(\boldsymbol{M}_{T-1})^2) = \mathrm{Tr}(\boldsymbol{M}_{T-1})$$

where we apply Lemma B.2 in the last equality. Note that $\mathrm{Tr}(\boldsymbol{M}_{T-1}) = \sum_{t=0}^{T-1} \|\boldsymbol{g}_t\|_2^2 + \epsilon d$, so it follows from (38) that

$$\|\boldsymbol{V}_{T-1}\|_{\mathrm{op}} \leq \sqrt{2(\|\boldsymbol{\Sigma}\|_{\mathrm{op}} + \|\bar{\boldsymbol{g}}_0\|_2^2 + \Lambda_{\mathcal{H}}(f)^2 d\eta^2)T^3 + \epsilon d}.$$

This completes the proof. $\qquad\square$

### D.3  PROOF FOR THE CUMULATIVE AND EMA VARIANTS

We will derive the convergence rate of cumulative optimizers and EMA optimizers by reducting from Theorem D.2. In the stochastic case, the dependence on $T$ of both convergence rate is $\tilde{O}(T^{-1/4})$, matching previous results on specific examples AdaGrad and Adam (Xie et al., 2025a; Li et al., 2025).

**Theorem D.7.** *For any $\epsilon \geq 0$, $\eta > 0$, and $T \in \mathbb{N}$, let $\{\boldsymbol{x}_t\}_{t=0}^{T}$ be the iterates of Algorithm 1 with well-structured preconditioner set $\mathcal{H}$, where the update of $\boldsymbol{M}_t$ follows the cumulative version, i.e., $\boldsymbol{M}_t = \boldsymbol{M}_{t-1} + \boldsymbol{g}_t\boldsymbol{g}_t^\top$ for all $t \in [T]$. Let $\Lambda_\mathcal{H}(f)$ be the adaptive smoothness of the loss $f$ according to Definition 2.4. Then under Assumption D.1, it holds that*

$$\mathbb{E}\left[\frac{1}{T}\sum_{t=0}^{T-1}\|\nabla f(\boldsymbol{x}_t)\|_{\mathcal{H},*}\right] \leq \frac{1}{\sqrt{T}}\xi + \frac{1}{T^{1/4}}\operatorname{Tr}[P_\mathcal{H}(\boldsymbol{\Sigma} + \frac{\epsilon}{T}\boldsymbol{I}_d)]^{\frac{1}{2}}\sqrt{\xi}$$

*where $\xi$ is given by*

$$\xi = \tilde{O}\left(\frac{\Delta_0}{\eta} + \left(\eta \cdot \Lambda_\mathcal{H}(f) + d\,\|\boldsymbol{\Sigma}\|_{\mathrm{op}}^{1/2}\right)\log^2 d\right).$$

*Moreover, when setting the learning rate $\eta = \sqrt{\frac{\Delta_0}{\Lambda_\mathcal{H}(f)\log^2 d}}$, it holds that $\xi = \tilde{O}\left(\sqrt{\Delta_0 \cdot \Lambda_\mathcal{H}(f)}\log d + d\,\|\boldsymbol{\Sigma}\|_{\mathrm{op}}^{1/2}\log^2 d\right)$.*

*Proof of Theorem D.7.* The desired result directly follows from Theorem D.2 with $\beta = 1$, where we additionally apply the bound $\operatorname{Tr}[P_\mathcal{H}(T\boldsymbol{\Sigma} + \epsilon\boldsymbol{I}_d)] \leq \sqrt{T}\operatorname{Tr}[P_\mathcal{H}(\boldsymbol{\Sigma} + \frac{\epsilon}{T}\boldsymbol{I}_d)]$ using Lemma B.3. $\square$

**Theorem D.8.** *For any $\epsilon \geq 0$, $\beta \in (0,1)$, $\eta > 0$, and $T \in \mathbb{N}$, let $\{\boldsymbol{x}_t\}_{t=0}^{T}$ be the iterates of Algorithm 1 with well-structured preconditioner set $\mathcal{H}$, where the update of $\boldsymbol{M}_t$ follows the EMA version, i.e., $\boldsymbol{M}_t = \beta\boldsymbol{M}_{t-1} + (1-\beta)\boldsymbol{g}_t\boldsymbol{g}_t^\top$ for all $t \in [T]$. Let $\Lambda_\mathcal{H}(f)$ be the adaptive smoothness of the loss $f$ according to Definition 2.4. Then under Assumption D.1, it holds that*

$$\mathbb{E}\left[\frac{1}{T}\sum_{t=0}^{T-1}\|\nabla f(\boldsymbol{x}_t)\|_{\mathcal{H},*}\right] \leq \frac{1}{\sqrt{T}}\kappa + \frac{1}{T^{1/4}}\operatorname{Tr}[P_\mathcal{H}(\boldsymbol{\Sigma} + \epsilon\boldsymbol{I}_d)]^{\frac{1}{2}}\sqrt{\kappa}$$

*where $\kappa$ is given by*

$$\kappa = \tilde{O}\left(\frac{\Delta_0}{\eta\sqrt{T}} + \frac{\eta \cdot \Lambda_\mathcal{H}(f) + d\sqrt{1-\beta}\|\boldsymbol{\Sigma}\|_{\mathrm{op}}^{1/2}}{\sqrt{T}}\left(\frac{1}{\beta}T + \frac{\log d}{1-\beta}\right)\log d\right).$$

*Moreover, when setting $1-\beta = \Theta(\frac{\log d}{T})$ and $\eta = \sqrt{\frac{\Delta_0}{\Lambda_\mathcal{H}(f)\cdot T\log^2 d}}$, it holds that*

$$\kappa = \tilde{O}\left(\left(\sqrt{\Delta_0 \cdot \Lambda_\mathcal{H}(f)} + d\,\|\boldsymbol{\Sigma}\|_{\mathrm{op}}^{1/2}\right)\log d\right).$$

*When there is no noise, i.e., $f_t \equiv f$, it holds that*

$$\mathbb{E}\left[\frac{1}{T}\sum_{t=0}^{T-1}\|\nabla f(\boldsymbol{x}_t)\|_{\mathcal{H},*}\right] \leq \frac{1}{\sqrt{T}}\kappa + \frac{\sqrt{d}\epsilon^{1/4}}{T^{1/4}}\sqrt{\kappa}$$

*where $\kappa$ is given by*

$$\kappa = \tilde{O}\left(\frac{\Delta_0}{\eta\sqrt{T}} + \frac{\eta \cdot \Lambda_\mathcal{H}(f)}{\sqrt{T}}\left(\frac{1}{\beta}T + \frac{\log d}{1-\beta}\right)\log d\right).$$

*Moreover, when setting $1-\beta = \Theta(\frac{\log d}{T})$ and $\eta = \sqrt{\frac{\Delta_0}{\Lambda_\mathcal{H}(f)\cdot T\log^2 d}}$, it holds that*

$$\kappa = \tilde{O}\left(\sqrt{\Delta_0 \cdot \Lambda_\mathcal{H}(f)}\log d\right).$$

*Proof of Theorem D.8.* As mentioned in Section 3, Algorithm 5 is equivalent to Algorithm 6 with $\epsilon/(1-\beta)$ in place of $\epsilon$ and $\eta/\sqrt{1-\beta}$ in place of $\eta$. Therefore, the convergence rate of Algorithm 5 follows from Theorem D.2 with this change of hyperparameters. Specifically,

$$\frac{1}{T}\mathbb{E}\sum_{t=0}^{T-1}\|\bar{\boldsymbol{g}}_t\|_{\mathcal{H},*} \leq \frac{\sqrt{\sum_{i=0}^{T-1}\beta^{i/2}}}{T}\xi + \frac{\sqrt{\operatorname{Tr}[P_\mathcal{H}((\sum_{i=0}^{T-1}\beta^i)\boldsymbol{\Sigma} + \frac{\epsilon}{1-\beta}\boldsymbol{I}_d)]}}{\sqrt{T}}\sqrt{\xi} \qquad (39)$$

where $\xi$ satisfies

$$\xi = \tilde{O}\left(\frac{\Delta_0\sqrt{1-\beta}}{\eta} + \left(\frac{\eta}{\sqrt{1-\beta}}\Lambda_{\mathcal{H}}(f) + d\|\mathbf{\Sigma}\|_{\mathrm{op}}^{1/2}\right)\left(\frac{1-\beta}{\beta}T + \log d\right)\log d\right). \tag{40}$$

We can further simplify the result for $\beta < 1$ as follows. Note that $\sum_{i=0}^{T-1}\beta^i \leq 1/(1-\beta)$ and

$$\sum_{i=0}^{T-1}\beta^{i/2} \leq \frac{1}{1-\sqrt{\beta}} = \frac{1+\sqrt{\beta}}{1-\beta} \leq \frac{2}{1-\beta}.$$

It then follows from Lemma B.3 that

$$\mathrm{Tr}\left[P_{\mathcal{H}}\left(\left(\sum_{i=0}^{T-1}\beta^i\right)\mathbf{\Sigma} + \frac{\epsilon}{1-\beta}\mathbf{I}_d\right)\right] \leq \mathrm{Tr}\left[P_{\mathcal{H}}\left(\frac{1}{1-\beta}(\mathbf{\Sigma} + \epsilon\mathbf{I}_d)\right)\right] = \frac{1}{\sqrt{1-\beta}}\mathrm{Tr}[P_{\mathcal{H}}(\mathbf{\Sigma} + \epsilon\mathbf{I}_d)].$$

Applying these to (39), we get

$$\mathbb{E}\frac{1}{T}\sum_{t=0}^{T-1}\|\bar{\boldsymbol{g}}_t\|_{\mathcal{H},*} \leq \frac{\sqrt{2}}{T}\frac{\xi}{\sqrt{1-\beta}} + \frac{\sqrt{\mathrm{Tr}[P_{\mathcal{H}}(\mathbf{\Sigma} + \epsilon\mathbf{I}_d)]}}{\sqrt{T}}\sqrt{\frac{\xi}{\sqrt{1-\beta}}}.$$

Denote $\kappa = \xi/\sqrt{(1-\beta)T}$. Then we have

$$\mathbb{E}\frac{1}{T}\sum_{t=0}^{T-1}\|\bar{\boldsymbol{g}}_t\|_{\mathcal{H},*} \leq \frac{\sqrt{2}}{\sqrt{T}}\kappa + \frac{\sqrt{\mathrm{Tr}[P_{\mathcal{H}}(\mathbf{\Sigma} + \epsilon\mathbf{I}_d)]}}{T^{1/4}}\sqrt{\kappa}.$$

By (40), we know that $\kappa$ satisfies that

$$\kappa = \tilde{O}\left(\frac{\Delta_0}{\eta\sqrt{T}} + \frac{\eta\Lambda_{\mathcal{H}}(f) + d\sqrt{1-\beta}\|\mathbf{\Sigma}\|_{\mathrm{op}}^{1/2}}{\sqrt{T}}\left(\frac{1}{\beta}T + \frac{\log d}{1-\beta}\right)\log d\right).$$

This completes the proof. $\qquad\square$

# E PROOF FOR THE ACCELERATED ALGORITHM

In this section, we will define $\boldsymbol{H}^* = \arg\min_{\boldsymbol{H}\in\mathcal{H},\mathrm{Tr}(\boldsymbol{H})\leq 1}L_{\|\cdot\|_{\boldsymbol{H}}}(f)$. Our proof follows the strategy developed by Kovalev (2025a). For completeness, we reproduce some of the arguments to make the exposition self-contained.

## E.1 STOCHASTIC CASE WITHOUT PROJECTION

**Theorem 4.3.** *For a well-structured preconditioner set $\mathcal{H}$, let $f$ be a convex loss function whose $\mathcal{H}$-smoothness constant is $\Lambda_{\mathcal{H}}(f) \in (0,\infty)$ according to Definition 2.4. For $\epsilon > 0$, $T > 0$, consider Algorithm 2 with $\alpha_t = 2/(t+2)$ for $t = 0, 1, \ldots, T-1$. Suppose $\boldsymbol{x}^*$ is the global minima and $\max_{t=0,1,\ldots,T-1}\|\boldsymbol{x}_t - \boldsymbol{x}^*\|_{\mathcal{H}} \leq D$ for some $D > 0$ and Assumption 4.2 holds with adaptive gradient variance $\sigma_{\mathcal{H}}(\{f_t\}_{t=1}^T)^2 \leq \sigma_{\mathcal{H}}^2$ for some $\sigma_{\mathcal{H}} \in [0,\infty)$. Then it holds that*

$$\mathbb{E}[f(\bar{\boldsymbol{x}}_T) - f(\boldsymbol{x}^*)] \leq \frac{2D^2\epsilon}{\eta(T+1)^2}\mathbb{E}\,\mathrm{Tr}(\boldsymbol{V}_{T-1}^{-1}) + \left(\frac{D^2}{2\eta} - \frac{\eta}{2}\right)\mathbb{E}\frac{4}{(T+1)^2}\sum_{t=0}^{T-1}\boldsymbol{g}_t^\top\boldsymbol{V}_t^{-1}\boldsymbol{g}_t$$

$$+ \frac{2\eta^2}{(T+1)^2}\cdot\Lambda_{\mathcal{H}}(f)\cdot\tilde{O}(\log^2 d) + \frac{\eta}{T^{1/2}}\sigma_{\mathcal{H}}\cdot\tilde{O}(\log d).$$

*Moreover, when choosing learning rate $\eta = D$, the convergence rate becomes*

$$\mathbb{E}[f(\bar{\boldsymbol{x}}_T) - f(\boldsymbol{x}^*)] = \tilde{O}\left(\frac{\Lambda_{\mathcal{H}}(f)D^2\log^2 d + d\sqrt{\epsilon}D}{T^2} + \frac{\sigma_{\mathcal{H}}D\log d}{\sqrt{T}}\right).$$

*Proof of Theorem 4.3.* Combining the results of Lemma E.2 and Lemma E.3, we get

$$\sum_{t=0}^{T-1} \mathbb{E}[f^{\alpha_t,\bar{\boldsymbol{x}}_t}(\boldsymbol{x}_{t+1}) - f^{\alpha_t,\bar{\boldsymbol{x}}_t}(\boldsymbol{x}^*)] \le \mathbb{E}\left[\frac{D^2\epsilon}{2\eta}\operatorname{Tr}(\boldsymbol{V}_{T-1}^{-1}) + \left(\frac{D^2}{2\eta} - \frac{\eta}{2}\right)\sum_{t=0}^{T-1}\boldsymbol{g}_t^\top \boldsymbol{V}_t^{-1}\boldsymbol{g}_t\right]$$

$$+ \mathbb{E}\left[\eta\sum_{t=0}^{T-1}\langle \boldsymbol{g}_t - \nabla f^{\alpha_t,\bar{\boldsymbol{x}}_t}(\boldsymbol{x}_t), \boldsymbol{V}_t^{-1}\boldsymbol{g}_t\rangle\right]$$

$$+ \mathbb{E}\left[\frac{\Lambda_{\mathcal{H}}(f)\eta^2}{2}\sum_{t=0}^{T-1}\|\boldsymbol{V}_t^{-1}\boldsymbol{g}_t\|_{\boldsymbol{H}^*}^2\right]. \tag{41}$$

It remains to further bound the last two terms.

For the last term on the right-hand side of (41), we apply Lemma 3.3 to get

$$\sum_{t=0}^{T-1}\|\boldsymbol{V}_t^{-1}\boldsymbol{g}_t\|_{\boldsymbol{H}^*}^2 \le \operatorname{Tr}[\boldsymbol{H}^*]\cdot\tilde{O}(\log^2 d) = \Lambda_{\mathcal{H}}(f)\cdot\tilde{O}(\log^2 d) \tag{42}$$

where the second step follows from the definition of $\boldsymbol{H}^*$.

For the third term on the right-hand side of (41), consider any $\boldsymbol{H} \in \mathcal{H}$ with $\operatorname{Tr}(\boldsymbol{H}) \le 1$, and then it follows from the Cauchy-Schwarz inequality that

$$\mathbb{E}\left|\sum_{t=0}^{T-1}\langle \boldsymbol{g}_t - \nabla f^{\alpha_t,\bar{\boldsymbol{x}}_t}(\boldsymbol{x}_t), \boldsymbol{V}_t^{-1}\boldsymbol{g}_t\rangle\right| \le \sum_{t=0}^{T-1}\left(\mathbb{E}\|\boldsymbol{g}_t - \nabla f^{\alpha_t,\bar{\boldsymbol{x}}_t}(\boldsymbol{x}_t)\|_2^2\right)^{1/2}\left(\mathbb{E}\|\boldsymbol{V}_t^{-1}\boldsymbol{g}_t\|_2^2\right)^{1/2}$$

$$\le \left(\mathbb{E}\sum_{t=0}^{T-1}\left\|\boldsymbol{g}_t - \nabla f^{\alpha_t,\bar{\boldsymbol{x}}_t}(\boldsymbol{x}_t)\right\|_{\boldsymbol{H}^{-1}}^2\right)^{1/2}\left(\mathbb{E}\sum_{t=0}^{T-1}\|\boldsymbol{V}_t^{-1}\boldsymbol{g}_t\|_{\boldsymbol{H}}^2\right)^{1/2}.$$

Here we similarly have $\mathbb{E}\sum_{t=0}^{T-1}\|\boldsymbol{V}_t^{-1}\boldsymbol{g}_t\|_{\boldsymbol{H}}^2 = \operatorname{Tr}[\boldsymbol{H}]\cdot\tilde{O}(\log^2 d) \le \tilde{O}(\log^2 d)$ according to Lemma 3.3. Then we further minimize over $\boldsymbol{H} \in \mathcal{H}$ with $\operatorname{Tr}[\boldsymbol{H}] \le 1$ to get

$$\min_{\boldsymbol{H}\in\mathcal{H},\operatorname{Tr}(\boldsymbol{H})\le 1}\mathbb{E}\sum_{t=0}^{T-1}\left\|\boldsymbol{g}_t - \nabla f^{\alpha_t,\bar{\boldsymbol{x}}_t}(\boldsymbol{x}_t)\right\|_{\boldsymbol{H}^{-1}}^2$$

$$= \min_{\boldsymbol{H}\in\mathcal{H},\operatorname{Tr}(\boldsymbol{H})\le 1}\mathbb{E}\sum_{t=0}^{T-1}\frac{1}{\alpha_t^2}\|\nabla f_t(\alpha_t\boldsymbol{x}_t + (1-\alpha_t)\bar{\boldsymbol{x}}_t) - \nabla f(\alpha_t\boldsymbol{x}_t + (1-\alpha_t)\bar{\boldsymbol{x}}_t)\|_{\boldsymbol{H}^{-1}}^2$$

$$\le \sigma_{\mathcal{H}}^2\sum_{t=0}^{T-1}\frac{1}{\alpha_t^2}$$

where the last equation follows from Assumption 4.2 and the condition on $\sigma_{\mathcal{H}}$. Therefore, we obtain

$$\mathbb{E}\left|\sum_{t=0}^{T-1}\langle \boldsymbol{g}_t - \nabla f^{\alpha_t,\bar{\boldsymbol{x}}_t}(\boldsymbol{x}_t), \boldsymbol{V}_t^{-1}\boldsymbol{g}_t\rangle\right| \le \sigma_{\mathcal{H}}\Big(\sum_{t=0}^{T-1}\alpha_t^{-2}\Big)^{1/2}\cdot\tilde{O}(\log^2 d) \le \sigma_{\mathcal{H}}T^{3/2}\cdot\tilde{O}(\log^2 d) \tag{43}$$

where we have plugged in the choice of $\alpha_t = 2/(t+2)$.

Finally, combining (41), (42) and (43), we obtain

$$\mathbb{E}\sum_{t=0}^{T-1}(f^{\alpha_t,\bar{\boldsymbol{x}}_t}(\boldsymbol{x}_{t+1}) - f^{\alpha_t,\bar{\boldsymbol{x}}_t}(\boldsymbol{x}^*)) \le \frac{D^2\epsilon}{2\eta}\mathbb{E}\operatorname{Tr}(\boldsymbol{V}_{T-1}^{-1}) + \left(\frac{D^2}{2\eta} - \frac{\eta}{2}\right)\mathbb{E}\sum_{t=0}^{T-1}\boldsymbol{g}_t^\top \boldsymbol{V}_t^{-1}\boldsymbol{g}_t$$

$$+ \frac{\eta^2}{2}\Lambda_{\mathcal{H}}(f)\cdot\tilde{O}(\log^2 d) + \eta\sigma_{\mathcal{H}}T^{3/2}\cdot\tilde{O}(\log d)$$

The proof is then completed by further applying Lemma E.1. $\qquad\square$

**Lemma E.1.** *Under the setting of Theorem 4.3, it holds that*

$$\mathbb{E}[f(\bar{\boldsymbol{x}}_T) - f(\boldsymbol{x}^*)] \leq \frac{4}{(T+1)^2} \sum_{t=0}^{T-1} \mathbb{E}\left[f^{\alpha_t,\bar{\boldsymbol{x}}_t}(\boldsymbol{x}_{t+1}) - f^{\alpha_t,\bar{\boldsymbol{x}}_t}(\boldsymbol{x}^*)\right] \tag{44}$$

*Proof of Lemma E.1.* Plugging in the definition of $f^{\alpha_t,\bar{\boldsymbol{x}}_t}$ in (8), we have

$$\sum_{t=0}^{T-1} \mathbb{E}\left[f^{\alpha_t,\bar{\boldsymbol{x}}_t}(\boldsymbol{x}^*) - f^{\alpha_t,\bar{\boldsymbol{x}}_t}(\boldsymbol{x}_{t+1})\right] = \sum_{t=0}^{T} \alpha_t^{-2} \mathbb{E}[f(\alpha_t \boldsymbol{x}^* + (1-\alpha_t)\bar{\boldsymbol{x}}_t) - f(\alpha_t \boldsymbol{x}_{t+1} + (1-\alpha_t)\bar{\boldsymbol{x}}_t)]$$

$$\leq \sum_{t=0}^{T-1} \alpha_t^{-2} \mathbb{E}[\alpha_t f(\boldsymbol{x}^*) + (1-\alpha_t)f(\bar{\boldsymbol{x}}_t) - f(\bar{\boldsymbol{x}}_{t+1})]$$

$$= \sum_{t=0}^{T-1} \alpha_t^{-2} \mathbb{E}[(1-\alpha_t)(f(\bar{\boldsymbol{x}}_t) - f(\boldsymbol{x}^*)) - (f(\bar{\boldsymbol{x}}_{t+1}) - f(\boldsymbol{x}^*))]$$

$$= \sum_{t=1}^{T-1} (\alpha_t^{-2}(1-\alpha_t) - \alpha_{t-1}^{-2}) \mathbb{E}[f(\bar{\boldsymbol{x}}_t) - f(\boldsymbol{x}^*)]$$

$$+ \alpha_0^{-2}(1-\alpha_0) \mathbb{E}[f(\bar{\boldsymbol{x}}_0) - f(\boldsymbol{x}^*)] - \alpha_{T-1}^{-2} \mathbb{E}[f(\bar{\boldsymbol{x}}_T) - f(\boldsymbol{x}^*)]$$

where the inequality follows from the convexity of $f$. Plugging in the choice of $\alpha_t = 2/(t+2)$, we further obtain

$$\sum_{t=0}^{T-1} \mathbb{E}\left[f^{\alpha_t,\bar{\boldsymbol{x}}_t}(\boldsymbol{x}^*) - f^{\alpha_t,\bar{\boldsymbol{x}}_t}(\boldsymbol{x}_{t+1})\right] = \frac{(T+1)^2}{4} \mathbb{E}[f(\boldsymbol{x}^*) - f(\bar{\boldsymbol{x}}_T)] - \frac{1}{4} \sum_{t=1}^{T-1} \mathbb{E}[f(\bar{\boldsymbol{x}}_t) - f(\boldsymbol{x}^*)]$$

$$\leq \frac{(T+1)^2}{4} \mathbb{E}[f(\boldsymbol{x}^*) - f(\bar{\boldsymbol{x}}_T)].$$

This completes the proof. □

**Lemma E.2.** *Under the setting of Theorem 4.3, it holds that*

$$\sum_{t=0}^{T-1} \mathbb{E}\left[f^{\alpha_t,\bar{\boldsymbol{x}}_t}(\boldsymbol{x}_t) - f^{\alpha_t,\bar{\boldsymbol{x}}_t}(\boldsymbol{x}^*)\right] \leq \mathbb{E}\left[\frac{D^2\epsilon}{2\eta} \operatorname{Tr}(\boldsymbol{V}_{T-1}^{-1}) + \left(\frac{D^2}{2\eta} + \frac{\eta}{2}\right) \sum_{t=0}^{T-1} \boldsymbol{g}_t^\top \boldsymbol{V}_t^{-1} \boldsymbol{g}_t\right].$$

*Proof of Lemma E.2.* For every $t = 0, 1, \ldots, T-1$, by the convexity of $f^{\alpha_t,\bar{\boldsymbol{x}}_t}$, it follows from the Taylor expansion of $f^{\alpha_t,\bar{\boldsymbol{x}}_t}$ at $\boldsymbol{x}_t$ that

$$f^{\alpha_t,\bar{\boldsymbol{x}}_t}(\boldsymbol{x}_t) - f^{\alpha_t,\bar{\boldsymbol{x}}_t}(\boldsymbol{x}^*) \leq \langle \nabla f^{\alpha_t,\bar{\boldsymbol{x}}_t}(\boldsymbol{x}_t), \boldsymbol{x}_t - \boldsymbol{x}^* \rangle = \langle \boldsymbol{g}_t, \boldsymbol{x}_t - \boldsymbol{x}^* \rangle + \langle \nabla f^{\alpha_t,\bar{\boldsymbol{x}}_t}(\boldsymbol{x}_t) - \boldsymbol{g}_t, \boldsymbol{x}_t - \boldsymbol{x}^* \rangle$$

Since $\boldsymbol{g}_t$ is an unbiased estimate of $\nabla f^{\alpha_t,\bar{\boldsymbol{x}}_t}(\boldsymbol{x}_t)$, taking expectation on both sides yields

$$\mathbb{E}\left[f^{\alpha_t,\bar{\boldsymbol{x}}_t}(\boldsymbol{x}_t) - f^{\alpha_t,\bar{\boldsymbol{x}}_t}(\boldsymbol{x}^*)\right] \leq \mathbb{E}[\langle \boldsymbol{g}_t, \boldsymbol{x}_t - \boldsymbol{x}^* \rangle]$$

By definition $\boldsymbol{x}_{t+1} = \boldsymbol{x}_t - \eta \boldsymbol{V}_t^{-1} \boldsymbol{g}_t$, so it follows from Lemma E.4 that

$$\langle \boldsymbol{g}_t, \boldsymbol{x}_t - \boldsymbol{x}^* \rangle \leq \frac{1}{2\eta} \|\boldsymbol{x}_t - \boldsymbol{x}^*\|_{\boldsymbol{V}_t}^2 - \frac{1}{2\eta} \|\boldsymbol{x}_{t+1} - \boldsymbol{x}^*\|_{\boldsymbol{V}_t}^2 + \frac{\eta}{2} \|\boldsymbol{g}_t\|_{\boldsymbol{V}_t^{-1}}^2$$

Define $\boldsymbol{V}_{-1} = 0$ for convenience. Then combining the above two equations yields

$$\mathbb{E}\left[f^{\alpha_t,\bar{\boldsymbol{x}}_t}(\boldsymbol{x}_t) - f^{\alpha_t,\bar{\boldsymbol{x}}_t}(\boldsymbol{x}^*)\right] \leq \mathbb{E}\left[\frac{1}{2\eta} \|\boldsymbol{x}_t - \boldsymbol{x}^*\|_{\boldsymbol{V}_t}^2 - \frac{1}{2\eta} \|\boldsymbol{x}_{t+1} - \boldsymbol{x}^*\|_{\boldsymbol{V}_t}^2 + \frac{\eta}{2} \|\boldsymbol{V}_t^{-1}\boldsymbol{g}_t\|_{\boldsymbol{V}_t}^2\right]$$

$$= \mathbb{E}\left[\frac{1}{2\eta} \|\boldsymbol{x}_t - \boldsymbol{x}^*\|_{\boldsymbol{V}_{t-1}}^2 - \frac{1}{2\eta} \|\boldsymbol{x}_{t+1} - \boldsymbol{x}^*\|_{\boldsymbol{V}_t}^2\right]$$

$$+ \mathbb{E}\left[\frac{1}{2\eta} \|\boldsymbol{x}_t - \boldsymbol{x}^*\|_{\boldsymbol{V}_t - \boldsymbol{V}_{t-1}}^2 + \frac{\eta}{2} \|\boldsymbol{V}_t^{-1}\boldsymbol{g}_t\|_{\boldsymbol{V}_t}^2\right].$$

Summing over $t$, we obtain

$$\sum_{t=0}^{T-1} \mathbb{E}\left[f^{\alpha_t,\bar{\boldsymbol{x}}_t}(\boldsymbol{x}_t) - f^{\alpha_t,\bar{\boldsymbol{x}}_t}(\boldsymbol{x}^*)\right] \leq \mathbb{E}\left[-\frac{1}{2\eta}\|\boldsymbol{x}_T - \boldsymbol{x}^*\|_{\boldsymbol{V}_{T-1}}^2 + \frac{1}{2\eta}\sum_{t=0}^{T-1}\|\boldsymbol{x}_t - \boldsymbol{x}^*\|_{\boldsymbol{V}_t - \boldsymbol{V}_{t-1}}^2\right]$$
$$+ \mathbb{E}\left[\frac{\eta}{2}\sum_{t=0}^{T-1}\|\boldsymbol{V}_t^{-1}\boldsymbol{g}_t\|_{\boldsymbol{V}_t}^2\right].$$

Since $\boldsymbol{V}_t \succeq \boldsymbol{V}_{t-1}$, we have $\|\boldsymbol{x}_t - \boldsymbol{x}^*\|_{\boldsymbol{V}_t - \boldsymbol{V}_{t-1}}^2 \leq \|\boldsymbol{x}_t - \boldsymbol{x}^*\|_{\mathcal{H}}^2 \operatorname{Tr}(\boldsymbol{V}_t - \boldsymbol{V}_{t-1}) \leq D^2 \cdot \operatorname{Tr}(\boldsymbol{V}_t - \boldsymbol{V}_{t-1})$, where the second inequality holds because of the assumption that $\max_{t\in[T]}\|\boldsymbol{x}_t - \boldsymbol{x}^*\|_{\mathcal{H}} \leq D$. Therefore, we further have

$$\sum_{t=0}^{T-1} \mathbb{E}\left[f^{\alpha_t,\bar{\boldsymbol{x}}_t}(\boldsymbol{x}_t) - f^{\alpha_t,\bar{\boldsymbol{x}}_t}(\boldsymbol{x}^*)\right] \leq \mathbb{E}\left[\frac{D^2}{2\eta}\sum_{t=0}^{T-1}\operatorname{Tr}(\boldsymbol{V}_t - \boldsymbol{V}_{t-1}) + \frac{\eta}{2}\sum_{t=0}^{T-1}\|\boldsymbol{V}_t^{-1}\boldsymbol{g}_t\|_{\boldsymbol{V}_t}^2\right]$$
$$= \mathbb{E}\left[\frac{D^2}{2\eta}\operatorname{Tr}(\boldsymbol{V}_{T-1}) + \frac{\eta}{2}\sum_{t=0}^{T-1}\|\boldsymbol{V}_t^{-1}\boldsymbol{g}_t\|_{\boldsymbol{V}_t}^2\right]$$
$$= \mathbb{E}\left[\frac{D^2}{2\eta}\langle\boldsymbol{M}_{T-1} + \epsilon\boldsymbol{I}_d, \boldsymbol{V}_{T-1}^{-1}\rangle + \frac{\eta}{2}\sum_{t=0}^{T-1}\|\boldsymbol{V}_t^{-1}\boldsymbol{g}_t\|_{\boldsymbol{V}_t}^2\right]$$
$$= \mathbb{E}\left[\frac{D^2}{2\eta}\epsilon \cdot \operatorname{Tr}(\boldsymbol{V}_{T-1}^{-1}) + \frac{D^2}{2\eta}\sum_{t=0}^{T-1}\boldsymbol{g}_t^\top\boldsymbol{V}_{T-1}^{-1}\boldsymbol{g}_t + \frac{\eta}{2}\sum_{t=0}^{T-1}\boldsymbol{g}_t^\top\boldsymbol{V}_t^{-1}\boldsymbol{g}_t\right].$$

Since $\boldsymbol{V}_t \preceq \boldsymbol{V}_{T-1}$ for all $t \in [T-1]$, we have $\boldsymbol{g}_t^\top\boldsymbol{V}_{T-1}^{-1}\boldsymbol{g}_t \leq \boldsymbol{g}_t^\top\boldsymbol{V}_t^{-1}\boldsymbol{g}_t$ for all $t \in [T-1]$. Consequently,

$$\sum_{t=0}^{T-1}\mathbb{E}[f^{\alpha_t,\bar{\boldsymbol{x}}_t}(\boldsymbol{x}_t) - f^{\alpha_t,\bar{\boldsymbol{x}}_t}(\boldsymbol{x}^*)] \leq \mathbb{E}\left[\frac{D^2\epsilon}{2\eta}\operatorname{Tr}(\boldsymbol{V}_{T-1}^{-1}) + \left(\frac{D^2}{2\eta} + \frac{\eta}{2}\right)\sum_{t=0}^{T-1}\boldsymbol{g}_t^\top\boldsymbol{V}_t^{-1}\boldsymbol{g}_t\right].$$

This completes the proof. $\qquad\square$

**Lemma E.3.** *Under the setting of Theorem 4.3, it holds that*

$$\sum_{t=0}^{T-1}(f^{\alpha_t,\bar{\boldsymbol{x}}_t}(\boldsymbol{x}_{t+1}) - f^{\alpha_t,\bar{\boldsymbol{x}}_t}(\boldsymbol{x}^*)) \leq \sum_{t=0}^{T-1}(f^{\alpha_t,\bar{\boldsymbol{x}}_t}(\boldsymbol{x}_t) - f^{\alpha_t,\bar{\boldsymbol{x}}_t}(\boldsymbol{x}^*)) - \eta\sum_{t=0}^{T-1}\|\boldsymbol{V}_t^{-1}\boldsymbol{g}_t\|_{\boldsymbol{V}_t}^2$$
$$+ \eta\sum_{t=0}^{T-1}\langle\boldsymbol{g}_t - \nabla f^{\alpha_t,\bar{\boldsymbol{x}}_t}(\boldsymbol{x}_t), \boldsymbol{V}_t^{-1}\boldsymbol{g}_t\rangle + \frac{\Lambda_{\mathcal{H}}(f)\eta^2}{2}\sum_{t=0}^{T-1}\|\boldsymbol{V}_t^{-1}\boldsymbol{g}_t\|_{\boldsymbol{H}^*}^2.$$

*Proof of Lemma E.3.* By Definition 2.4, we have that

$$f(\boldsymbol{y}) \leq f(\boldsymbol{x}) + \langle\nabla f(\boldsymbol{x}), \boldsymbol{y} - \boldsymbol{x}\rangle + \frac{\Lambda_{\mathcal{H}}(f)}{2}\|\boldsymbol{x} - \boldsymbol{y}\|_{\boldsymbol{H}^*}^2.$$

By the definition of $f^{\alpha_t,\bar{\boldsymbol{x}}_t}$ in (8),

$$f^{\alpha_t,\bar{\boldsymbol{x}}_t}(\boldsymbol{x}_{t+1}) \leq f^{\alpha_t,\bar{\boldsymbol{x}}_t}(\boldsymbol{x}_t) + \langle\nabla f^{\alpha_t,\bar{\boldsymbol{x}}_t}(\boldsymbol{x}_t), \boldsymbol{x}_{t+1} - \boldsymbol{x}_t\rangle + \frac{\Lambda_{\mathcal{H}}(f)}{2}\|\boldsymbol{x}_{t+1} - \boldsymbol{x}_t\|_{\boldsymbol{H}^*}^2$$
$$= f^{\alpha_t,\bar{\boldsymbol{x}}_t}(\boldsymbol{x}_t) - \eta\langle\nabla f^{\alpha_t,\bar{\boldsymbol{x}}_t}(\boldsymbol{x}_t), \boldsymbol{V}_t^{-1}\boldsymbol{g}_t\rangle + \frac{\Lambda_{\mathcal{H}}(f)\eta^2}{2}\|\boldsymbol{V}_t^{-1}\boldsymbol{g}_t\|_{\boldsymbol{H}^*}^2$$
$$= f^{\alpha_t,\bar{\boldsymbol{x}}_t}(\boldsymbol{x}_t) - \eta\langle\boldsymbol{g}_t, \boldsymbol{V}_t^{-1}\boldsymbol{g}_t\rangle + \eta\langle\boldsymbol{g}_t - \nabla f^{\alpha_t,\bar{\boldsymbol{x}}_t}(\boldsymbol{x}_t), \boldsymbol{V}_t^{-1}\boldsymbol{g}_t\rangle + \frac{\Lambda_{\mathcal{H}}(f)\eta^2}{2}\|\boldsymbol{V}_t^{-1}\boldsymbol{g}_t\|_{\boldsymbol{H}^*}^2$$

where we plug in the update rule for $\boldsymbol{x}_{t+1}$ in Algorithm 2. Summing over $t$ yields

$$\sum_{t=0}^{T-1}(f^{\alpha_t,\bar{\boldsymbol{x}}_t}(\boldsymbol{x}_{t+1}) - f^{\alpha_t,\bar{\boldsymbol{x}}_t}(\boldsymbol{x}^*)) \leq \sum_{t=0}^{T-1}(f^{\alpha_t,\bar{\boldsymbol{x}}_t}(\boldsymbol{x}_t) - f^{\alpha_t,\bar{\boldsymbol{x}}_t}(\boldsymbol{x}^*)) - \eta\sum_{t=0}^{T-1}\|\boldsymbol{V}_t^{-1}\boldsymbol{g}_t\|_{\boldsymbol{V}_t}^2$$
$$+ \eta\sum_{t=0}^{T-1}\langle\boldsymbol{g}_t - \nabla f^{\alpha_t,\bar{\boldsymbol{x}}_t}(\boldsymbol{x}_t), \boldsymbol{V}_t^{-1}\boldsymbol{g}_t\rangle + \frac{\Lambda_{\mathcal{H}}(f)\eta^2}{2}\sum_{t=0}^{T-1}\|\boldsymbol{V}_t^{-1}\boldsymbol{g}_t\|_{\boldsymbol{H}^*}^2.$$

This completes the proof. $\qquad\square$

The following lemma is a standard result for mirror descent.

**Lemma E.4** (Lemma 5, Gupta et al. 2017)**.** *For any $\boldsymbol{x}_0 \in \mathcal{X}$, $\boldsymbol{g} \in \mathbb{R}^d$ and $\boldsymbol{H} \succeq 0$, let $\boldsymbol{x}_1 = \arg\min_{x \in \mathcal{X}} \langle \boldsymbol{g}, \boldsymbol{x}_0 \rangle + \|\boldsymbol{x} - \boldsymbol{x}_0\|_{\boldsymbol{H}}^2$. Then it holds that*

$$\langle \boldsymbol{g}, \boldsymbol{x}_0 - \boldsymbol{x}_1 \rangle \leq \frac{1}{2} \|\boldsymbol{H}^{-1} \boldsymbol{g}\|_{\boldsymbol{H}}^2 + \frac{1}{2} \|\boldsymbol{x}_0 - \boldsymbol{x}_1\|_{\boldsymbol{H}}^2.$$

*Moreover, for any $\boldsymbol{x} \in \mathcal{X}$, it holds that*

$$\langle \boldsymbol{g}, \boldsymbol{x}_1 - \boldsymbol{x} \rangle \leq \frac{1}{2} \|\boldsymbol{x}_0 - \boldsymbol{x}\|_{\boldsymbol{H}}^2 - \frac{1}{2} \|\boldsymbol{x}_1 - \boldsymbol{x}\|_{\boldsymbol{H}}^2 - \frac{1}{2} \|\boldsymbol{x}_0 - \boldsymbol{x}_1\|_{\boldsymbol{H}}^2,$$

$$\langle \boldsymbol{g}, \boldsymbol{x}_0 - \boldsymbol{x} \rangle \leq \frac{1}{2} \|\boldsymbol{x}_0 - \boldsymbol{x}\|_{\boldsymbol{H}}^2 - \frac{1}{2} \|\boldsymbol{x}_1 - \boldsymbol{x}\|_{\boldsymbol{H}}^2 + \frac{1}{2} \|\boldsymbol{H}^{-1} \boldsymbol{g}\|_{\boldsymbol{H}}^2.$$

### E.2 STOCHASTIC CASE WITH PROJECTION

---

**Algorithm 8** Accelerated Adaptive Algorithm with Projection

---

**Hyperparam:** $\epsilon \geq 0$, total steps $T$, learning rate $\eta$, convex cone $\mathcal{H} \subset \mathcal{S}_+$, radius $D$?
**Input:** initialization $\boldsymbol{x}_0 = \bar{\boldsymbol{x}}_0$, a sequence of positive constants $\alpha_0, \ldots, \alpha_T \in (0,1]$

$\quad \boldsymbol{M}_{-1} \leftarrow \boldsymbol{0}$
$\quad$ **for** $t = 0, 1, 2, \ldots, T$ :
$\quad\quad \boldsymbol{g}_t \leftarrow \nabla f_t^{\alpha_t, \bar{\boldsymbol{x}}_t}(\boldsymbol{x}_t)$ where $f_t^{\alpha_t, \bar{\boldsymbol{x}}_t}$ is defined in (8)
$\quad\quad \boldsymbol{M}_t \leftarrow \boldsymbol{M}_{t-1} + \boldsymbol{g}_t \boldsymbol{g}_t^\top$
$\quad\quad \boldsymbol{V}_t \leftarrow \arg\min_{\boldsymbol{H} \in \mathcal{H}} \left\langle \boldsymbol{M}_t + \epsilon \boldsymbol{I}_d, \boldsymbol{H}^{-1} \right\rangle + \mathrm{Tr}(\boldsymbol{H})$
$\quad\quad \boldsymbol{x}_{t+1/2} \leftarrow \boldsymbol{x}_t - \eta \boldsymbol{V}_t^{-1} \boldsymbol{g}_t$
$\quad\quad \boldsymbol{x}_{t+1} \leftarrow \arg\min_{\|\boldsymbol{x}\|_{\mathcal{H}} \leq D} \|\boldsymbol{x} - \boldsymbol{x}_{t+1/2}\|_{\boldsymbol{V}_t}^2$
$\quad\quad \bar{\boldsymbol{x}}_{t+1} \leftarrow \alpha_t \boldsymbol{x}_{t+1/2} + (1 - \alpha_t) \bar{\boldsymbol{x}}_t$

$\quad$ **return** $\bar{\boldsymbol{x}}_T$

---

Theorem E.5 shows the convergence rate of Algorithm 8, which is the same as Algorithm 2 while removing the dependence on prior knowledge of $D$.

**Theorem E.5.** *For a well-structured preconditioner set $\mathcal{H}$, let $f$ be a convex loss function whose $\mathcal{H}$-smoothness constant is $\Lambda_{\mathcal{H}}(f) \in (0, \infty)$ according to Definition 2.4. For $\epsilon > 0$, $T > 0$, consider Algorithm 8 with $\alpha_t = 2/(t+2)$ for $t = 0, 1, \ldots, T-1$. Suppose the global minima $\boldsymbol{x}^*$ satisfies that $\|\boldsymbol{x}^*\|_{\mathcal{H}} \leq D$ and Assumption 4.2 holds with adaptive gradient variance $\sigma_{\mathcal{H}}(\{f_t\}_{t=1}^T)^2 \leq \sigma_{\mathcal{H}}^2$ for some $\sigma_{\mathcal{H}} \in [0, \infty)$. Then it holds that*

$$\mathbb{E}[f(\bar{\boldsymbol{x}}_T) - f(\boldsymbol{x}^*)] \leq \frac{2D^2 \epsilon}{\eta(T+1)^2} \mathbb{E}\,\mathrm{Tr}(\boldsymbol{V}_{T-1}^{-1}) + \left(\frac{D^2}{2\eta} - \frac{\eta}{2}\right) \frac{4}{(T+1)^2} \mathbb{E} \sum_{t=0}^{T-1} \boldsymbol{g}_t^\top \boldsymbol{V}_t^{-1} \boldsymbol{g}_t$$

$$+ \frac{2\eta^2}{(T+1)^2} \cdot \Lambda_{\mathcal{H}}(f) \cdot \tilde{O}(\log^2 d) + \frac{\eta}{T^{1/2}} \sigma_{\mathcal{H}} \cdot \tilde{O}(\log d).$$

*Moreover, when choosing learning rate $\eta = D$, the convergence rate becomes*

$$\mathbb{E}[f(\bar{\boldsymbol{x}}_T) - f(\boldsymbol{x}^*)] = \tilde{O}\left(\frac{\Lambda_{\mathcal{H}}(f) D^2 \log^2 d + d\sqrt{\epsilon} D}{T^2} + \frac{\sigma_{\mathcal{H}} D \log d}{\sqrt{T}}\right).$$

*Proof of Theorem E.5.* Combining the results of Lemma E.7 and Lemma E.8, we get

$$\sum_{t=0}^{T-1} \mathbb{E}[f^{\alpha_t, \bar{\boldsymbol{x}}_t}(\boldsymbol{x}_{t+1/2}) - f^{\alpha_t, \bar{\boldsymbol{x}}_t}(\boldsymbol{x}^*)] \leq \mathbb{E}\left[\frac{D^2 \epsilon}{2\eta} \mathrm{Tr}(\boldsymbol{V}_{T-1}^{-1}) + \left(\frac{D^2}{2\eta} - \frac{\eta}{2}\right) \sum_{t=0}^{T-1} \boldsymbol{g}_t^\top \boldsymbol{V}_t^{-1} \boldsymbol{g}_t\right]$$

$$+ \mathbb{E}\left[\eta \sum_{t=0}^{T-1} \langle \boldsymbol{g}_t - \nabla f^{\alpha_t, \bar{\boldsymbol{x}}_t}(\boldsymbol{x}_t), \boldsymbol{V}_t^{-1} \boldsymbol{g}_t \rangle\right]$$

$$+ \mathbb{E}\left[\frac{\Lambda_{\mathcal{H}}(f) \eta^2}{2} \sum_{t=0}^{T-1} \|\boldsymbol{V}_t^{-1} \boldsymbol{g}_t\|_{\boldsymbol{H}^*}^2\right].$$

The rest of the proof will be the same as the proof of Theorem 4.3. $\qquad\square$

**Lemma E.6.** *Under the setting of Theorem E.5, it holds that*

$$\mathbb{E}[f(\bar{\boldsymbol{x}}_T) - f(\boldsymbol{x}^*)] \leq \frac{4}{(T+1)^2} \sum_{t=0}^{T-1} \mathbb{E}\left[f^{\alpha_t, \bar{\boldsymbol{x}}_t}(\boldsymbol{x}_{t+1/2}) - f^{\alpha_t, \bar{\boldsymbol{x}}_t}(\boldsymbol{x}^*)\right] \tag{45}$$

*Proof of Lemma E.6.* The proof is the same as the proof of Lemma E.1 with $\boldsymbol{x}_{t+1/2}$ in place of $\boldsymbol{x}_{t+1}$. $\qquad\square$

Define $\boldsymbol{V}_{-1} = 0$ for convenience.

**Lemma E.7.** *Under the setting of Theorem E.5, it holds that*

$$\sum_{t=0}^{T-1} \mathbb{E}\left[f^{\alpha_t, \bar{\boldsymbol{x}}_t}(\boldsymbol{x}_t) - f^{\alpha_t, \bar{\boldsymbol{x}}_t}(\boldsymbol{x}^*)\right] \leq \mathbb{E}\left[\frac{D^2 \epsilon}{2\eta} \operatorname{Tr}(\boldsymbol{V}_{T-1}^{-1}) + \left(\frac{D^2}{2\eta} + \frac{\eta}{2}\right) \sum_{t=0}^{T-1} \boldsymbol{g}_t^\top \boldsymbol{V}_t^{-1} \boldsymbol{g}_t\right].$$

*Proof of Lemma E.7.* The proof is the same as the proof of Lemma E.2, except that controlling $\langle \boldsymbol{g}_t, \boldsymbol{x}_t - \boldsymbol{x}^* \rangle$ requires some extra work. Specifically, by definition $\boldsymbol{x}_{t+1/2} = \boldsymbol{x}_t - \eta \boldsymbol{V}_t^{-1} \boldsymbol{g}_t$, so it follows from Lemma E.4 that

$$\langle \boldsymbol{g}_t, \boldsymbol{x}_t - \boldsymbol{x}^* \rangle \leq \frac{1}{2\eta} \|\boldsymbol{x}_t - \boldsymbol{x}^*\|_{\boldsymbol{V}_t}^2 - \frac{1}{2\eta} \|\boldsymbol{x}_{t+1/2} - \boldsymbol{x}^*\|_{\boldsymbol{V}_t}^2 + \frac{\eta}{2} \|\boldsymbol{V}_t^{-1} \boldsymbol{g}_t\|_{\boldsymbol{V}_t}^2.$$

Since $\boldsymbol{x}_{t+1} = \arg\min_{\boldsymbol{x} \in \mathcal{X}, \|\boldsymbol{x}\|_{\mathcal{H}} \leq D} \|\boldsymbol{x} - \boldsymbol{x}_{t+1/2}\|_{\boldsymbol{V}_t}^2$ and $\|\boldsymbol{x}^*\|_{\mathcal{H}} \leq D$, it holds that $\frac{\mathrm{d}}{\mathrm{d}\lambda}\|\boldsymbol{x}_{t+1} + \lambda(\boldsymbol{x}^* - \boldsymbol{x}_{t+1}) - \boldsymbol{x}_{t+1/2}\|_{\boldsymbol{V}_t}^2\big|_{\lambda=0} \geq 0$, which implies that $(\boldsymbol{x}^* - \boldsymbol{x}_{t+1})^\top \boldsymbol{V}_t(\boldsymbol{x}_{t+1} - \boldsymbol{x}_{t+1/2}) \geq 0$. Then it follows that $\|\boldsymbol{x}_{t+1/2} - \boldsymbol{x}^*\|_{\boldsymbol{V}_t}^2 = \|\boldsymbol{x}_{t+1} - \boldsymbol{x}^*\|_{\boldsymbol{V}_t}^2 + 2(\boldsymbol{x}^* - \boldsymbol{x}_{t+1})^\top \boldsymbol{V}_t(\boldsymbol{x}_{t+1} - \boldsymbol{x}_{t+1/2}) + \|\boldsymbol{x}_{t+1} - \boldsymbol{x}_{t+1/2}\|_{\boldsymbol{V}_t}^2 \geq \|\boldsymbol{x}_{t+1} - \boldsymbol{x}^*\|_{\boldsymbol{V}_t}^2$. Therefore, applying this to the above inequality yields

$$\langle \boldsymbol{g}_t, \boldsymbol{x}_t - \boldsymbol{x}^* \rangle \leq \frac{1}{2\eta} \|\boldsymbol{x}_t - \boldsymbol{x}^*\|_{\boldsymbol{V}_t}^2 - \frac{1}{2\eta} \|\boldsymbol{x}_{t+1} - \boldsymbol{x}^*\|_{\boldsymbol{V}_t}^2 + \frac{\eta}{2} \|\boldsymbol{V}_t^{-1} \boldsymbol{g}_t\|_{\boldsymbol{V}_t}^2.$$

Then the rest of the proof is the same. $\qquad\square$

**Lemma E.8.** *Under the setting of Theorem E.5, it holds that*

$$\sum_{t=0}^{T-1} (f^{\alpha_t, \bar{\boldsymbol{x}}_t}(\boldsymbol{x}_{t+1/2}) - f^{\alpha_t, \bar{\boldsymbol{x}}_t}(\boldsymbol{x}^*)) \leq \sum_{t=0}^{T-1} (f^{\alpha_t, \bar{\boldsymbol{x}}_t}(\boldsymbol{x}_t) - f^{\alpha_t, \bar{\boldsymbol{x}}_t}(\boldsymbol{x}^*)) - \eta \sum_{t=0}^{T-1} \|\boldsymbol{V}_t^{-1} \boldsymbol{g}_t\|_{\boldsymbol{V}_t}^2$$
$$+ \eta \sum_{t=0}^{T-1} \langle \boldsymbol{g}_t - \nabla f^{\alpha_t, \bar{\boldsymbol{x}}_t}(\boldsymbol{x}_t), \boldsymbol{V}_t^{-1} \boldsymbol{g}_t \rangle + \frac{\eta^2}{2} \sum_{t=0}^{T-1} \|\boldsymbol{V}_t^{-1} \boldsymbol{g}_t\|_{\boldsymbol{H}^*}^2.$$

*Proof of Lemma E.8.* The proof is the same as the proof of Lemma E.3 with $\boldsymbol{x}_{t+1/2}$ in place of $\boldsymbol{x}_{t+1}$. $\qquad\square$

## F    PROOF FOR NORMALIZED STEEPEST DESCENT

Suppose that $\|\cdot\|$ is a norm on $\mathbb{R}^d$ and $\|\cdot\|_*$ is its dual norm. The following descent lemma characterizes the progress of one step of steepest descent under $\|\cdot\|$.

**Lemma F.1** (Descent lemma). *Let $f : \mathbb{R}^d \to \mathbb{R}$ be a differentiable function. For any $\boldsymbol{m} \in \mathbb{R}^d$, let $\boldsymbol{u} = \arg\max_{\|\boldsymbol{v}\| \leq 1} \langle \boldsymbol{m}, \boldsymbol{v} \rangle$. Then for any $\eta > 0$, it holds that*

$$f(\boldsymbol{x} - \eta \boldsymbol{u}) \leq f(\boldsymbol{x}) - \eta \|\nabla f(\boldsymbol{x})\|_* + 2\eta \|\nabla f(\boldsymbol{x}) - \boldsymbol{m}\|_* + \frac{\eta^2}{2} L_{\|\cdot\|}(f).$$

*Proof of Lemma F.1.* By definition of the smoothness in Definition 2.3, we have

$$f(\boldsymbol{x} - \eta \boldsymbol{u}) \leq f(\boldsymbol{x}) - \nabla f(\boldsymbol{x})^\top (\eta \boldsymbol{u}) + \frac{\|\eta \boldsymbol{u}\|^2}{2} L_{\|\cdot\|}(f)$$

$$= f(\boldsymbol{x}) - \eta \langle \nabla f(\boldsymbol{x}) - \boldsymbol{m}, \boldsymbol{u} \rangle - \eta \langle \boldsymbol{m}, \boldsymbol{u} \rangle + \frac{\eta^2}{2} L_{\|\cdot\|}(f)$$

$$\leq f(\boldsymbol{x}) + \eta \|\nabla f(\boldsymbol{x}) - \boldsymbol{m}\|_* - \eta \|\boldsymbol{m}\|_* + \frac{\eta^2}{2} L_{\|\cdot\|}(f)$$

where in the last step we use $\|u\| = 1$ and $\langle m, u \rangle = \|m\|_*$ by the definition of $u$. Then we further apply the triangle inequality $\|m\|_* \geq \|\nabla f(x)\|_* - \|\nabla f(x) - m\|_*$ to get

$$f(x - \eta u) \leq f(x) - \eta \|\nabla f(x)\|_* + 2\eta \|\nabla f(x) - m\|_* + \frac{\eta^2}{2} L_{\|\cdot\|}(f).$$

This concludes the proof. $\qquad\square$

**Proposition F.2.** *For any $\epsilon \geq 0$, $\alpha \in (0, 1)$, $\eta > 0$, and $T \in \mathbb{N}$, let $\{x_t\}_{t=0}^T$ be the iterates of Algorithm 3 with $\|\cdot\|$. Let $L_{\|\cdot\|}(f)$ be the smoothness of the loss $f$ w.r.t. $\|\cdot\|$ according to Definition 2.3. Suppose Assumption 4.2 holds. Then it holds that*

$$\frac{1}{T} \sum_{t=0}^{T-1} \mathbb{E} \|\nabla f(x_t)\|_* \leq \frac{\Delta_0}{\eta T} + \frac{L_{\|\cdot\|}(f)\eta}{2} + \frac{2}{\alpha T} \mathbb{E} \|m_0 - \nabla f(x_0)\|_* + \frac{2(1-\alpha)\eta}{\alpha} L_{\|\cdot\|}(f)$$

$$+ \frac{2\alpha}{T} \sum_{t=0}^{T-1} \mathbb{E} \left\| \sum_{i=1}^t (1-\alpha)^{t-i} (g_i - \nabla f(x_i)) \right\|_*$$

*Proposition F.2.* Apply Lemma F.1 with $\|\cdot\|$ yields that

$$f(x_{t+1}) - f(x_t) \leq -\eta \|\nabla f(x_t)\|_* + \frac{L_{\|\cdot\|}(f)\eta^2}{2} + 2\eta \|\nabla f(x_t) - m_t\|_*. \tag{46}$$

Rearranging the telescoping sum of Eq. 46 from $t = 0$ to $T - 1$, we have that

$$\frac{1}{T} \sum_{t=0}^{T-1} \mathbb{E} \|\nabla f(x_t)\|_* \leq \frac{1}{\eta T} [f(x_0) - \min f(x)] + \frac{L_{\|\cdot\|}(f)\eta}{2} + \frac{2}{T} \sum_{t=0}^{T-1} \mathbb{E} \|\nabla f(x_t) - m_t\|_*. \tag{47}$$

By the update rule of Algorithm 3, we have $m_t = (1-\alpha)m_{t-1} + \alpha g_t$, so
$$m_t - \nabla f(x_t) = (1-\alpha)(m_{t-1} - \nabla f(x_{t-1})) + (1-\alpha)(\nabla f(x_{t-1}) - \nabla f(x_t)) + \alpha(g_t - \nabla f(x_t)).$$
Unrolling the above recursion, we get that
$$m_t - \nabla f(x_t) = (1-\alpha)^t (m_0 - \nabla f(x_0))$$
$$+ \sum_{i=1}^t (1-\alpha)^{t-i} [(1-\alpha)(\nabla f(x_{i-1}) - \nabla f(x_i)) + \alpha(g_i - \nabla f(x_i))].$$

and

$$\mathbb{E} \|\nabla f(x_t) - m_t\|_* \leq (1-\alpha)^t \mathbb{E} \|m_0 - \nabla f(x_0)\|_* + \sum_{i=1}^t (1-\alpha)^{t-i+1} \|\nabla f(x_{i-1}) - \nabla f(x_i)\|_*$$

$$+ \alpha \mathbb{E} \left\| \sum_{i=1}^t (1-\alpha)^{t-i} (g_i - \nabla f(x_i)) \right\|_* \tag{48}$$

We know that

$$\|\nabla f(x_{i-1}) - \nabla f(x_i)\|_* \leq L_{\|\cdot\|}(f) \|x_{i-1} - x_i\| \leq \eta L_{\|\cdot\|}(f). \tag{49}$$

Then plugging (49) into (48), we can get that

$$\mathbb{E} \|\nabla f(x_t) - m_t\|_* \leq (1-\alpha)^t \mathbb{E} \|m_0 - \nabla f(x_0)\|_* + (1-\alpha)\eta L_{\|\cdot\|}(f) \sum_{i=1}^t (1-\alpha)^{t-i}$$

$$+ \alpha \mathbb{E} \left\| \sum_{i=1}^t (1-\alpha)^{t-i} (g_i - \nabla f(x_i)) \right\|_*$$

$$\leq (1-\alpha)^t \mathbb{E} \|m_0 - \nabla f(x_0)\|_* + \frac{1-\alpha}{\alpha} \eta L_{\|\cdot\|}(f)$$

$$+ \alpha \mathbb{E} \left\| \sum_{i=1}^t (1-\alpha)^{t-i} (g_i - \nabla f(x_i)) \right\|_*.$$

Plugging it in Eq. 47, we have that

$$\frac{1}{T}\sum_{t=0}^{T-1}\mathbb{E}\left\|\nabla f(\boldsymbol{x}_t)\right\|_* \leq \frac{\Delta_0}{\eta T} + \frac{L_{\|\cdot\|}(f)\eta}{2} + \frac{2}{T}\sum_{t=0}^{T-1}\left[(1-\alpha)^t\mathbb{E}\left\|\boldsymbol{m}_0 - \nabla f(\boldsymbol{x}_0)\right\|_*\right]$$

$$+ \frac{2}{T}\sum_{t=0}^{T-1}\left[\frac{1-\alpha}{\alpha}\eta L_{\|\cdot\|}(f) + \alpha\mathbb{E}\left\|\sum_{i=1}^{t}(1-\alpha)^{t-i}(\boldsymbol{g}_i - \nabla f(\boldsymbol{x}_i))\right\|_*\right]$$

$$= \frac{\Delta_0}{\eta T} + \frac{L_{\|\cdot\|}(f)\eta}{2} + \frac{2}{\alpha T}\mathbb{E}\left\|\boldsymbol{m}_0 - \nabla f(\boldsymbol{x}_0)\right\|_* + \frac{2(1-\alpha)\eta}{\alpha}L_{\|\cdot\|}(f)$$

$$+ \frac{2\alpha}{T}\sum_{t=0}^{T-1}\mathbb{E}\left\|\sum_{i=1}^{t}(1-\alpha)^{t-i}(\boldsymbol{g}_i - \nabla f(\boldsymbol{x}_i))\right\|_*$$

$\square$

## F.1 CONVERGENCE RATE WITH RESPECT TO ADAPTIVE GRADIENT VARIANCE

The proof of Theorem 4.5 largely borrows from Kovalev (2025b), upon which we change the dependence on adaptive noise and improve some coefficient constant.

**Theorem 4.5.** *Let $\mathcal{H}$ be a well-structured preconditioner set. For any $\epsilon \geq 0$, $\alpha \in (0,1)$, $\eta > 0$, and $T \in \mathbb{N}$, let $\{\boldsymbol{x}_t\}_{t=0}^{T}$ be the iterates of Algorithm 3 with $\|\cdot\| = \|\cdot\|_{\mathcal{H}}$ and $\boldsymbol{m}_0 = \nabla f_0(\boldsymbol{x}_0)$. Let $L_{\|\cdot\|_{\mathcal{H}}}(f)$ be the smoothness of the loss $f$ w.r.t. $\|\cdot\|_{\mathcal{H}}$ according to Definition 2.3. Suppose Assumption 4.2 holds with adaptive gradient variance $\sigma_{\mathcal{H}}(\{f_t\}_{t=1}^{T})^2 \leq \sigma_{\mathcal{H}}^2$. Then it holds that*

$$\mathbb{E}\frac{1}{T}\sum_{t=0}^{T-1}\left\|\nabla f(\boldsymbol{x}_t)\right\|_{\mathcal{H},*} \leq \frac{\Delta_0}{\eta T} + \frac{2\eta}{\alpha}L_{\|\cdot\|}(f) + \frac{2\sigma_{\mathcal{H}}}{\alpha T} + 2\sigma_{\mathcal{H}}\sqrt{\alpha}.$$

*Let $a_0 = \sqrt{\Delta_0 L_{\|\cdot\|}(f)}/\sigma_{\mathcal{H}}$. If $a_0 < 1$, then*

- *When $T < a_0^{-6}$, we choose $\alpha = T^{-2/3}$ and $\eta = \sqrt{\frac{\Delta_0}{L_{\|\cdot\|_{\mathcal{H}}}(f)}}T^{-5/12}$. The rate is $O\left(\sigma_{\mathcal{H}}T^{-1/3}\right)$.*

- *When $T \geq a_0^{-6}$, we choose $\alpha = \frac{a_0}{\sqrt{T}}$, $\eta = \frac{\Delta_0^{3/4}T^{-3/4}}{L_{\|\cdot\|_{\mathcal{H}}}(f)^{1/4}\sigma_{\mathcal{H}}^{1/2}}$. The rate is $O\left(\frac{(\Delta_0 L_{\|\cdot\|_{\mathcal{H}}}(f))^{1/4}\sqrt{\sigma_{\mathcal{H}}}}{T^{1/4}}\right)$.*

*If $a_0 \geq 1$, then*

- *When $T \leq a_0^2$, we choose $\alpha = 1$ and $\eta = \sqrt{\frac{\Delta_0}{L_{\|\cdot\|_{\mathcal{H}}}(f)}}T^{-1/2}$. The rate is $O\left(\sqrt{\Delta_0 L_{\|\cdot\|_{\mathcal{H}}}(f)}T^{-1/2}\right)$.*

- *When $T \geq a_0^2$, we choose $\alpha = \frac{a_0}{\sqrt{T}}$ and $\eta = \frac{\Delta_0^{3/4}T^{-3/4}}{L_{\|\cdot\|_{\mathcal{H}}}(f)^{1/4}\sigma_{\mathcal{H}}^{1/2}}$. The rate is $O\left(\frac{(\Delta_0 L_{\|\cdot\|_{\mathcal{H}}}(f))^{1/4}\sqrt{\sigma_{\mathcal{H}}}}{T^{1/4}}\right)$.*

*Proof of Theorem 4.5.* By applying Proposition F.2 with $\|\cdot\| = \|\cdot\|_{\mathcal{H}}$, we have that

$$\frac{1}{T}\sum_{t=0}^{T-1}\mathbb{E}\left\|\nabla f(\boldsymbol{x}_t)\right\|_{\mathcal{H},*} \leq \frac{\Delta_0}{\eta T} + \frac{L_{\|\cdot\|}(f)\eta}{2} + \frac{2}{\alpha T}\mathbb{E}\left\|\boldsymbol{m}_0 - \nabla f(\boldsymbol{x}_0)\right\|_{\mathcal{H},*} + \frac{2(1-\alpha)\eta}{\alpha}L_{\|\cdot\|}(f)$$

$$+ \frac{2}{T}\sum_{t=0}^{T-1}\mathbb{E}\left\|\sum_{i=1}^{t}(1-\alpha)^{t-i}\alpha(\boldsymbol{g}_i - \nabla f(\boldsymbol{x}_i))\right\|_{\mathcal{H},*}. \tag{50}$$

We only need to bound $\mathbb{E}\left\|\boldsymbol{m}_0 - \nabla f(\boldsymbol{x}_0)\right\|_{\mathcal{H},*}$ and $\mathbb{E}\left\|\sum_{i=1}^{t}(1-\alpha)^{t-i}\alpha(\boldsymbol{g}_i - \nabla f(\boldsymbol{x}_i))\right\|_{\mathcal{H},*}$. For notational simplicity, we define $\omega_i = (1-\alpha)^{t-i}\alpha$ and $\boldsymbol{\delta}_i = \boldsymbol{g}_i - \nabla f(\boldsymbol{x}_i)$. By Assumption 4.2, we

know that $\mathbb{E}[\boldsymbol{\delta}_j|\boldsymbol{\delta}_i] = 0$ for $j > 1$. Then we have that

$$\mathbb{E}\left\|\sum_{i=1}^{t}(1-\alpha)^{t-i}\alpha(\boldsymbol{g}_i - \nabla f(\boldsymbol{x}_i))\right\|_{\mathcal{H},*}^2 = \mathbb{E}\left\|\sum_{i=1}^{t}\omega_i\boldsymbol{\delta}_i\right\|_{\mathcal{H},*}^2$$

$$= \mathbb{E}\min_{\boldsymbol{H}\in\mathcal{H},\mathrm{Tr}(\boldsymbol{H})\leq 1}\Big(\sum_{i=1}^{t}\omega_i\boldsymbol{\delta}_i\Big)\boldsymbol{H}^{-1}\Big(\sum_{i=1}^{t}\omega_i\boldsymbol{\delta}_i\Big)$$

$$\leq \min_{\boldsymbol{H}\in\mathcal{H},\mathrm{Tr}(\boldsymbol{H})\leq 1}\mathbb{E}\Big(\sum_{i=1}^{t}\omega_i\boldsymbol{\delta}_i\Big)\boldsymbol{H}^{-1}\Big(\sum_{i=1}^{t}\omega_i\boldsymbol{\delta}_i\Big)$$

$$= \min_{\boldsymbol{H}\in\mathcal{H},\mathrm{Tr}(\boldsymbol{H})\leq 1}\mathbb{E}\sum_{i=1}^{t}\omega_i^2\boldsymbol{\delta}_i^\top\boldsymbol{H}^{-1}\boldsymbol{\delta}_i$$

$$= \min_{\boldsymbol{H}\in\mathcal{H},\mathrm{Tr}(\boldsymbol{H})\leq 1}\mathbb{E}\sum_{i=1}^{t}\omega_i^2\|\boldsymbol{\delta}_i\|_{\boldsymbol{H}^{-1}}^2.$$

Now by the condition on $\sigma_{\mathcal{H}}$ and the definition of the adaptive gradient variance in Definition 4.1, we know that $\min_{\boldsymbol{H}\in\mathcal{H},\mathrm{Tr}(\boldsymbol{H})\leq 1}\mathbb{E}\|\boldsymbol{\delta}_i\|_{\boldsymbol{H}^{-1}}^2 \leq \sigma_{\mathcal{H}}^2$ for each $i \in [t]$. Therefore, we further have

$$\mathbb{E}\left\|\sum_{i=1}^{t}(1-\alpha)^{t-i}\alpha(\boldsymbol{g}_i - \nabla f(\boldsymbol{x}_i))\right\|_{\mathcal{H},*}^2 \leq \sigma_{\mathcal{H}}^2\sum_{i=1}^{t}\omega_i^2 \leq \sigma_{\mathcal{H}}^2\alpha \tag{51}$$

where the second inequality holds by the definition of $\omega_i$ and the condition that $\alpha \in (0,1)$. Similarly, we also have

$$\mathbb{E}\|\boldsymbol{m}_0 - \nabla f(\boldsymbol{x}_0)\|_{\mathcal{H},*}^2 = \mathbb{E}\|\boldsymbol{g}_0 - \nabla f(\boldsymbol{x}_0)\|_{\mathcal{H},*}^2 \leq \sigma_{\mathcal{H}}^2. \tag{52}$$

Plugging in Eq. 51 and Eq. 52 into Eq. 50, we can get

$$\mathbb{E}\frac{1}{T}\sum_{t=0}^{T-1}\|\nabla f(\boldsymbol{x}_t)\|_{\mathcal{H},*} \leq \frac{\Delta_0}{\eta T} + \frac{\eta L_{\|\cdot\|_{\mathcal{H}}}(f)}{2} + \frac{2\sigma_{\mathcal{H}}}{\alpha T} + \frac{2(1-\alpha)\eta}{\alpha}L_{\|\cdot\|_{\mathcal{H}}}(f) + 2\sqrt{\alpha}\sigma_{\mathcal{H}}$$

$$\leq \frac{\Delta_0}{\eta T} + \frac{2\eta}{\alpha}L_{\|\cdot\|_{\mathcal{H}}}(f) + \frac{2\sigma_{\mathcal{H}}}{\alpha T} + 2\sqrt{\alpha}\sigma_{\mathcal{H}}. \tag{53}$$

Next, we will choose appropriate hyperparameters $\eta$ and $\alpha$ to achieve the optimal rate.

For any fixed $\alpha < 1$, to balance the first two terms on the right-hand side of (53), we choose

$$\eta = \sqrt{\frac{\Delta_0\alpha}{2TL_{\|\cdot\|_{\mathcal{H}}}(f)}}.$$

Also, for simplicity, denote $a_0 = \sqrt{\Delta_0 L_{\|\cdot\|_{\mathcal{H}}}(f)}/\sigma_{\mathcal{H}}$. Then the convergence rate becomes

$$\mathbb{E}\frac{1}{T}\sum_{t=0}^{T-1}\|\nabla f(\boldsymbol{x}_t)\|_{\mathcal{H},*} \leq 2\sqrt{\frac{2\Delta_0 L_{\|\cdot\|_{\mathcal{H}}}(f)}{\alpha T}} + \frac{2\sigma_{\mathcal{H}}}{\alpha T} + 2\sqrt{\alpha}\sigma_{\mathcal{H}}$$

$$\leq 2\sqrt{2}\sigma_{\mathcal{H}}(f)\Big(\frac{a_0}{\sqrt{\alpha T}} + \frac{1}{\alpha T} + \sqrt{\alpha}\Big). \tag{54}$$

We further stratify the values of $\alpha, a_0$ and $T$ to minimize the right-hand side:

- When $\alpha \leq T^{-2/3}$, $\frac{1}{\alpha T} \geq \sqrt{\alpha}$. We only need to minimize $\frac{1}{\alpha T} + \frac{a_0}{\sqrt{\alpha}\sqrt{T}}$, which is achieved at $\alpha = T^{-2/3}$ because it is a monotone decreasing function in $\alpha$. So the convergence rate is always no better than that in the case of $\alpha \geq T^{-2/3}$.

- When $\alpha \geq T^{-2/3}$, $\frac{1}{\alpha T} \leq \sqrt{\alpha}$. We only need to minimize $\sqrt{\alpha} + \frac{a_0}{\sqrt{\alpha T}}$. Then $\frac{a_0}{\sqrt{T}}$ is the global minimizer but there are constraints $T^{-2/3} \leq \alpha \leq 1$.

- When $a_0 \leq T^{-1/6}$, we have that $\frac{a_0}{\sqrt{T}} \leq T^{-2/3}$. Then we need to choose $\alpha = T^{-2/3}$ and the rate is $\sigma_{\mathcal{H}} T^{-1/3} + \sqrt{\Delta_0 L_{\|\cdot\|_{\mathcal{H}}}(f)} T^{-1/6} \leq 2\sigma_{\mathcal{H}} T^{-1/3}$.

- When $a_0 \geq \sqrt{T}$, we have that $\frac{a_0}{\sqrt{T}} \geq 1$. Then we need to choose $\alpha = 1$ and the rate is $\sigma_{\mathcal{H}} + \sqrt{\Delta_0 L_{\|\cdot\|_{\mathcal{H}}}(f)} T^{-1/2} \leq 2\sqrt{\Delta_0 L_{\|\cdot\|_{\mathcal{H}}}(f)} T^{-1/2}$.

- When $T^{-1/6} \leq a_0 \leq \sqrt{T}$, we have that $T^{-2/3} \leq \frac{a_0}{\sqrt{T}} \leq 1$. We can choose $\alpha = \frac{a_0}{\sqrt{T}}$. The rate becomes $(\Delta_0 L_{\|\cdot\|_{\mathcal{H}}}(f))^{1/4} \sqrt{\sigma_{\mathcal{H}}} T^{-1/4}$.

This completes the proof. $\qquad\square$

### F.2 CONVERGENCE RATE UNDER GENERAL-NORM ASSUMPTION

**Definition F.3** (Distortion of norms). *For any target norm $\|\cdot\|$ and a reference norm $\|\cdot\|_{\mathsf{ref}}$ on a finite-dimensional vector space, the corresponding distortion of $\|\cdot\|$ with respect to $\|\cdot\|_{\mathsf{ref}}$ is defined as*

$$\psi(\|\cdot\|, \|\cdot\|_{\mathsf{ref}}) := \sup_{\boldsymbol{x} \neq 0} \frac{\|\boldsymbol{x}\|}{\|\boldsymbol{x}\|_{\mathsf{ref}}} \cdot \sup_{\boldsymbol{x} \neq 0} \frac{\|\boldsymbol{x}\|_{\mathsf{ref}}}{\|\boldsymbol{x}\|}. \tag{55}$$

*Since all norms on the finite-dimensional space are equivalent, $\psi(\|\cdot\|, \|\cdot\|_{\mathsf{ref}})$ is always finite.*

**Theorem 4.6.** *For any $\epsilon \geq 0$, $\alpha \in (0,1)$, $\eta > 0$, and $T \in \mathbb{N}$, let $\{\boldsymbol{x}_t\}_{t=0}^T$ be the iterates of Algorithm 3 with any norm $\|\cdot\|$. Suppose Assumption 4.2 holds with the gradient variance $\sigma_{\|\cdot\|_*}(\{f_t\}_{t=1}^T)^2 \leq \sigma_{\|\cdot\|_*}^2$ for some $\sigma_{\|\cdot\|_*} \in [0, \infty)$. Then it holds that*

$$\mathbb{E} \frac{1}{T} \sum_{t=0}^{T-1} \|\nabla f(\boldsymbol{x}_t)\|_* \leq \frac{\Delta_0}{\eta T} + \frac{2\eta}{\alpha} L_{\|\cdot\|}(f) + \frac{2}{\alpha T} \sigma_{\|\cdot\|_*} + 2\sigma_{\|\cdot\|_*} \cdot \min\left(1, \alpha^{1/2}\psi(\|\cdot\|_*, \|\cdot\|_2)\right)$$

*where $\psi(\|\cdot\|_*, \|\cdot\|_2) = \sup_{\boldsymbol{x}} \frac{\|\boldsymbol{x}\|_*}{\|\boldsymbol{x}\|_2} \cdot \sup_{\boldsymbol{x}} \frac{\|\boldsymbol{x}\|_2}{\|\boldsymbol{x}\|_*}$ measures the distortion between the two norms.*

*Proof of Theorem 4.6.* Since $\mathbb{E}[\|\boldsymbol{m}_0 - \nabla f(\boldsymbol{x}_0)\|_*] \leq \sigma_{\|\cdot\|_*}$, it follows from Proposition F.2 that

$$\frac{1}{T} \sum_{t=0}^{T-1} \mathbb{E} \|\nabla f(\boldsymbol{x}_t)\|_* \leq \frac{\Delta_0}{\eta T} + \frac{L_{\|\cdot\|}(f)\eta}{2} + \frac{2\sigma_{\|\cdot\|_*}}{\alpha T} + \frac{2(1-\alpha)\eta}{\alpha} L_{\|\cdot\|}(f)$$

$$+ \frac{2\alpha}{T} \sum_{t=0}^{T-1} \mathbb{E} \left\| \sum_{i=1}^t (1-\alpha)^{t-i}(\boldsymbol{g}_i - \nabla f(\boldsymbol{x}_i)) \right\|_*. \tag{56}$$

There are two ways to control the last terms, and we discuss them separately below.

For the first approach, we directly apply triangle inequality to get

$$\alpha \cdot \mathbb{E} \left\| \sum_{i=1}^t (1-\alpha)^{t-i}(\boldsymbol{g}_i - \nabla f(\boldsymbol{x}_i)) \right\|_* \leq \mathbb{E} \sum_{i=1}^t \alpha(1-\alpha)^{t-i} \|\boldsymbol{g}_i - \nabla f(\boldsymbol{x}_i)\|_*$$

$$\leq \sum_{i=1}^t \alpha(1-\alpha)^{t-i} \left[\mathbb{E} \|\boldsymbol{g}_i - \nabla f(\boldsymbol{x}_i)\|_*^2\right]^{1/2} \leq \sigma_{\|\cdot\|_*}. \tag{57}$$

For the second approach, we first convert $\|\cdot\|_*$ to $\|\cdot\|_2$ and get

$$\mathbb{E}\left[\left\| \alpha \sum_{i=1}^t (1-\alpha)^{t-i}(\boldsymbol{g}_i - \nabla f(\boldsymbol{x}_i)) \right\|_*^2\right] \leq \left(\sup_{\boldsymbol{x}} \frac{\|\boldsymbol{x}\|_*}{\|\boldsymbol{x}\|_2}\right)^2 \cdot \mathbb{E}\left[\left\| \sum_{i=1}^t \alpha(1-\alpha)^{t-i}(\boldsymbol{g}_i - \nabla f(\boldsymbol{x}_i)) \right\|_2^2\right]$$

$$= \left(\sup_{\boldsymbol{x}} \frac{\|\boldsymbol{x}\|_*}{\|\boldsymbol{x}\|_2}\right)^2 \cdot \sum_{i=1}^t \alpha^2(1-\alpha)^{2(t-i)} \cdot \mathbb{E}[\|\boldsymbol{g}_i - \nabla f(\boldsymbol{x}_i)\|_2^2]$$

$$\leq \left(\sup_{\boldsymbol{x}} \frac{\|\boldsymbol{x}\|_*}{\|\boldsymbol{x}\|_2}\right)^2 \left(\sup_{\boldsymbol{x}} \frac{\|\boldsymbol{x}\|_2}{\|\boldsymbol{x}\|_*}\right)^2 \cdot \sum_{i=0}^t (1-\alpha)^{2i}\alpha^2 \mathbb{E}[\|\boldsymbol{g}_i - \nabla f(\boldsymbol{x}_i)\|_*^2]$$

$$\leq \psi(\|\cdot\|_*, \|\cdot\|_2)^2 \cdot \sigma_{\|\cdot\|_*}^2 \cdot \alpha.$$

Using Jensen's inequality, this further implies that

$$\alpha \cdot \mathbb{E}\left\|\sum_{i=1}^{t}(1-\alpha)^{t-i}(\boldsymbol{g}_i - \nabla f(\boldsymbol{x}_i))\right\|_* \leq \alpha^{1/2}\psi(\|\cdot\|_*, \|\cdot\|_2) \cdot \sigma_{\|\cdot\|_*}. \tag{58}$$

Now, by taking the minimum over the two bounds in (57) and (58), we obtain from (56) that

$$\frac{1}{T}\sum_{t=0}^{T-1}\mathbb{E}\|\nabla f(\boldsymbol{x}_t)\|_* \leq \frac{\Delta_0}{\eta T} + \frac{L_{\|\cdot\|}(f)\eta}{2} + \frac{2\sigma_{\|\cdot\|_*}}{\alpha T} + \frac{2(1-\alpha)\eta}{\alpha}L_{\|\cdot\|}(f)$$

$$+ 2\sigma_{\|\cdot\|_*} \cdot \min\left(1, \alpha^{1/2}\psi(\|\cdot\|_*, \|\cdot\|_2)\right).$$

This completes the proof. $\qquad\square$

### F.3 Dimension dependent lower bound for NSD

In this subsection, we construct the example for the lower bound in Theorem 4.7.

**Lemma F.4.** *Fix any $\epsilon > 0$. For any distinct points $x_0, x_1, \ldots, x_T \in \mathbb{R}$ where $T + 1 \leq \frac{1}{4\epsilon^2}$, there exists a function $p : \mathbb{R} \to \mathbb{R}$, such that its derivative $p'$ is 1-Lipschitz, $p(x_0) - \inf_x p(x) \leq 1$ and $p'(x_t) = -\epsilon$ for $t = 0, 1, \ldots, T$.*

*Proof of Lemma F.4.* We use $g(x)$ to denote $p'(x)$ in the proof. For any set of points $\{x_t\}_{t=0}^{T} \subseteq \mathbb{R}$, we will construct $g(x)$ such that $g(x) = -\epsilon$ for $x < x_0$ and $g(x + \frac{1}{\epsilon}) = g(x)$ for $x \geq x_0$. We first explain what it requires to construct this function $g$: For such a $g$, because of the periodicity, we can always find $x_0 = y_0 \leq y_1 \leq \cdots \leq y_{T'} < x_0 + \frac{1}{\epsilon}$ for $T' \leq T$ so that for any $x_t \geq x_0$, there is a $y_s$ such that there exists $n \in \mathbb{N}$ satisfying $x_t = y_s + \frac{1}{\epsilon} \cdot n$ and therefore $g(x_t) = g(y_s)$. For any $x_t < x_0$, $g(x_t) = -\epsilon$ by definition. Also for $x \leq x_0$, it holds that $p(x_0) \leq p(x)$ and $g(x)$ is 1-Lipschitz. Then it suffices to construct $g(x)$ in the interval $[x_0, x_0 + \frac{1}{\epsilon})$ to satisfy the following conditions

- $g(x)$ is 1-Lipschitz in $[x_0, x_0 + \frac{1}{\epsilon}]$.

- For any $x > x_0$, it can be written as $x = x_0 + \delta + n\epsilon^{-1}$ for some $\delta \in [0, \epsilon^{-1})$ and non-negative integer $n$, and thus $p(x) - p(x_0) = \int_{x_0}^{x_0+\delta} g(x)\mathrm{d}x + n\int_{x_0}^{x_0+\epsilon^{-1}} g(x)\mathrm{d}x$. Therefore, to ensure that $p(x_0) - \inf_x p(x) \leq 1$, we require $p(x_0) - p(x_0 + \delta) = \int_{x_0}^{x_0+\delta} g(x)\mathrm{d}x \geq -1$ for any $0 \leq \delta \leq \frac{1}{\epsilon}$ and $\int_{x_0}^{x_0+\frac{1}{\epsilon}} g(x)\mathrm{d}x \geq 0$.

- $g(y_s) = -\epsilon$ for $s = 0, 1, \ldots, T'$.

We will further define $y_{T'+1} = x_0 + \frac{1}{\epsilon}$ in the following construction.

For each $t = 0, 1, \ldots, T'$, we construct $g$ on the interval $[y_t, y_{t+1}]$ as follows:

$$g(x) = \begin{cases} -\epsilon + (x - y_t), & x \in [y_t, (y_t + y_{t+1})/2]; \\ -\epsilon - (x - y_{t+1}), & x \in ((y_t + y_{t+1})/2, y_{t+1}]. \end{cases}$$

It is straightforward to see that $g$ is 1-Lipschitz on $[x_0, x_0 + \epsilon^{-1}]$ and $g(y_t) = -\epsilon$ for $t = 0, 1, \ldots, T' + 1$. Also, note that $g(x) \geq -\epsilon$ for all $x \in [x_0, x_0 + \epsilon^{-1}]$. Thus, for any $0 \leq \delta \leq \frac{1}{\epsilon}$, it always holds that $\int_{x_0}^{x_0+\delta} g(x)\mathrm{d}x \geq -\epsilon\delta \geq -1$. Then it only remains to show $\int_{x_0}^{x_0+\frac{1}{\epsilon}} g(x)\mathrm{d}x \geq 0$.

By direct calculation, we have

$$\int_{x_0}^{x_0+\frac{1}{\epsilon}} g(x)\mathrm{d}x = \sum_{t=0}^{T'} \int_{y_t}^{y_{t+1}} g(x)\mathrm{d}x = \sum_{t=0}^{T'} \left( -\epsilon(y_{t+1}-y_t) + \frac{(y_{t+1}-y_t)^2}{4} \right)$$

$$= -1 + \frac{1}{4}\sum_{t=0}^{T'}(y_{t+1}-y_t)^2$$

$$\geq -1 + \frac{1}{4}\frac{(\sum_{t=0}^{T'}(y_{t+1}-y_t))^2}{T'+1}$$

$$= -1 + \frac{1}{4\epsilon^2(T'+1)}$$

where the inequality holds by Cauchy-Schwarz inequality. Since $T'+1 \leq T+1 \leq \frac{1}{4\epsilon^2}$, it immediately follows that $\int_{x_0}^{x_0+\frac{1}{\epsilon}} g(x)\mathrm{d}x \geq 0$. This completes the proof. $\qquad\square$

**Theorem 4.7.** *For any fixed $\Delta_0, L, \sigma^2, d, T$, learning rate $\eta$, and any averaging parameter $\alpha$, there exists a loss function $f : \mathbb{R}^d \to \mathbb{R}$, a sequence of stochastic iid loss functions $f_0, f_1, \cdots, f_{T-1}$ and an initialization $\boldsymbol{x}_0$ satisfying the following conditions:(1) $f(\boldsymbol{x}_0) - \inf_{\boldsymbol{x}} f(\boldsymbol{x}) = \Delta_0$ and $L_{\|\cdot\|_\infty}(f) \leq L$; (2) For any $\boldsymbol{x} \in \mathbb{R}^d$, it holds that $\mathbb{E}[\nabla f_t(\boldsymbol{x})] = \nabla f(\boldsymbol{x})$ and $\mathbb{E}[\|\nabla f_t(\boldsymbol{x}) - \nabla f(\boldsymbol{x})\|_1^2] \leq \sigma^2$.*

*When running Algorithm 3 with $\|\cdot\| = \|\cdot\|_\infty$, learning rate $\eta$, averaging parameter $\alpha$ and initialization $\boldsymbol{x}_0 = \mathbf{0}$, it holds that*

$$\mathbb{E}\left[\min_{t\in[T]} \|\nabla f(\boldsymbol{x}_t)\|_1\right] = \min\{e^{-2}5^{-\frac{1}{4}}(dL\Delta_0\sigma^2)^{\frac{1}{4}}T^{-\frac{1}{2}}, e^{-2}5^{-\frac{1}{2}}\sigma\}$$

*Proof of Theorem 4.7.* Inspired by Lemma 1 in Chewi et al. (2023), we first rescale the original problem to a parameter-free scaling for simplicity. Indeed, for any $f, \{f_t\}_{t=0}^{T-1}, \boldsymbol{x}_0$ satisfying the conditions (a) and (b), we can define $h(\boldsymbol{x}) = \Delta_0^{-1} \cdot f(\sqrt{\Delta_0/L}\boldsymbol{x})$ and $h_t(\boldsymbol{x}) = \Delta_0^{-1} \cdot f_t(\sqrt{\Delta_0/L}\boldsymbol{x})$. Then it can be verified that $h$ satisfies that $h(\boldsymbol{x}_0) - \inf_{\boldsymbol{x}} h(\boldsymbol{x}) = 1$ and $L_{\|\cdot\|_\infty}(h) \leq 1$. Therefore, it suffices to construct $h$ and associated stochastic loss functions $h_0, h_1, \ldots, h_{T-1}$ such that $h(\boldsymbol{x}_0) - \inf_{\boldsymbol{x}} h(\boldsymbol{x}) = 1$, $L_{\|\cdot\|_\infty}(h) \leq 1$ and $\mathbb{E}[\|\nabla h_t(\boldsymbol{x}) - \nabla h(\boldsymbol{x})\|_1] \leq \sigma/\sqrt{L\Delta_0}$, and then the construction is completed with the reverse transform $f(\boldsymbol{x}) = \Delta_0 \cdot h(\sqrt{L/\Delta_0}\boldsymbol{x})$ and $f_t(\boldsymbol{x}) = \Delta_0 \cdot h_t(\sqrt{L/\Delta_0}\boldsymbol{x})$.

Correspondingly, we can rescale the learning rate and the iterates of Algorithm 3 on the loss function $f$ to transform $\{\boldsymbol{x}_t\}_{t=0}^T$ to be iterates obtained by running Algorithm 3 on the loss function $h$: It can be easily checked that $\boldsymbol{x}_t = \sqrt{\Delta_0/L}\tilde{\boldsymbol{x}}_t$, where $\{\tilde{\boldsymbol{x}}_t\}_{t=0}^{T-1}$ are the iterates obtained by running Algorithm 3 on $h_0, \ldots, h_{T-1}$ with learning rate $\eta\sqrt{L/\Delta_0}$ and momentum factor $\alpha$. Then if it holds that $\mathbb{E}[\min_{t=0}^{T-1} \|\nabla h(\tilde{\boldsymbol{x}}_t)\|_1] = \min\left\{e^{-2}5^{-\frac{1}{4}}(d\sigma'^2)^{1/4}T^{-1/2}, e^{-2}5^{-\frac{1}{2}}\sigma'\right\}$ where $\sigma' = \sigma/\sqrt{L\Delta_0}$, we immediately have

$$\mathbb{E}\left[\min_{t=0,1,\ldots,T-1} \|\nabla f(\boldsymbol{x}_t)\|_1\right] = \mathbb{E}\left[\min_{t=0,1,\ldots,T-1} \sqrt{L\Delta_0} \cdot \|\nabla h(\tilde{\boldsymbol{x}}_t)\|_1\right]$$

$$= \min\{e^{-2}5^{-\frac{1}{4}}(dL\Delta_0\sigma^2)^{\frac{1}{4}}T^{-\frac{1}{2}}, e^{-2}5^{-\frac{1}{2}}\sigma\}$$

which gives the desired result in Theorem 4.7. Given the above discussion, we only need to construct a loss function $f : \mathbb{R}^d \to \mathbb{R}$ and associated stochastic loss functions $f_0, f_1, \ldots, f_{T-1}$ such that $f(\boldsymbol{x}_0) - \inf_{\boldsymbol{x}} f(\boldsymbol{x}) = 1$, $L_{\|\cdot\|_\infty}(f) \leq 1$ and $\mathbb{E}[\|\nabla f_t(\boldsymbol{x}) - \nabla f(\boldsymbol{x})\|_1^2] \leq \sigma^2$ for all $t = 0, \ldots, T-1$, and show that

$$\mathbb{E}\left[\min_{t=0,\ldots,T-1} \|\nabla f(\boldsymbol{x}_t)\|_1\right] = \min\left(e^{-2}5^{-\frac{1}{4}}\frac{d^{\frac{1}{4}}\sigma^{\frac{1}{2}}}{T^{\frac{1}{2}}}, e^{-2}5^{-\frac{1}{2}}\sigma\right)$$

Below we present a construction for such a hard instance.

**Construction of loss functions.** We consider initialization $x_0 = 0$. For convenience, we define the target quantity as

$$\epsilon = \min\left(\frac{d^{1/4}\sigma^{1/2}}{5^{1/4}T^{1/2}}, \frac{\sigma}{\sqrt{5}}\right). \tag{59}$$

We first construct a sequence of independently random noise vectors $\boldsymbol{\delta}_0, \boldsymbol{\delta}_1, \ldots, \boldsymbol{\delta}_{T-1}$ as follows. For constant $C = \frac{\sigma^2}{5\epsilon}$ and constant $\theta = \frac{5\epsilon^2}{d\sigma^2}$, each random vector $\boldsymbol{\delta}_t = -C[R_{t,1}, \cdots, R_{t,d}]$ where iid variables $R_{t,i} \sim \text{Bernoulli}(\theta)$. The Bernoulli distribution is well-defined because $\theta \leq \frac{1}{d} \leq 1$ by the choice of $\epsilon$ in Eq. 59. Then we know that $\mathbb{E}[\boldsymbol{\delta}_t] = -C\theta \mathbf{1}_d = -\frac{\epsilon}{d}\mathbf{1}_d$,

$$\mathbb{E}[\|\boldsymbol{\delta}_t\|_1] = d\mathbb{E}[|\delta_{t,1}|] = Cd\theta = \epsilon$$

and

$$
\begin{aligned}
\mathbb{E}\left[\|\boldsymbol{\delta}_t - \mathbb{E}[\boldsymbol{\delta}_t]\|_1^2\right] &= C^2\mathbb{E}\left[\left(\sum_{i=1}^d |R_{t,i} - \theta|\right)^2\right] \\
&= C^2\left[d\mathbb{E}[|R_{t,1} - \theta|^2] + d(d-1)\mathbb{E}[|R_{t,1} - \theta|]^2\right] \\
&= C^2\left[d\left[\theta(1-\theta)^2 + (1-\theta)\theta^2\right] + d(d-1)[2\theta(1-\theta)]^2\right] \\
&= C^2\left[d\theta(1-\theta) + d(d-1)4\theta^2(1-\theta)^2\right] \\
&\leq C^2\left(d\theta + 4d^2\theta^2\right) \\
&\leq 5C^2 d\theta = \sigma^2
\end{aligned}
\tag{60}
$$

The last inequality holds because of $d\theta \leq 1$ by the definition of $\theta$. And we can verify that $\mathbb{E}[\|\boldsymbol{\delta}_t\|_1] = \epsilon$ and $\mathbb{E}\left[\|\boldsymbol{\delta}_t - \mathbb{E}[\boldsymbol{\delta}_t]\|_1^2\right] \leq 5C^2 d\theta = \sigma^2$ by plugging in $C = \frac{\sigma^2}{5\epsilon}$ and $\theta = \frac{5\epsilon^2}{d\sigma^2}$.

Next, we construct the loss function $f$. Consider the set of points $S_{\eta,\epsilon} = \{0, \eta, 2\eta, \ldots, (N-1)\eta\}$ with $N = \lfloor\frac{1}{4\epsilon^2}\rfloor$. We then invoke Lemma F.4 with set $S_{\eta,\epsilon}$ and level $-\epsilon$ to obtain a one-dimensional function $p$, such that its derivative $p'$ is 1-Lipschitz, $p(0) - \inf_x p(x) \leq 1$ and $p'(k\eta) = -\epsilon$ for all $k \leq N$. Now we define $f : \mathbb{R}^d \to \mathbb{R}$ as

$$f(\boldsymbol{x}) = \frac{1}{d}\sum_{i=1}^d p(x_i) \tag{61}$$

Leveraging the properties of $p$, we can show that for any $\boldsymbol{x}, \boldsymbol{x}' \in \mathbb{R}^d$,

$$f(\boldsymbol{x}_0) - \inf_{\boldsymbol{x}} f(\boldsymbol{x}) \leq \frac{1}{d}\sum_{i=1}^d \left(p(0) - \inf_x p(x)\right) \leq 1,$$

$$\|\nabla f(\boldsymbol{x}) - \nabla f(\boldsymbol{x}')\|_1 = \frac{1}{d}\sum_{i=1}^d |p'(x_i) - p'(x_i')| \leq \|\boldsymbol{x} - \boldsymbol{x}'\|_\infty,$$

where the last inequality holds because $p'$ is 1-Lipschitz. This shows that $f$ satisfies the desired condition $(a)$. Then for each $t = 0, 1, \ldots, T-1$, we define the stochastic loss function $f_t$ as

$$f_t(\boldsymbol{x}) = f(\boldsymbol{x}) + \langle\boldsymbol{\delta}_t, \boldsymbol{x}\rangle + \frac{\epsilon}{d}\langle\mathbf{1}_d, \boldsymbol{x}\rangle. \tag{62}$$

Its gradient is given by

$$\nabla f_t(\boldsymbol{x}) = \nabla f(\boldsymbol{x}) + \boldsymbol{\delta}_t + \frac{\epsilon}{d}\mathbf{1}_d.$$

By the previous construction of $\boldsymbol{\delta}_t$, it is clear that $\mathbb{E}[\nabla f_t(\boldsymbol{x})] = \nabla f(\boldsymbol{x})$, and it follows from (60) that the variance of $\nabla f_t(\boldsymbol{x})$ under $\|\cdot\|_1$ satisfies $\mathbb{E}[\|\nabla f_t(\boldsymbol{x}) - \nabla f(\boldsymbol{x})\|_1^2] = \mathbb{E}[\|\boldsymbol{\delta}_t - \mathbb{E}[\boldsymbol{\delta}_t]\|_1^2] \leq \sigma^2$. Therefore, the stochastic loss functions $f_0, f_1, \ldots, f_{T-1}$ satisfy the condition $(b)$. Next, we proceed to establish the lower bound for $\mathbb{E}[\min_{t=0}^{T-1}\|\nabla f(\boldsymbol{x}_t)\|_1]$, where $\boldsymbol{x}_0, \boldsymbol{x}_1, \ldots, \boldsymbol{x}_{T-1}$ are obtained by running Algorithm 3 on the constructed loss functions $f_0, f_1, \ldots, f_{T-1}$ with $\boldsymbol{x}_0 = 0$.

**Lower bound on gradient norm.** First note that by the construction of the loss functions and the choice $x_0 = 0$, the coordinate-wise dynamics have the same distribution across coordinates, i.e., $(x_{0,i}, x_{1,i}, \ldots, x_{T-1,i})$ has the same distribution as $(x_{0,j}, x_{1,j}, \ldots, x_{T-1,j})$ for all $i, j \in [d]$. Utilizing this observation, we have

$$
\begin{aligned}
\mathbb{E}\Big[ \min_{t \in [T]} \|\nabla f(\boldsymbol{x}_t)\|_1 \Big] &= \mathbb{E}\Big[ \min_{t \in [T]} \frac{1}{d} \sum_{i=1}^d |p'(x_{t,i})| \Big] \\
&\geq \frac{1}{d} \sum_{i=1}^d \mathbb{E}\Big[ \min_{t \in [T]} |p'(x_{t,i})| \Big] \\
&= \mathbb{E}\Big[ \min_{t \in [T]} |p'(x_{t,1})| \Big].
\end{aligned}
\tag{63}
$$

So it suffices to focus on the dynamics in the first coordinate.

By the construction of the function $p$ in the previous step, we have $p'(0) = -\epsilon$. Therefore, when $x_{t,1} = 0$, we have $[\nabla f_t(\boldsymbol{x})]_1 = p'(x_1)/d + \delta_{t,1} + \epsilon/d = \delta_{t,1}$. Consequently, since $x_{0,1} = 0$, the first coordinate of $\boldsymbol{x}_t$ will remain to be zero until at some step $\delta_{t,1} \neq 0$, in which case we must have $\delta_{t,1} = -C$. This implies that

$$
\min\{t \in [T] : x_{t,1} \neq 0\} - 1 = \min\Big\{ t \in [T] : [\nabla f_t(\boldsymbol{x}_t)]_1 = -C \Big\} =: \tau,
$$

and this stopping time $\tau$ follows a geometric distribution with parameter $\theta = \frac{5\epsilon^2}{d\sigma^2}$ again by the distribution of $\boldsymbol{\delta}_t$. Hence,

$$
\begin{aligned}
\mathbb{P}\Big[\tau \geq \frac{1}{\theta}\Big] = \mathbb{P}\Big[\tau \geq \lceil \frac{1}{\theta} \rceil \Big] &= (1 - \theta)^{\lceil \frac{1}{\theta} \rceil} \\
&\geq \exp\Big( -\theta \lceil \frac{1}{\theta} \rceil \Big) \\
&\geq e^{-1-\theta} \geq e^{-2}.
\end{aligned}
$$

Now conditioned on the event that $\tau \geq \frac{1}{\theta}$, we can show that

$$
\tau + N \geq \frac{1}{\theta} + \frac{1}{4\epsilon^2} \geq 2\sqrt{\frac{1}{\theta} \cdot \frac{1}{4\epsilon^2}} = \frac{1}{\sqrt{\theta}\epsilon} = \frac{\sqrt{d}\sigma}{\sqrt{5}\epsilon^2},
$$

which is no smaller than $T$ by the choice of $\epsilon$ in Eq. 59. Therefore, we have $x_{t,1} = 0$ for all $t = 0, 1, \ldots, \tau$ and there are at most $T - \tau$ distinct points among $x_{0,1}, x_{1,1}, \ldots, x_{T,1}$. Then since $|x_{t+1} - x_t|$ is either $0$ or $\eta$ by the update rule of Algorithm 3 with $\|\cdot\| = \|\cdot\|_\infty$, we see that $x_{0,1}, x_{1,1}, \ldots, x_{T,1} \in \{0, \eta, 2\eta, \ldots, (N-1)\eta\}$ on the event $\tau \geq \frac{1}{\theta}$, in which case we have $p'(x_{t,1}) = -\epsilon$ for all $t = 0, 1, \ldots, T$ by our construction of $p$. Leveraging this, it follows from Eq. 63 that

$$
\begin{aligned}
\mathbb{E}\Big[ \min_{t \in [T]} \|\nabla f(\boldsymbol{x}_t)\|_1 \Big] \geq \mathbb{E}\Big[ \min_{t \in [T]} |p'(x_{t,1})| \Big] &\geq \mathbb{P}\Big[\tau \geq \frac{1}{\theta}\Big] \cdot \mathbb{E}\Big[ \min_{t \in [T]} |p'(x_{t,1})| \,\Big|\, \tau \geq \frac{1}{\theta}\Big] \\
&\geq e^{-2}\epsilon
\end{aligned}
$$

This gives the desired lower bound and completes the proof. $\qquad\square$

