# OpenReview forum: "A Tale of Two Geometries: Adaptive Optimizers and Non-Euclidean Descent"
_ICLR.cc/2026/Conference — ICLR 2026 Poster_

### Official Review · Reviewer_aLQH · 2025-10-29

**Soundness:** 3
**Presentation:** 3
**Contribution:** 3
**Rating:** 8
**Confidence:** 4

**Summary:**

The paper relaxes the convexity assumption in the analysis of adaptive optimizers. It also introduces new assumptions in the analysis.

**Strengths:**

The main contribution is relaxing convexity assumption accompanied by the technical results.

Other contributions consider different assumptions in the analysis.
For acceleration the authors use the recently introduced smoothness in Kovalev’s analysis, but convexity is here. Despite the assumption used in other papers I do not consider its incorporation as a significant contribution – Kovalev’s assumptions are standard and more widely used in contrast to the considered assumption which was introduced very recently. But still it enables new rates.

**Weaknesses:**

- New variance assumption improves dimension dependence. But no surprise since the assumption is different. I mean the improvements are not because of improved analysis or new techniques. The discussion is given in Lines 450-454 and states that the dependence is unavoidable in the worst case.

Detailed discussion is needed. How does the adaptive variance assumption relate to the standard one? Does it hold in practice?

- Some theorems do not properly refer to the assumptions used.

- Line 465: 1/T^-3.5. \varepsilon?

**Questions:**

Line 053: the adaptive smoothness (cf. Definition 2.2), introduced by Xie et al. (2025b)
it not called adaptive smoothness in the origin (at least I could not find). It is confusing. Usually the algorithms are adaptive to smoothness. Please, clarify the naming. It is assumption, does it adapt?

---

> ### Author Response · Authors · 2025-11-21
>
> **Q1:** How does the adaptive variance assumption relate to the standard one? Does it hold in practice?
>
> **A1:** Intuitively, the standard gradient variance under $||\cdot||\_\mathcal{H}$ finds the locally best $H\in\mathcal{H}$ for each $x$, and in contrast the adaptive gradient variance finds the best global best $H\in\mathcal{H}$ simultaneously for all $x$. This is the intuition behind Proposition 4.2. Moreover, it can be checked that the adaptive gradient variance for $\mathcal{H}$ is at most $d$ times the standard gradient variance under $||\cdot||_\mathcal{H}$ and we have edited proposition 4.2 to rigorously show this. Therefore, when we assume that the standard gradient variance is finite, the corresponding adaptive gradient variance is also finite, though it can be $d$ times larger in the worst case.
>
> Furthermore, many existing works assume that the gradient variance is finite under $||\cdot||_2$. It can also be checked that the standard gradient variance with respect to any general well-structured preconditioner set $\mathcal{H}$ is at most $d$ times the standard gradient variance under $||\cdot||_2$. Taken together, this justifies the gradient variance assumptions.
>
> ---
>
> **Q2:** Clarify the naming of adaptive smoothness.
>
> **A2:** The adaptive smoothness was named as $\mathcal{H}$-smoothness in Xie et al., (2025b). We rename it as adaptive smoothness because it governs the convergence rates for adaptive optimizers in both the convex and nonconvex setting. The original name $\mathcal{H}$-smoothness is not informative enough to convey its connection to the adaptive optimizers so we decide to rename it in our work to fit into the story.
>
> ---
>
> **Q3:** Some theorems do not properly refer to the assumptions used.
>
> **A3:** We have examined all the theorem statements. We noticed a typo in Lemma 3.4 and realized that we did not explain $x^\*$ is the global minima in Theorem 4.4 and that we require $||x^*||_\mathcal{H} \leq D$ in Theorem C.5. We have corrected them in the revision. Please let us know if you find any other theorem statements ambiguous.
>
> ---
>
> **Q4:** Typo in line 465.
>
> **A4:** Thank you for catching the typo. We have corrected it accordingly.

---

### Official Review · Reviewer_MwWV · 2025-10-31

**Soundness:** 3
**Presentation:** 3
**Contribution:** 3
**Rating:** 4
**Confidence:** 2

**Summary:**

The paper compares standard non-Euclidean smoothness (governing NSD/Lion/Muon-style methods) with a stronger adaptive smoothness that underpins Adam/Adagrad/Shampoo-type optimizers. It extends adaptive smoothness to the nonconvex setting (unified analysis for well-structured preconditioners), proves accelerated $O(\tilde{T}^{-2})$ convex rates for adaptive methods (unattainable under standard $\ell_{\infty}$ smoothness), and introduces adaptive variance that yields dimension-free NSD rates in stochastic nonconvex optimization.

**Strengths:**

- Two genuinely different smoothness/variance regimes with algorithmic consequences. The paper formalizes why “adaptive optimizers is approximately NSD with a norm” is not the full story: adaptive methods converge under adaptive smoothness $\Lambda_{\mathcal H}(f)$, which is always $\ge$ the standard smoothness used by NSD (Prop. 2.3). Under adaptive smoothness, accelerated $O(\tilde T^{-2})$ convergence with Nesterov momentum is achieved (Thm. 4.4), while standard $\ell_\infty$-smoothness faces a $\Omega(T^{-1}/\log T)$ barrier (via prior lower bounds cited by the authors). The analogous adaptive variance notion similarly strengthens stochastic guarantees (Def. 4.1), yielding dimension-free NSD rates (Thm. 4.6) that are provably unattainable under only $\|\cdot\|_{\mathcal H,\ast}$-variance (Thm. 4.7). This idea is interesting and well substantiated by aligned theorems.
- Unified nonconvex analysis for adaptive preconditioners beyond diagonal cases. Theorem 3.2 (and its corollaries) covers weighted/EMA/cumulative updates over well-structured preconditioner sets $\mathcal H$ (matrix subalgebras), extending nonconvex analysis beyond diagonal preconditioners that prior works largely required. The matrix inequality (Lemma 3.4) controls  via  $\mathrm{Tr}(H)\,\|S_T\|_{\mathrm{op}}$ and explicates where extra $\log d$ factors enter for non-commutative $\mathcal H$. This step makes unified treatment possible and should be of independent interest.
-  Results are parameterized in terms of the geometry $\mathcal H$ (e.g., diagonal vs. general PSD cones), update style (weighted/EMA/cumulative), and budgeted noise control (minimizing $\mathrm{Tr}(P_{\mathcal H}(\Sigma))$ or $\sigma_{\mathcal H}(f)$). The two-phase behavior (deterministic $\tilde O((\Delta_0\Lambda_{\mathcal H}/T)^{1/2})$ vs. stochastic $\tilde O(T^{-1/4})$) and explicit \emph{log-factors} can be useful for practitioners anticipate when acceleration/normalization helps and when geometry-induced constants dominate.

**Weaknesses:**

- The main guarantees hinge on adaptive quantities—$\Lambda_{\mathcal H}(f)$ (minimizing trace bounds over a preconditioner set) and $\sigma_{\mathcal H}(f)$ (sup over $x,t$ with minimization over $H\in\mathcal H$). These are hard to estimate or upper-bound in realistic deep-learning settings. The paper discusses relationships to bounded covariance (e.g., via $\mathrm{Tr}(P_{\mathcal H}(\Sigma))$) but stops short of estimators/diagnostics practitioners could compute to check assumptions or guide geometry choice. Without empirical proxies (e.g., online estimates of $P_{\mathcal H}(\widehat\Sigma)$, curvature sketches, or trace surrogates), it’s difficult to translate the theory into optimizer selection.
- While Theorem 4.4 achieves an appealing $O(\tilde T^{-2})$ term, the bound carries \emph{$\log d$ penalties} and depends on a radius $D=\max_t\|x_t-x^\ast\|_{\mathcal H}$ (mitigated by a projected variant). The paper provides a projection-based algorithm with the same rate, but the \emph{constants and projection effects} (e.g., clipping bias, implementation overhead) are not explored. Similarly, non-commutative $\mathcal H$ introduces unavoidable $\log d$ blow up via Lemma 3.4; it would help to quantify when these factors remain small (e.g., block-diagonal or Kronecker-structured $\mathcal H$) or to show empirical results indicating the log-factors are not rate-limiting in practice.

**Questions:**

-  The paper compares to NSD’s standard-smoothness rates and highlights separations, but the discussion of near-optimality (w.r.t. $d$, $\tilde T$, and geometry) is mostly qualitative. For instance, are the nonconvex \emph{$\tilde O(T^{-1/4})$} stochastic adaptive rates minimax-optimal under the proposed adaptive variance? Are the $\log d$ factors necessary for broad non-commutative $\mathcal H$, or artifacts of proof technique?
- Could you specify practical projection oracles for common H (diagonal, Kronecker, spectral)?
- Can you propose online, low-cost estimators (e.g., streaming sketches of gradient covariances projected onto H or trace estimators for $P_H(\Sigma)$? to upper-bound $\Lambda_H(f)$ and $\sigma_H(f)$ during training? A brief “estimation recipe” would make the theory far more deployable and allow users to select H adaptively.

---

> ### Author Response · Authors · 2025-11-21
>
> We are happy that you find our claims about different geometry are well supported by our theoretical results and we appreciate that you acknowledge the significance of our unified proof. Also thanks for your questions that makes the broader story more compelling. We will address you concerns below.
>
> **Q1:** Efficient estimators for adaptive quantities.
>
> **A1:** Given our results that the convergence rate of adaptive optimizers is governed by the adaptive smoothness, we totally agree that it is an important question to develop efficient empirical estimators for such quantities to provide guidance for selecting the optimal optimizer in practice. However, this is beyond the scope of the current work, as we aim to develop theoretical understanding of the interplay between optimization algorithms and geometry of the loss landscape.
>
> Nonetheless, we are happy to discuss and propose some estimators for adaptive smoothness and adaptive variance for some specific choice of well-structured preconditioner set. According to the definitions the adaptive smoothness and adaptive gradient variance, there are two levels of difficulty for estimating these quantities:
>
> 1. First we need to estimate the local smoothness/variance, e.g., for each $x$. This is difficult when the dimension $d$ is very large. It is computationally infeasible to exactly solve $\min\_{H \in \mathcal{H}, -\nabla^2 f(x) \preceq H\preceq \nabla^2 f(x)} \text{Tr}(H)$, and instead we can use tools of Hessian vector product implemented in standard package like PyTorch. As for $P_\mathcal{H}(\Sigma)$ and adaptive variance, it is also computationally exhaustive to iterate over all the training samples to compute the stochastic gradients. We can only approximate the distribution with a small number of samples. The covariance matrix itself is also too big to explicitly compute and we need to leverage matrix vector product in a similar way of computing the adaptive smoothness.
> 2. The second level of difficulty is to aggregate the local estimates to get a global estimate, as we need to take supremum over all $x$. For example, it is well known to be hard to estimate the standard $\ell_2$ smoothness globally for loss functions in deep learning. While it might be possible that we can aggregate estimates from several checkpoints to get a better estimate for the global metric.
>
> Despite the difficulties above, we will mention some estimators for some specific examples of well-structured preconditioner set. [1] proposed estimators for two specific $\mathcal{H}=$ {$c \cdot I_d \mid c\geq 0$} and $\mathcal{H}=\text{diagonal PSD matrices}$. As for the adaptive variance $\sigma\_\mathcal{H}(f)$, we will also estimate it with the current stochastic gradient function $\nabla f_t(x)$. We will define estimators for three specific $\mathcal{H}$ below. For more general $\mathcal{H}$, we find it hard to define estimators without specifying the structure of $\mathcal{H}$ and leave it for future work.
>
> - For $\mathcal{H}=$ {$c \cdot I_d \mid c\geq 0$}, we can compute $\sigma_\mathcal{H}(f)^2=\min_{H \in \mathcal{H}, \text{Tr}(H)\leq 1} \mathbb{E} ||\nabla f(x)-\nabla f_t(x)||_{H^{-1}}^2=d \mathbb{E} ||\nabla f(x)-\nabla f_t(x)||_2^2$ because the only feasible $H$ is $\frac{1}{d} I_d$.
> - For $\mathcal{H}=$ diagonal PSD matrices, $\sigma\_\mathcal{H}(f)^2=\min_{H\in \mathcal{H}, \text{Tr}(H)\leq 1} \mathbb{E} ||\nabla f(x)-\nabla f_t(x)||\_{H^{-1}}^2=\min_{\sum_{i=1}^d h_i=1, h_i\geq 0}\sum_{i=1}^d \frac{1}{h_i} \mathbb{E} (\nabla_i f(x)-\nabla_i f_t(x))^2 =[\sum_{i=1}^d \sqrt{\mathbb{E} (\nabla_i f(x)-\nabla_i f_t(x))^2}]^2$.
> - For $\mathcal{H}=$ PSD matrices, $\sigma_\mathcal{H}(f)^2=\text{Tr}(\Sigma^{1/2})^2$. It is hard to obtain the entire covariance matrix and then compute square root because of the huge size of the matrix. [2] proposed an efficient estimator for $\text{Tr}(\Sigma^{1/2})$ given the access to the stochastic gradient vector.
>
> [1] Xie, Shuo, Mohamad Amin Mohamadi, and Zhiyuan Li. "Adam Exploits $\ell_\infty $-geometry of Loss Landscape via Coordinate-wise Adaptivity." *arXiv preprint arXiv:2410.08198* (2024).
>
> [2] Ubaru, Shashanka, Jie Chen, and Yousef Saad. "Fast estimation of tr(f(A)) via stochastic Lanczos quadrature." *SIAM Journal on Matrix Analysis and Applications* 38.4 (2017): 1075-1099.

---

> > ### Author Response · Authors · 2025-11-21
> >
> > **Q2:** Practical projection oracles for common $\mathcal{H}$**.**
> >
> > **A2:** We completely agree that it is also an important question to develop practical projection oracles for common $\mathcal{H}$. In particular, we provide the following two examples:
> >
> > - When $\mathcal{H}=$ {$c\cdot I_d\mid c\geq 0$}, $\text{argmin}\_{||x||\_{\mathcal{H}}\leq D}||x-y||\_{V_t}^2=\text{argmin}\_{||x||_2 \leq \sqrt{d} D} ||x-y||_2^2=\frac{y}{||y||_2} \min(||y||_2, \sqrt{d} D)$.
> > - When $\mathcal{H}=$ diagonal PSD matrices, $\text{argmin}\_{||x||\_{\mathcal{H}}\leq D} ||x-y||\_{V_t}^2=\text{argmin}\_{||x||\_\infty \leq D} ||x-y||_{V_t}^2$. Because $V_t$ is a diagonal matrix, we can optimize each coordinate independently, which is achieved by clipping $y_i$ into $[-D,D]$.
> >
> > For more general $\mathcal{H}$, we are not aware of efficient ways to perform projection into the norm balls under $||\cdot||_\mathcal{H}$. We leave this as an interesting direction for future work, as we focus on the iteration complexity (i.e., the number of times for computing the gradient) in the current work.
> >
> > ---
> >
> > **Q3:** Are the $\log d$ factors necessary for broad non-commutative $\mathcal{H}$?
> >
> > **A3:** This is a great question. It is indeed a key technical contribution of ours to develop Lemma 3.4 for non-commutative $\mathcal{H}$, which further relies on a novel matrix inequality stated in Lemma E.1 for the non-commutable case. The additional $\log d$ factor comes from the proof of Lemma E.1, and we do not know how to improve this in general.
> >
> > ---
> >
> > **Q4:** Optimality of the convergence guarantees.
> >
> > **A4:** It is an important question to investigate whether the convergence rate is optimal. In particular, regarding the number of steps $T$, [3] showed that the lower bound $\Omega(T^{-1/4})$ for all stochastic first-order algorithms, which include the adaptive algorithm defined in our work. Thus our rate is optimal in terms of $T$. We leave it for future work to study the optimality of dependence on other quantities.
> >
> > [3] Arjevani, Yossi, et al. "Lower bounds for non-convex stochastic optimization." *Mathematical Programming* 199.1 (2023): 165-214.

---

### Official Review · Reviewer_E5SU · 2025-11-03

**Soundness:** 4
**Presentation:** 2
**Contribution:** 3
**Rating:** 6
**Confidence:** 4

**Summary:**

This paper analyzes a class of adaptive preconditioning algorithms for convex and non-convex, stochastic optimization under an assumption on the objective function called adaptive smoothness. The non-convex section essentially extends the analysis of Xie at el (2025a) from the convex to the non-convex setting (Theorems 3.2 and 3.3), which requires a lot of technical work for the case of non-diagonal preconditioners. The convex section extends the accelerated $1/T^2$ result of Kovalev (2025a) from diagonal preconditioners to a more general class of preconditioners (Theorem 4.4), analyzes normalized steepest descent (NSD) under an adaptive noise assumption (Theorem 4.6), and provides a lower bound of sign descent (i.e. NSD w.r.t. $\ell_\infty) with a standard, non-adaptive noise assumption.

**Strengths:**

1. The paper is well-written and contributes to an important problem, namely understanding the improved performance of preconditioning based optimizers.
2. There are a lot of theoretical results, and from what I can tell, these results are not simple repetitions of known ideas. In particular, the results in the non-convex setting rely on matrix inequalities that involve a good bit of technical work. I checked the proofs of these matrix inequalities (Lemma 3.4 and all lemmas it relies on), and they appear correct.
3. Several of the results demonstrate a clear separation between regimes: the accelerated $1/T^2$ rate under adaptive smoothness (Theorem 4.4) improves on the known $1/T$ lower bound under $\ell_\infty$ smoothness. Also, the dimension-free rate of NSD under adaptive noise (Theorem 4.6) improves on the dimension-dependent lower bound under non-adaptive noise (Theorem 4.7).

**Weaknesses:**

1. There are several examples of exaggerated or inaccurate language related to the contribution of this work. I feel that the paper would be stronger if the writing was more direct, transparent, and objective about the contributions and the relationship to previous work. Below are some examples.

    1a. The paragraph on lines 64-71 seems to misrepresent the authors' contribution. The paragraph states that the contribution is to show the accelerated $1/T^2$ rate for preconditioned algorithms, but really the contribution is to extend the results of Kovalev (2025a) from diagonal pre-conditioners to a more general class. I think Kovalev (2025a) should definitely be cited here, and it should be clarified that the current paper extends the proof techniques of Kovalev (2025a).

    1b. Definition 2.2 should cite Xie et al (2025b). Under the current presentation, it is easy for the reader to mistakenly believe that the adaptive smoothness from Definition 2.2 is a novel concept introduced by the authors, and indeed I got this impression after reading the introduction and first few pages. I recommend that the authors clearly point out when they are building on previous work. This is especially important during the high-level discussion about the power of different smoothness notions: it is easy for the reader to mistakenly get the idea that all of these high-level concepts were introduced by the authors, when really (in my opinion) the main contribution of this work is technical (e.g. matrix inequalities to handle non-commutative preconditioner classes), not conceptual.

    1c. Lines 83-86 state that "our results demonstrate that adaptive smoothness and adaptive variance are not simply stronger versions of their standard counterparts but profoundly different conditions that unlock qualitatively stronger guarantees". How is $1/T^2$ qualitatively stronger than $1/T$ and not just quantitatively stronger? To me, this kind of language comes off as over-selling the work, when really the technical contributions speak for themselves. I encourage the authors to spend less time in the draft discussing the "profound" implications of their work and more time on direct, transparent discussion.

    1d. It is stated that Lemma 3.4 may be of independent interest. Given that Lemma 3.4 is a bound on terms specific to the algorithm considered in this paper, I don't see how this result could possibly be of interest outside the context of this algorithm.

2. Assumption 3.1 seems to require almost surely bounded noise, which is stronger than previous work. Is this a typo in the statement of Assumption 3.1, or does the proof actually require almost surely bounded noise? If it is the latter, then there needs to be some discussion of this fact and point out that stronger results are only achieved under stronger assumptions. I would also appreciate if the authors discuss why this stronger assumption is necessary in the proof.

**Questions:**

Questions:
1. Do the results in stochastic setting actually require almost surely bounded noise, as in Assumption 3.1? (see Weakness #2).

Small suggestions:
- It should be added as a condition to Lemma E.2 that $X$ and $A$ are diagonalizable, since the proof uses an eigendecomposition for both of these matrices. Similar conditions apply in Lemma E.1 for $X$ and $X-Y$. This does not affect the proof, since the analysis only applies these lemmas for symmetric matrices.

---

> ### Author Response · Authors · 2025-11-21
>
> Thanks for acknowledging the significance of our theoretical contribution and those suggestions on improving the writing quality. We will improve the presentation accordingly and we’d like to address your concerns below.
>
> **Q1:** The paragraph on lines 64-71 seems to misrepresent the authors’ contribution.
>
> **A1:** Thank you for the suggestion. We will cite the paper by Kovalev (2025a) in the introduction and clarify that the current proof extends the technique therein. Besides, we have mentioned in Section 4 that our technical result is an extension of Kovalev 2025a (line 344) and that the adaptive noise assumption is also inspired by Kovalev 2025a (line 327). There are also extensive comparison with it in Section 4.2 (line 382-391).
>
> ---
>
> **Q2:** Definition 2.2 should cite Xie et al (2025b)
>
> **A2**: Thank you for the suggestion. We will clearly mention in Definition 2.2 that the adaptive smoothness definition is from Xie et al (2025b) to avoid confusion. Besides, we have mentioned in line 53 that this definition is introduced in Xie et al (2025b).
>
> ---
>
> **Q3:** Wording of lines 83-86.
>
> **A3:** Thank you for the suggestion. We have adjusted the writing accordingly by discussing the results in a more direct way.
>
> ---
>
> **Q4:** How the result of Lemma 3.4 could be of interest outside the context of this algorithm?
>
> **A4:** We agree that the statement of Lemma 3.4 is specifically targeted at this optimization algorithm. However, this unified algorithm family already incorporates a wide range of popular algorithms including AdaGrad, Adam, full-matrix AdaGrad, one-sided Shampoo and we believe this lemma can be useful for future works on such algorithms.
>
> Another analogy is that many works for AdaGrad-Norm (Ward et al., 2020) and Adam (Defossez et al., 2022) used the result $\sum_{l=1}^T \frac{a_l}{\sum_{i=1}^l a_i} \leq \log(\sum_{i=1}^T a_i)+1$ and $\sum_{l=1}^T \frac{a_l}{\epsilon+\sum_{i=1}^l \beta_2^{l-i}a_i} \leq \log(1+\frac{\sum_{i=1}^T \beta_2^{T-i} a_i}{\epsilon})-N\ln(\beta_2$) for nonnegative $a_1, …, a_T$  to control the second order term. In this sense, lemma 3.4 can also be separated into two parts. The current first paragraph holds for general vector $g_1, …, g_T$ that are not necessarily the stochastic gradients from optimization process. Only the second paragraph that bounds $S_T$ requires additional property of $g_1, …, g_T$.
>
> Moreover, Lemma 3.4 relies on an important matrix inequality stated in Lemma E.1, which is not stated specifically for the adaptive algorithm and thus may be of independent interest in the broader context.
>
> In the revision, we have modified the discussion accordingly by clarifying why this technical contribution may be of independent interest.
>
> ---
>
> **Q5:** Do the results in stochastic setting actually require almost surely bounded noise as in Assumption 3.1?
>
> **A5:** Our current analysis for general adaptive algorithms in the stochastic nonconvex setting requires the assumption that the noise is almost surely bounded. This condition is only required in the proof of Lemma B.3, which controls the gap between using stochastic gradient and deterministic gradient, and the difficulty to extend it to the setting under a covariance assumption comes from the non-exchangeability of matrix multiplication. Besides, Lemma B.5 also relies on the noise assumption, but it still holds under an assumption on the noise covariance. In addition, we would like to clarify that the correct assumption 3.1 is $-\Sigma \preceq \nabla f(x_t) \nabla f(x_t)^\top -\nabla f_t(x_t) \nabla f_t(x_t)^\top \preceq \Sigma$ almost surely.
>
> We have revised the paper accordingly by adding related discussions and highlighting the changes in the main text and the steps that rely on the noise assumption in the appendix. We leave the improvement regarding the noise assumption to future work, as the goal of the current Section 3 is to show that the adaptive smoothness governs the convergence rate of adaptive optimizers.
>
> ---
>
> **Q6:** It should be added as a condition to Lemma E.2 that $\mathbf{A}$ and $\mathbf{X}$ are diagonalizable.
>
> **A6:** We would like to clarify that when we say $\mathbf{X}$ is positive definite and $\mathbf{A}$ is positive semi-definite, we are implicitly saying that they are symmetric matrices, which further implies that they are diagonalizable. We will make this explicit in the statement of Lemma E.2.
>
> [1] Ward, Rachel, Xiaoxia Wu, and Leon Bottou. "Adagrad stepsizes: Sharp convergence over nonconvex landscapes." *Journal of Machine Learning Research* 21.219 (2020): 1-30.
>
> [2] Défossez, Alexandre, et al. "A simple convergence proof of adam and adagrad." *arXiv preprint arXiv:2003.02395* (2020).

---

> > ### Comment · Reviewer_E5SU · 2025-11-24
> >
> > Thank you for addressing my concerns, I believe that the changes will improve the transparency of the paper. I have increased my score to accept.

---

### Official Review · Reviewer_7CSx · 2025-11-05

**Soundness:** 4
**Presentation:** 4
**Contribution:** 3
**Rating:** 6
**Confidence:** 3

**Summary:**

This paper provides a rigorous theoretical separation between adaptive optimizers (like Adam) and normalized steepest descent (NSD) (like SignGD). The authors' central thesis is that these two algorithm families utilize different assumptions, even when operating in the same non-Euclidean geometry. The paper shows that NSD's convergence is governed by standard smoothness ($L\_{||\cdot||\_{\mathcal{H}}}(f)$), while adaptive optimizers are governed by a provably stronger (more restrictive) "adaptive smoothness" ($\Lambda\_{\mathcal{H}}(f)$). The authors justify this distinction by showing that this stronger assumption is what allows adaptive optimizers to achieve Nesterov acceleration, a feat that is provably impossible under the standard smoothness assumption alone. This "tale of two notions" is extended to the stochastic setting with the introduction of "adaptive variance" (a more precise, structure-aware noise measure) and is complemented by a new unified convergence proof for adaptive methods in the non-convex setting.

**Strengths:**

- By identifying the two distinct smoothness notions and linking the stronger "adaptive smoothness" to the ability to achieve acceleration (which is provably impossible for the weaker, standard smoothness), the paper provides a clear separation.
- The authors' parallel "adaptive variance" argument is well-supported, as they provide not just a pessimistic, dimension-dependent upper bound for standard variance (Prop D.4) but also a corresponding lower bound (Thm 4.7) to prove that this $d$-dependence is unavoidable.
- Beyond its main thesis, the paper provides a strong technical contribution by delivering the first unified non-convex convergence analysis for adaptive optimizers using general well-structured preconditioner sets.

**Weaknesses:**

- I am not fully clear about the distinction between adaptive methods and NSD, as the author claims they exploit different notions of smoothness. For instance, under the adaptive smoothness assumption, NSD could potentially also achieve the same rate as in Theorem 3.3, and when combined with the acceleration technique in Eq. (4), could possibly also attain an $O(1/T^2)$ rate for convex functions. Conversely, adaptive methods could also match the rate of NSD under $L\_{\\|\cdot\\|\_{\mathcal{H}}}(f)$ smoothness.
- Regarding the theoretical results, Assumption 3.1 appears stronger than the standard boundedness-in-expectation condition. It requires that the noise be bounded for every realization of randomness. Additionally, Theorem 3.2 does not appear to guarantee convergence; could the authors clarify why this is the case?
- Finally, the cited lower bound for non-acceleration applies to $L\_{\\|\cdot\\|\_{\infty}}(f)$. It remains unclear whether this separation holds for a general $\mathcal{H}$.

**Questions:**

Please refer to the weaknesses section.

In addition:
- Is there a function that is not adaptively smooth but still has a finite $L\_{\\|\cdot\\|\_{\mathcal{H}}}(f)$? Otherwise, the two assumptions would characterize the same class of functions, differing only by constants.
- Regarding the practical implications, it seems that full-matrix AdaGrad should outperform Adam, since its corresponding $\mathcal{H}$ is a superset of that for Adam, leading to smaller adaptive smoothness. Is there any empirical evidence supporting this?

---

> ### Author Response · Authors · 2025-11-21
>
> Thanks for acknowledging that our theoretical results clearly explains the difference between geometries specialized by smoothness and noise variance. We are also happy that you find our unified proof technically strong and interesting. We will address your concerns below.
>
> **Q1**: Distinction between adaptive methods and NSD.
>
> **A1:** We would like to clarify that the convergence rates of adaptive methods and NSD are governed by the two different smoothness notions. Specifically, for a loss function $f$, the convergence rate of the adaptive optimizer with well-structured preconditioner set $\mathcal{H}$ is governed by the adaptive smoothness $\Lambda\_\mathcal{H}(f)$, while for the NSD under the $||\cdot||\_\mathcal{H}$ norm, its convergence rate is determined by the standard smoothness $L_{||\cdot||_\mathcal{H}}(f)$.
>
> Next, we address the reviewer’s three comments on the relationship between adaptive methods and NSD.
>
> 1. Under the setting of Theorem 3.3 (let us consider the deterministic setting for simplicity), the convergence rate of NSD under $||\cdot||\_\mathcal{H}$ is of order $O(\sqrt{\Delta_0 L\_{||\cdot||\_\mathcal{H}}(f)/T})$, and the convergence rate for the corresponding adaptive optimizer is of order $O(\sqrt{\Delta_0 \Lambda_\mathcal{H}(f)/T} \log d)$. Now if we assume that the adaptive smoothness $\Lambda_\mathcal{H}(f)\leq L$ for some $L>0$, then without any additional assumption on the standard smoothness $L_{||\cdot||\_\mathcal{H}}(f)$, we only have $L_{||\cdot||\_\mathcal{H}}(f) \leq \Lambda\_\mathcal{H}(f)\leq L$ because of proposition 2.3. In this case, the upper bounds for the convergence rates of NSD and the adaptive optimizer are the same up to an additional $\log d$ factor. Indeed, we can see that NSD actually has a better upper bound. This distinction corresponds to the claim that NSD and adaptive methods exploit different notions of smoothness.
> 2. For convex functions, we are not aware of results that show NSD can achieve the accelerated $O(1/T^2)$ rate by using the technique in Equation (4).
> 3. Meanwhile, we are also not aware of results that adaptive methods can match the rate of NSD under $||\cdot||\_\mathcal{H}$ smoothness for general $\mathcal{H}$ and loss functions. Our results show that the convergence rate of the adaptive optimizer depends on the adaptive smoothness $\Lambda\_\mathcal{H}(f)$, which can be arbitrarily larger than the standard smoothness $L_{||\cdot||\_\mathcal{H}}(f)$ as illustrated in the response to Q5 below.
>
> We would appreciate it if the reviewer can provide more details or literature references for the latter two cases.
>
> ---
>
> **Q2:** Assumption 3.1 appears stronger than the standard boundedness-in-expectation condition.
>
> **A2:** You are right that Assumption 3.1 is a relatively strong assumption, and our current proofs relies on this almost surely bounded noise assumption. Please find more details in the response to reviewer E5SU. We have added related discussion in the revision.
>
> ---
>
> **Q3:** Theorem 3.2 does not appear to guarantee convergence.
>
> **A3:** We would like to clarify that Theorem 3.2 provides a convergence guarantee for the *nonconvex* setting. In such a setting, it is a standard way to bound the averaged gradient norm. Specifically, it shows that the algorithm can output a solution whose gradient is small enough. It is impossible to guarantee that the algorithm can converge to the global minima because of nonconvexity here.
>
> ---
>
> **Q4:** It remains unclear if the non-acceleration separation holds for a general $\mathcal{H}$.
>
> **A4:** You are right that the cited lower bound applies only to the $\ell_1/\ell_\infty$ case, and it is an interesting open problem to derive such separation for general $\mathcal{H}$. We leave this for future work.

---

> > ### Author Response · Authors · 2025-11-21
> >
> > **Q5:** The function classes characterized by the two smoothness notions.
> >
> > **A5:** We agree that when $L_{||\cdot||\_\mathcal{H}}(f)<\infty$, it is true that the adaptive smoothness $\Lambda\_\mathcal{H}(f)<\infty$, and vice versa. However, we would like to emphasize that **the difference between these two quantities can depend on the dimension $d$ and thus can be arbitrarily large as $d$ grows**. We have edited proposition 2.2 to make the relationship more clear.
> >
> > We illustrate such $d$-dependence is unavoidable by a specific example of the cross entropy loss on a softmax function: We use $\sigma(x)$ to denote the softmax function when $x \in \mathbb{R}^d$ and consider $f(x)=KL(p||\sigma(x))$ where $p$ is the target distribution satisfying $\sum_{i=1}^d p_i=1$ and $p_i \propto 1/i$. Then the gradient is $\nabla f(x)=\sigma(x)-p$ and the Hessian matrix is $\nabla^2 f(x)=\text{diag}(\sigma(x))-\sigma(x) \sigma(x)^\top$. We will show the standard smoothness $L_{||\cdot||\_\mathcal{H}}(f)=O(1)$ and adaptive smoothness $\Lambda_{||\cdot||\_\mathcal{H}}(f)=\Omega(d)$ for $\mathcal{H}=\{\text{diagonal PSD matrices}\}$.
> >
> > For this specific $\mathcal{H}$,  $||\cdot||\_\mathcal{H}$ is $\ell_\infty$ norm and note that $L\_{||\cdot||\_\mathcal{H}}(f)=\sup\_{x\in \mathbb{R}^d} \sup\_{||u||\_\infty \leq 1} u^\top \nabla^2 f(x) u \leq \sup\_{x\in \mathbb{R}^d} \sup\_{||u||\_\infty \leq 1} u^\top \text{diag}(\sigma(x))u= \sup\_{x\in \mathbb{R}^d} \sup\_{||u||\_\infty \leq 1} \sum_{i=1}^d \sigma(x)\_i u_i^2 \leq \sup_{x\in \mathbb{R}^d} \sup_{||u||\_\infty \leq 1} \sum_{i=1}^d \sigma(x)_i =1.$
> >
> > But for adaptive smoothness, we can show below that it is at least $\frac{d}{2}$. According to the definition, we need to minimize $\text{Tr}(A)$ such that diagonal matrix $A=\text{diag}(a_1, \cdots, a_d) \succeq \nabla^2 f(x)$ for every $x \in \mathbb{R}^d$. For any $i<j$, we can always find a sequence of $x$ such that the limit vector $\sigma^\*$ of $\sigma(x)$ satisfies $\sigma^\*_i=\sigma^\*_j=\frac{1}{2}$ and every other entry is $0$. If $A$ needs to dominate $\text{diag}(\sigma^*)- \sigma^{\*}\sigma^{\* \top}$, it requires $x^\top Ax \geq x^\top(\text{diag}(\sigma^\*)- \sigma^\*\sigma^{\*\top})x=\frac{1}{4}x_i^2-\frac{1}{2} x_i x_j +\frac{1}{4} x_j^2$ to hold for every $x\in \mathbb{R}^d$. When only $x_i$ and $x_j$ are nonzero, it implies $a_i x_i^2+a_j^2 \geq \frac{1}{4}x_i^2-\frac{1}{2} x_i x_j +\frac{1}{4} x_j^2$, which is equivalent to $(a_i-\frac{1}{4})(a_j-\frac{1}{4})\geq \frac{1}{16}$. It implies $a_i+a_j \geq 1$ for any $i<j$ and thus $\text{Tr}(A)\geq \frac{d}{2}$.
> >
> > ---
> >
> > **Q6:** Compare the convergence rates of full-matrix AdaGrad and Adam.
> >
> > **A6:** We would like to clarify that the nonconvex convergence rate *doesn’t* suggest that full-matrix AdaGrad is better than Adam. The adaptive smoothness for full-matrix AdaGrad is indeed smaller than the adaptive smoothness for Adam. However, note that the norm used to measure the size of the gradient, $||\nabla f(x_t)||\_{\mathcal{H},*}$, is different for full-matrix AdaGrad and Adam. For full-matrix AdaGrad, it corresponds to $||\nabla f(x_t)||_2$, while for Adam, it corresponds to $||\nabla f(x_t)||_1$. Since $||\nabla f(x_t)||_2\leq ||\nabla f(x_t)||_1$ and these two norms can differ by a multiplicative factor of $d$ in the worst case, our results do not imply that full-matrix AdaGrad has a better convergence rate than Adam.

---

### Meta-Review · Area_Chair_HFrq · 2025-12-17

**Summary:**

The reviewers agree that this paper provides new insight into the theory of preconditioned methods, a timely and important topic. Besides new results, the paper contributes with new mathematical techniques. There are several weaknesses, including the almost sure nature of Assumption 3.1. Nevertheless, I believe that the reviewers agree that the paper should be accepted. I suggest the authors incorporate their rebuttal comments into the camera-ready paper.

**Reviewer Concerns:**

I think most concerns are addressed. However, the almost sure nature of Assumption 3.1, a stronger assumption, compared to the classical variance-bounded assumption, is still in the paper and cannot be improved due to the technical difficulties. Moreover, I didn't find the response to the first weakness, "New variance assumption improves dimension dependence. But no surprise since the assumption is different. I mean the improvements are not because of improved analysis or new techniques" by Reviewer aLQH.

**Reviewer Scores:**

The scores of the reviewers are mostly positive and recommend acceptance. Reviewer E5SU decided to increase the score to 8.

---

### Decision · Program_Chairs · 2026-01-26

Accept (Poster)